# Protein purification with light via a genetically encoded azobenzene side chain

Peter Mayrhofer ⓘ , Markus R. Anneser ⓘ , Kristina Schira ⓘ , Carina A. Sommer ⓘ , Ina Theobald, Martin Schlapschy, Stefan Achatz & Arne Skerra ⓘ ✉

Affinity chromatography is the method of choice for the rapid purification of proteins from cell extracts or culture supernatants. Here, we present the light-responsive Azo-tag, a short peptide comprising p-(phenylazo)-L-phenylalanine (Pap), whose side chain can be switched from its *trans*-ground state to the metastable *cis*-configuration by irradiation with mild UV light. Since only *trans*-Pap shows strong affinity to α-cyclodextrin (α-CD), a protein exhibiting the Azo-tag selectively binds to an α-CD chromatography matrix under daylight or in the dark but elutes quickly under physiological buffer flow when illuminating the column at 355 nm. We demonstrate the light-controlled single-step purification – termed Excitography – of diverse proteins, including enzymes and antibody fragments, without necessitating competing agents or harsh buffer conditions as normally applied. While affinity chromatography has so far been governed by chemical interactions, introducing control by electromagnetic radiation as a physical principle adds another dimension to this widely applied separation technique.

Affinity chromatography has revolutionized protein purification in many areas, from basic research up to biotechnological and biopharmaceutical applications. In this technique, a specific ligand or binding partner (for example, an enzyme inhibitor or a specific antibody) of the protein of interest (POI) is covalently immobilized onto a stationary phase, usually a biocompatible hydrophilic polymer such as agarose. The POI is applied as a mixed solution with various contaminants – typically from a cell extract or culture supernatant – in the buffered aqueous mobile phase whereby it forms a non-covalent complex with the immobilized ligand. After thoroughly washing out the contaminants under buffer flow, the POI is specifically eluted in a distinct dissociation step. This can be achieved either with a solution of a competing ligand or by changing the buffer conditions of the mobile phase, for example to high or low pH, elevated salt concentration, or by applying a chemical denaturant, detergent or chelating agent.

Affinity (or bioaffinity) chromatography emerged more than half a century ago as a procedure mainly for the purification of enzymes, but also of antibodies or antigens – as well as specific binding proteins such as avidin – from natural sources[1,2]. More recently, after the advent of gene technology and the possibility of overexpressing recombinant proteins in a host cell of choice, the concept of the Affinity Tag emerged. In this advanced technique, the immobilized ligand does not primarily interact with the POI itself – e.g. by targeting an enzyme active site, a ligand-binding site or an immunological paratope or epitope – but with a distinct peptide sequence that is fused to the N- or C-terminus of a POI and enables specific complex formation.

An early example was the *myc*-tag[3], a short epitope peptide used in combination with a monoclonal antibody (MAb), which can be immobilized onto the stationary phase, allowing elution of a captured tagged protein under acidic conditions. This was followed by the His-tag together with immobilized metal ion affinity chromatography (IMAC), initially applied under strongly denaturing conditions (6 M GdnHCl) with acid elution[4] and, subsequently, under physiological buffer conditions using competitive elution with imidazole[5]. Further examples include the FLAG peptide tag, utilizing a MAb that binds the peptide only in the presence of $Ca^{2+}$, thus allowing elution in the presence of EDTA[6], and the *Strep*-tag, which engages the widely applied protein reagent streptavidin, whose binding activity can be competed with a biotin derivative[7,8].

Despite wide application, including the industrial use of protein A affinity chromatography for the purification of antibody-based

Chair of Biological Chemistry, School of Life Sciences, Technical University of Munich, 85354 Freising, Germany. ✉e-mail: skerra@tum.de

biopharmaceuticals[9,10], there is a general disadvantage associated with bioaffinity chromatography: the eluted POI is usually contaminated with a ligand, or a denaturant, detergent or chelator, which may interfere with subsequent use for biochemical, cell culture or in vivo experiments. Furthermore, its functional integrity can be affected by the application of extreme pH, high salt concentration, depletion of metal ions or action of the denaturant or detergent. Therefore, the affinity-purified protein normally has to undergo an additional purification step, often size exclusion chromatography (SEC) or ion exchange chromatography (IEC), but at least a buffer exchange (e.g. with a desalting column, by dialysis or cross-flow filtration), before it is ready for experimental study or practical use.

Consequently, an elution method that does not involve any addition of reagents, or change of buffer conditions, is highly desirable as it would directly yield the isolated functional protein in the mobile phase in which it was applied – or in a desired buffer of choice. Here, we exploit a chemical group that has light-switchable shape and binding properties, azobenzene[11], which allows the specific elution of the POI from the chromatography matrix just by illumination at an appropriate wavelength. This chemical group is employed in the form of a non-canonical amino acid (ncAA), which is genetically fused to the POI as part of the short Azo-tag, in combination with a stationary phase carrying cyclodextrin (CD) groups, whose mutual interaction can be controlled by light-induced switching between the *trans*- and *cis*-configurational states of the azobenzene moiety (Fig. 1).

## Results

### *cis*/*trans*-Dependent complex formation between p-(phenyl-azo)-L-phenylalanine (Pap) and cyclodextrins (CDs)

In the search for a suitable ncAA that can be cotranslationally incorporated into recombinant proteins and shows a light-switchable

change in configuration and molecular shape (Fig. 2) we chose p-(phenylazo)-L-phenylalanine (Pap). Pap (also known as AzoPhe or AzoF) was previously incorporated in the frame of protein-functional studies into a semisynthetic bovine ribonuclease S[12], the *Escherichia coli* catabolite activator protein (CAP) as well as recombinant myoglobin from sperm whale[13] and into the subunit HisH of a glutaminase from *Thermotoga maritima*[14], for example. The recombinant proteins were produced using *E. coli* as expression host and an orthogonal pair of an engineered *Methanocaldococcus jannaschii* suppressor tRNA[CUA] and the cognate tyrosyl-tRNA synthetase which had been evolved to accept this ncAA substrate[13]. However, the spectroscopic properties of Pap have been only partially characterized up to now[13,15–17] even though numerous other azobenzene derivatives were synthesized and their photo-isomerization studied[18]. Therefore, we prepared Pap (Supplementary Fig. S2) following a published procedure[13] and investigated its absorption spectra as well as its isomerization between the *trans*- and *cis*-states in greater detail.

Pap as a free amino acid (at pH 8.0, equilibrated under daylight) in the low-energy state, i.e. in its *trans*-configuration, exhibits a pronounced absorption maximum (Fig. 2D; for a more profound spectroscopic analysis, see Supplementary Figs. S6 and S7) in the near UV region at 326 nm ($\varepsilon_{326} = 21500 \pm 411 \, M^{-1} \, cm^{-1}$) and a much weaker second absorption maximum in the violet/blue light region at ~423 nm ($\varepsilon_{423} = 1380 \pm 86 \, M^{-1} \, cm^{-1}$). A quantitative switch to the higher-energy state, i.e. the *cis*-configuration, was easily achieved by illuminating the solution in a quartz cuvette from the top (Supplementary Fig. S1) with mild UV light at 355 nm using a 1.2−2.4 mW LED for ≤30 min (Fig. 2E). The resulting solution of *cis*-Pap exhibited a prominent, less strong absorption maximum shifted to a shorter wavelength, at 293 nm ($\varepsilon_{293} = 6700 \pm 337 \, M^{-1} \, cm^{-1}$), as well as a second minor absorption maximum in the visible region at almost the same wavelength as *trans*-Pap, ~426 nm, with slightly larger amplitude ($\varepsilon_{426} = 2120 \pm 90 \, M^{-1}$

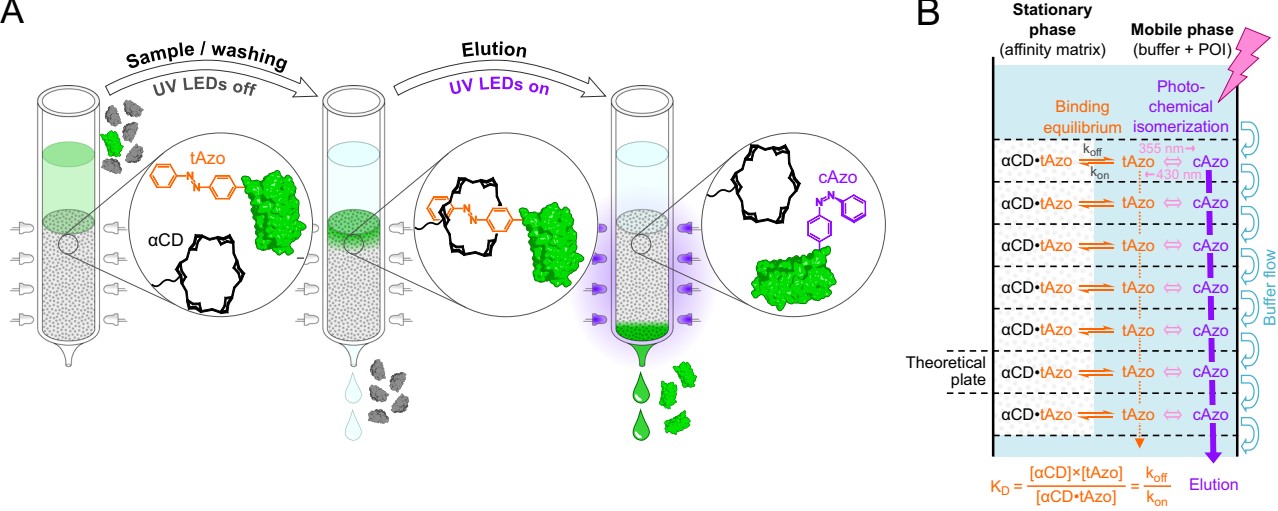

**Fig. 1 | Light-controlled protein purification via Excitography. A** A protein mixture or cell extract containing the protein of interest (POI) equipped with the Azo-tag is applied to an α-CD affinity column. With the Pap side chain in the lower-energy *trans*-configuration, the POI binds to the α-CD groups of the column matrix whereas all contaminating proteins are quickly washed out. Subsequent illumination with near UV light triggers configurational switch of the azobenzene moiety to the more bulky *cis*-state which effects desorption from the affinity matrix. Thus, the POI is eluted in the chromatography buffer of choice, without adding reagents, salts or change of pH. **B** Interplay between the dynamic chemical equilibrium of noncovalent complex formation between α-CD and the Azo-compound (either free Pap or the Azo-tagged POI) according to the Law of Mass Action and the light-induced *cis*/*trans*-isomerization of the latter. The photostationary composition comprises a variable *cis*:*trans*-ratio depending on the wavelength and mode of irradiation,

which may either stay fixed (when washing in the dark) or can be dynamic (that is, under continuous illumination). Due to the almost undetectable affinity between α-CD and the Azo-group in the *cis*-state, this isomer elutes with the buffer flow essentially without retardation (here depicted with a thick vertical arrow). On the other hand, in the *trans*-configuration – with its moderate $K_D$ value of around 100 μM – the Azo-compound is strongly retarded on the affinity matrix due to tight complex formation with the α-CD groups. However, there will always be some noticeable migration (thin vertical dotted arrow) due to the continuous re-equilibration between free and bound Azo-compound at each theoretical plate of the column (here depicted with horizontal dotted lines) during the buffer flow, which is governed by the dissociation and association kinetics. (For further details and the practical influence of illumination during the washing step, after sample load, see Supplementary Fig. S14).

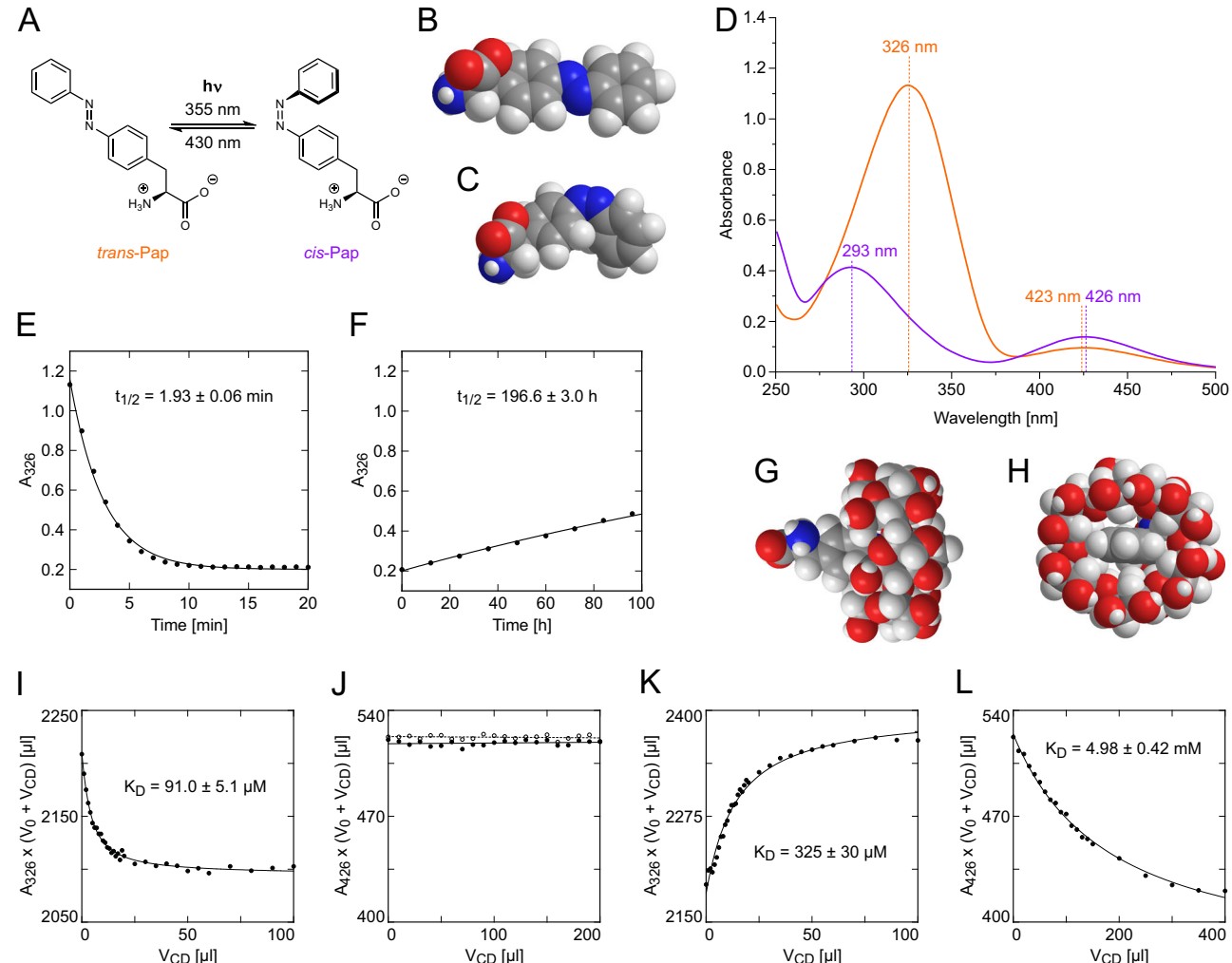

**Fig. 2 | Structure and photochemistry of Pap as well as complex formation between its *trans*- or *cis*-state and α-CD or β-CD. A** Photo-induced *cis*/*trans*-isomerization of Pap. **B, C** While the structure of the azobenzene side chain in *trans*-Pap is mostly planar and elongated (**B**), the one of *cis*-Pap is twisted and more bulky due to the steric repulsion between the two aromatic rings (**C**). **D** UV-Vis spectra of the *trans*- and *cis*-states of Pap. A 50 μM solution of Pap in 100 mM Tris/HCl pH 8.0 either equilibrated under daylight (orange) or after irradiation with 355 nm UV light for 30 min (violet), respectively. Wavelengths of absorption maxima are indicated. **E** Time-dependent increase in absorption at 326 nm during irradiation of a 50 μM *trans*-Pap solution at 355 nm with a UV LED from the top. **F** Thermal re-isomerization of *cis*-Pap (50 μM) at 25 ± 1 °C in the dark as spectrophotometrically monitored at long measurement intervals (12 h). The data in (**E**) and (**F**) were subjected to exponential curve fit (applying asymptotic values for (**F**) from (**E**)). **G, H** Structural model of the complex between *trans*-Pap and α-CD (side view and front view, respectively; energy-minimized with the MM2 method using Chem-Draw3D). **I** Titration of 50 μM *trans*-Pap in 2 mL 100 mM Tris/HCl pH 8.0 with a 50 mM solution of α-CD in the same buffer. **J** Titration of *cis*-Pap (100 μM) with α-CD (solid circles) in comparison with buffer alone (hollow circles). **K** Titration of *trans*-Pap (50 μM) with β-CD. **L** Titration of *cis*-Pap (100 μM) with β-CD. For all titrations − except for (**J**), where a straight line fit was applied − the small change (negative or positive, respectively) in absorbance at the diagnostic wavelengths of 426 nm for *cis*-Pap and 326 nm for *trans*-Pap during complex formation with the CD was monitored and the curves were fitted using Eq. 1.

cm$^{-1}$). Hence, similar to the published spectroscopic properties of plain azobenzene (*trans*: $\varepsilon_{320}$ = 22000 M$^{-1}$ cm$^{-1}$, $\varepsilon_{440}$ = 400 M$^{-1}$ cm$^{-1}$; *cis*: $\varepsilon_{270}$ = 5000 M$^{-1}$ cm$^{-1}$, $\varepsilon_{450}$ = 1500 M$^{-1}$ cm$^{-1}$)[19], Pap shows distinct absorption bands that not only allow the individual spectroscopic detection of its *trans*- and *cis*-states but, importantly, also offer two well-separated wavelengths to specifically trigger switching between these configurations: e.g., in theory, illumination at 326 nm for *trans*→*cis* and at 426 nm for *cis*→*trans*.

To more precisely assess the relative *cis*/*trans* composition of Pap in the photostationary state (PSS) upon illumination at different wavelengths we performed ¹H-NMR spectroscopy, which allowed us to individually quantify the well separated peaks in the aromatic region for both isomers (see Suppl. Results). Astonishingly, and somewhat in contrast to the expectations from the scientific literature in this field, the *cis*-state of Pap in aqueous solution can almost be quantitatively populated (around 95 %) upon irradiation by 355 nm LED light, with significantly better yield than at the wavelength of 365 nm which has been commonly used for switching Pap to the *cis*-state in other laboratories[14,17]. This is in good agreement with the high ratio $\varepsilon_{trans}$/$\varepsilon_{cis}$ = 8.3 at a wavelength of 355 nm (Supplementary Fig. S3A), assuming an approximately constant quantum yield, $\Phi_c$, for the *trans*→*cis* iso-merisation across the π→π* transition band, as described for azobenzene[20]. Conversely, only a proportion of 70−80 % *trans*-isomer was obtained in the PSS upon illumination at daylight or with mono-chromatic blue LED light at around 430 nm, which is less than might have been expected for this energetically favored configuration. Thus, we used visible light of 430 nm, or simply daylight, for switching *cis*→*trans*, which for Pap only slowly occurs via thermal relaxation (see below). Of note, both chosen wavelengths are remote from the char-acteristic absorption bands of proteins (~280 nm for the aromatic side

chains and ≤225 nm for the peptide backbone) as well as nucleic acids (~260 nm).

Next, we measured the kinetic stability of the higher-energy *cis*-state in aqueous solution after UV exposure. To this end, we prepared a 50 μM *cis*-Pap solution in 100 mM Tris/HCl pH 8.0 by illumination at 355 nm, directly in the quartz cuvette as above, and then placed this cuvette into a diode array spectrophotometer in the dark. The absorbance of the solution at 326 nm was measured at different intervals of 2–12 h – to minimize an influence of illumination at the diagnostic wavelength of ≤1 s per measurement – for up to 100 h. The rise in absorption at this wavelength indicated the increasing population of *trans*-Pap due to thermal relaxation, which could be fitted with a monoexponential decay function converging at $t_{1/2} \approx 196$ h (Fig. 2F; Suppl. Results). Remarkably, this half-life of approximately 8 days is drastically longer than the one reported for the parent compound azobenzene in organic solvent[21], which means that experiments with the almost pure *cis*-isomer of Pap, once generated, can be carried out in aqueous solution at room temperature in a practically reasonable time frame – as long as kept in the dark – and do not require permanent exposure to UV light.

While host-guest complex formation between certain azobenzene derivatives and CDs – or structurally related host cavities – was studied before, mainly in the area of nanomaterial research[22–26], no such data have been reported for Pap. Taking into consideration the limited solubility of the free Pap amino acid in Tris/HCl pH 8.0, we chose the method of spectroscopic titration of a diluted Pap solution with a concentrated CD stock solution in the same buffer and monitoring the small change in the amplitude of the corresponding absorption band (Supplementary Fig. S3C) via quick measurement in the diode array spectrophotometer, as above, for each titration step after equilibration in the dark (Fig. 2I–L). The data were fitted according to the Law of Mass Action for bimolecular complex formation (Eq. 1).

Titration of *trans*-Pap with α-CD (cyclomaltohexaose) led to a remarkably low dissociation constant, $K_D = 91$ μM (Fig. 2I). Hence, the affinity between *trans*-Pap and α-CD is stronger by an order of magnitude than could be expected from published measurements with *trans*-azobenzene as part of a polyacrylamide-based hydrogel using ¹H-NMR[23,26]. Conversely, the same measurement with *cis*-Pap led to no detectable spectroscopic effect, thus preventing the determination of a $K_D$ value for its putative complex with α-CD (Fig. 2J). Again, this was not fully expected since affinities reported for *cis*-azobenzene in a polyacrylamide hydrogel had indicated a $K_D$ value of 29 mM[23]. Further titrations of *trans*-Pap and *cis*-Pap performed here (Fig. 2K, L) using the larger β-CD (cyclomaltoheptaose, 2-hydroxypropyl-derivatized; see Suppl. Methods) resulted in $K_D = 325$ μM for the slim *trans*-state and a 15-fold higher $K_D$ value of 4.98 mM for the more bulky *cis*-configuration (cf. Fig. 2B, C). Taken together, in contrast to β-CD, whose affinities revealed a more moderate difference, α-CD appeared to exclusively form a tight complex with *trans*-Pap whereas exhibiting vanishingly low, if any, affinity towards *cis*-Pap.

In fact, these distinct binding activities of α-CD towards the light-switchable *cis*- and *trans*-states of Pap were in a range that seemed suitable for applications in affinity chromatography. For comparison, the *Strep*-tag[8] revealed a dissociation constant of $13.0 \pm 1.3$ μM for complex formation with streptavidin[27], which was sufficient to enable the one-step purification of corresponding fusion proteins from complex cell extracts. Therefore, to test the performance of Pap and its potential light-dependent binding activity under chromatographic conditions in aqueous solution, we synthesized a chromatography matrix with covalently immobilized α-CD groups at high density by chemical coupling of cyclomaltohexaose to epoxy-activated Sepharose 6B[28]. In this setting, the long hydrophilic linker provided by the activated polysaccharide matrix and its conjugation via an ether bond to one of the many hydroxy-groups on the hydrophilic surface of α-CD

were expected to ensure on average good sterical accessibility of its hydrophobic inner pocket (Supplementary Fig. S8). At the same time, the swollen gel matrix (soaked with aqueous buffer) showed a high transmission, leading to ≥50 % light intensity within a typical packed chromatography column at bench scale (Supplementary Fig. S8). Furthermore, it is well established that upon passage through an opaque medium the incoming light is scattered isotropically and that its mean path length is invariant with respect to the microstructure of this medium[29].

The resulting α-CD affinity matrix was used to prepare a chromatography column with 1 mL bed volume (7 mm diameter). When a 5 mM solution of Pap (100 μL) was applied under daylight (i.e. in its predominant *trans*-state) the ncAA immediately adsorbed to the matrix in the upper part of the column, as directly visible due to its intense yellow color (see Supplementary Fig. S9). The compound was largely retained in this zone, showing just modest movement with the flow of the mobile phase when washing the column with ten bed volumes of 100 mM Tris/HCl pH 8.0, 500 mM NaCl. This pronounced retention effect was in line with the stable (yet dynamic) complex formation between α-CD and *trans*-Pap (see Fig. 1) observed before in solution. However, when the column was illuminated at 355 nm for 10 min with a laterally positioned set of LEDs (see Supplementary Fig. 1), the yellow band became mobilized immediately and fully eluted within 1.5 bed volumes of the same buffer. Direct measurement of the absorption spectrum of the eluted fraction revealed the presence of Pap in its *cis*-state by showing only absorption bands at 293 and 426 nm and no peak at 326 nm (see Supplementary Fig. S9B). This experiment provided initial proof that the light-switchable *cis/trans*-isomerization of the ncAA Pap can be used to control its retention *versus* elution under chromatographic conditions on an α-CD affinity column. These encouraging results suggested the incorporation of Pap into recombinant proteins and to investigate if these adopt a similar light-dependent chromatographic behavior.

## Development of a genetic system for the efficient incorporation of Pap into recombinant proteins

To incorporate Pap into a recombinant POI, as part of our so-called Azo-tag, we employed the strategy of amber stop codon (TAG) suppression in *E. coli* using an orthogonal pair comprising a heterologous tRNA$^{CUA}$ and a cognate aminoacyl-tRNA synthetase (aaRS)[30] encoded by a one-plasmid system for all genetic components[31]. However, instead of the system from *M. jannaschii* adapted to the Pap substrate as used by others before[13,14], we chose the pyrrolysyl-tRNA synthetase (PylRS) from *Methanosarcina mazei* with altered substrate specificity (as described further below) and its cognate tRNA$^{Pyl}$ [32]. This naturally evolved orthogonal pair is known for its more robust amber suppression in *E. coli* and less cross-reactivity with endogenous canonical amino acids[33] while PylRS had been engineered already to accept amino acid substrates with certain substituted azobenzene side chains[34].

Our expression vector pSB19 (Fig. 3) carries in total three heterologous genes, each under the control of a different promoter. tRNA$^{Pyl}$ is constitutively expressed from the *lpp* promoter, whereas the structural gene for PylRS has been placed under control of the arabinose-inducible *ara*BAD promoter/operator (p/o). Finally, the coding region for the POI is cloned under control of the *lac*UV5$^{p/o}$ which is inducible by lactose or isopropyl-β-D-thiogalactopyranoside (IPTG)[5]. Initially, we utilized superfolder GFP (sfGFP)[35] in order to select *E. coli* cells carrying a mutated PylRS for the incorporation of Pap at its structurally permissible sequence position 39 via fluorescence-activated cell sorting (FACS)[31] (see below and Suppl. Results). After having evolved PylRS with the desired substrate specificity, the coding region for sfGFPa39 was replaced by the ones of various different POIs equipped with the Azo-tag (see Supplementary Table S1) as described further below.

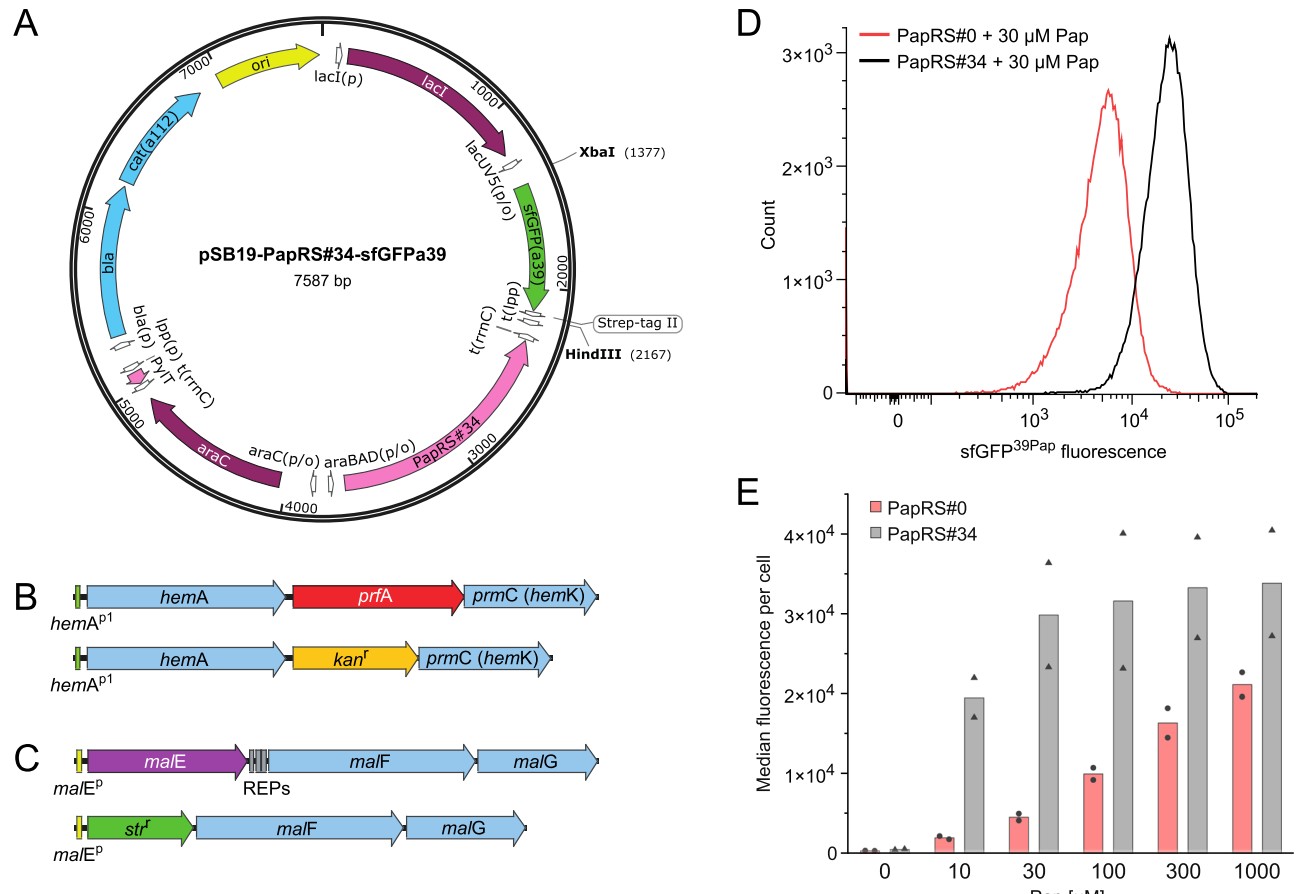

**Fig. 3 | Genetic system for the efficient cotranslational incorporation of Pap into recombinant proteins produced in *E. coli*. A** Expression vector pSB19 encoding all three components needed for the biosynthesis of a POI carrying Pap, which is encoded by an amber stop codon: (i) the coding region for the POI (here: sfGFPa39) under control of the *lac*UV5[p/o], (ii) the gene for tRNA[Pyl] carrying an anticodon complementary to the amber stop codon, expressed from the *lpp* promoter, (iii) the coding region for an engineered version of PylRS, PapRS#34, under control of the *ara*BAD[p/o]. Apart from the relevant regulatory regions (repressor genes *ara*C and *lac*I as well as origin of replication), the vector additionally harbors a transcriptional fusion of the genes for β-lactamase and chloramphenicol-acetyl-transferase, which confer ampicillin (Amp) and chloramphenicol (Cam) resistance, respectively. The latter was equipped with an amber stop codon at a permissible position (residue 112), thus allowing selection for *E. coli* growth in the presence of Cam during the directed evolution of PylRS for accepting a ncAA substrate. **B** Replacement of the *prf*A cistron within the operon flanked by the *hem*A and *prm*C

(alias *hem*K) genes on the chromosome of the *E. coli* B strain NEBExpress by the coding region for the kanamycin resistance protein, which was accompanied by a gene duplication (see Suppl. Results), thus leading to diminished transcriptional activity for RF-1 (the relevant promoter, P1, is indicated). **C** Subsequent deletion of the *mal*E cistron, as part of the *mal* operon, in the genome of NEBExpress(lowRF1) and its replacement by the coding region for the streptomycin resistance protein, resulting in NEBExpress(lowRF1/ΔMBP). **D**, **E** Directed evolution of PylRS to efficiently charge tRNA[Pyl] with Pap. **D** Exemplary FACS measurements (sfGFP[39Pap] signals, 10[5] events each) comparing the initial PylRS mutant encoded on pSB15-PapRS#0 (red) with the pSB19-PapRS#34 plasmid (black), both harboring the sfGFPa39 reporter protein gene, in the NEBExpress(lowRF1) strain background. **E** MFI of sfGFPa39 expression determined by FACS (10[5] events each; N = 2 individual experiments) in the presence of different Pap concentrations in the culture medium, using the same experimental setup.

To boost the efficiency of amber stop codon suppression by tRNA[Pyl], which is counter-acted by the ribosomal release factor 1 (RF-1) of *E. coli*, we attempted to construct an expression strain in which its gene, *prf*A, is deleted (Fig. 3B). Based on an earlier report on the inactivation of RF-1 in *E. coli* suggesting that RF-1 is indispensable for *E. coli* K-12 strains, due to the lower activity of its mutated RF-2 allele[36], we chose the *E. coli* B strain NEBExpress® (New England Biolabs, Ipswich, MA), which encodes an intact RF-2, as chassis strain. To this end, the *prf*A cistron within the *hem*A-*prf*A-*prm*C(*hem*K) operon was precisely replaced by the coding region for the kanamycin resistance protein (*kan*[r]) using the λ-Red recombination system[37]. However, careful genome sequence analysis of the single clone obtained indicated that a gene duplication had occurred in a way that the remaining copy of intact *prf*A suffered from lower transcriptional activity (see Suppl. Results). Notably, the resulting strain, NEBExpress(lowRF1), showed essentially unaffected growth but strongly enhanced amber suppression activity – approximately 50 % *versus* the typically 10−20 %

suppression in a wild-type background – in combination with the pSB19 vector, as also demonstrated by a five-fold higher yield of a fluorescent reporter protein containing the Pap residue after purification via the C-terminal *Strep*-tag II (see Suppl. Results). Thus, this *E. coli* derivative was used during the development of a PylRS mutant enabling the cotranslational incorporation of Pap and for the subsequent affinity purification experiments.

The evolution of a Pap-specific PylRS mutant started from the *M. mazei* enzyme which had been modified to incorporate *meta*-substituted derivatives of Pap for applications in click chemistry (MmPSCaaRS, harboring four amino acid exchanges from the wild-type protein: A302T, L309S, N346V, and C348G)[34]. We prepared a synthetic gene for this enzyme while additionally introducing a conservative amino acid mutation, K192R, which also occurs in the highly homologous PylRS from *M. barkeri*[38], thus generating a second *Bsa*I restriction site (together with the one introduced via silent mutations at the amino acid position G423/L424). This gene/protein format

(dubbed MmPSCaaRS(BsaI), initially cloned on the vector pSB15 carrying different promoter elements, see Methods) facilitated cassette mutagenesis and subcloning of the central coding region encompassing the amino acid substrate pocket and active site of the modified PylRS.

Next, the mutation Y384F, which enhances ncAA incorporation by PylRS over a wide range of substrates comprising extended Lys side chains[39], was introduced (mutant PapRS#0), resulting in higher activity both towards the unsubstituted Pap and its *para*-amino derivative, NH₂-Pap (4.4-fold and 2.5-fold increase, respectively, according to the median of the fluorescence intensity, MFI, determined by FACS analysis of cells expressing the sfGFPa39 reporter protein). NH₂-Pap was tentatively employed as a better soluble, albeit chemically less stable (cf. Supplementary Fig. S17) derivative of Pap. Notably, the latter showed immediate precipitation when diluted from its alkaline stock solution into the culture medium of *E. coli*. Interestingly, this phenomenon was effectively prevented by preparing a stock solution of 50 mM Pap in 200 mM NaOH supplemented with 200 mM β-CD, thus taking advantage of its reversible complex formation with both the *cis*- and *trans*-states of Pap, as determined above.

To further improve the incorporation of Pap (and also of NH₂-Pap; see Suppl. Results) by PapRS#0, its directed evolution was pursued (Supplementary Fig. S10). To this end, the central PylRS-encoding region encompassing residues D196–E425 (flanked by the pair of *Bsa*I restriction sites) was amplified by error-prone polymerase chain reaction (PCR), thus generating a mutated plasmid library. Transformants of NEBExpress(lowRF1) were grown in the presence of 1 mM of the ncAA and subjected to FACS using the sfGFPa39 fluorescence as readout. After several iterations of positive and negative selection[31], emerging mutants revealing specifically enhanced fluorescence were analyzed by plasmid sequencing and compared by FACS analyses of individually cultured clones. Promising amino acid exchanges were subsequently combined by subcloning, finally resulting in the mutant PapRS#34 which carried three additional amino acid exchanges: F295L, N304S and V346A.

The combined advantages of the evolved PapRS#34, the expression vector pSB19 and the improved genetic background of NEBExpress(lowRF1) became evident when examining the amount of Pap in the culture medium that was needed to generate the maximal signal of sfGFPa39 in the FACS measurements. Signal saturation was approached already at a very low Pap concentration of 30–100 μM (Fig. 3D, E), compared to a 1 mM concentration as applied at the onset of this project as well as in other published studies[14,34]. This also made the use of β-CD for solubilizing Pap in the bacterial culture less important in our hands. Consequently, this optimized expression system was employed for the preparative production of various POIs (Supplementary Table S1) to test their performance in a light-controlled affinity chromatography.

### Affinity purification of Azo-tagged proteins on the α-CD affinity column controlled by light

To investigate the application of the light-switchable Pap side chain for purification purposes if incorporated at an exposed position into a biosynthetic POI, initially two colored model proteins were chosen: Azurin, a natural blue copper protein from *Pseudomonas aeruginosa*[40], and mScarlet(3), an engineered monomeric red fluorescent protein from corals[41,42]. To avoid possible steric interference of the interaction between α-CD and Pap in the context of the macromolecular protein, the amber stop codon was positioned at the very C-terminus of the reporter protein. Separated by two slim Gly residues, the *Strep*-tag II affinity peptide[8] was arranged upstream for convenience, whereas Pap was directly followed downstream by a terminal ochre stop codon (TAA, not suppressed by the PylRS system; Fig. 4A). Apart from serving as an additional spacer between Pap and the target protein, the *Strep*-tag II also allowed the initial purification of the POI by Strep-Tactin affinity chromatography, prior to assessing its chromatographic behavior mediated by the Azo-tag in combination with the α-CD affinity column.

While mScarlet was produced in the cytoplasm of *E. coli*, Azurin was secreted into the bacterial periplasm, with the help of the OmpA signal peptide, and subsequently reconstituted with Cu²⁺ ions in the periplasmic extract[7]. The coding regions for Azurin and mScarlet, both equipped with the combined *Strep*/Azo-tag at their C-termini, were each cloned on the pSB19-PapRS#34 vector – thus replacing sfGFPa39 which had served as fluorescent reporter for the PylRS evolution described above – and produced in *E. coli* NEBExpress(lowRF1), then purified to homogeneity via the *Strep*-tag II. When Azurin or mScarlet were subsequently loaded onto a 1 mL α-CD affinity column, each protein accumulated at the top of the resin. Even after washing with chromatography buffer A (25 mM Tris/Cl pH 8.0, 150 mM NaCl), a large fraction of the POI was retained there (Fig. 4B). Thus, Pap in its low-energy *trans*-state was able to interact with the immobilized α-CD groups and form a stable non-covalent complex even if incorporated into a recombinant protein. On the other hand, upon exposure to UV light at 355 nm both POIs eluted immediately with the buffer flow (see below). Therefore, as with the free ncAA initially investigated, the interaction between immobilized α-CD groups and a POI exposing the Pap side chain can be reversed by its light-induced isomerization into the *cis*-configuration (see Fig. 1A).

The effect of UV light exposure on the binding of the POI was further demonstrated by quantifying the mScarlet fluorescence of individual fractions collected in the course of the α-CD affinity chromatography (Fig. 4C). As long as exposed to daylight – which stabilizes the *trans*-configuration of the azo-group as explained above, even though with a proportion of 20–25 % in the *cis*-state – mScarlet-*Strep*-GG-Pap was largely retained on the column, with only minor fluorescence detectable in the flow-through upon continued washing. Only after specific illumination at 355 nm (via lateral exposure to LED light in the shaded laboratory, cf. Supplementary Fig. S1D) the bound protein quantitatively eluted. For comparison, exposure to the UV light already during the loading and washing steps led to the immediate elution of mScarlet-*Strep*-GG-Pap in the flow-through. More detailed investigation of the role of illumination, at 430 nm or 355 nm, respectively, during sample loading/washing steps and the elution phase revealed that the proportion of bound Azo-tagged POI can actually be boosted by constant exposure to blue light (Supplementary Fig. S14) owing to the dynamic equilibrium between *cis*- and *trans*-isomers in the PSS (see Fig. 1B). However, this High Yield mode also leads to less tight retention on the α-CD affinity column such that in practice (see below) the High Purity mode, with sample loading and washing in the dark (where the photostationary *cis*-/*trans*-isomer composition is frozen), may be beneficial.

To further demonstrate a chromatographic purification effect mediated by the Azo-tag, a mixture of the separately prepared mScarlet-*Strep* protein, this time lacking the C-terminal Pap residue, with the Azurin-*Strep*-GG-Pap from above was applied to the α-CD affinity column (Fig. 4D and Supplementary Movie 1). While mScarlet-*Strep* entered the column and eluted with the buffer flow without delay or visibly interacting with the resin – thus also confirming the negligible role of the *Strep*-tag II – the Azo-tagged Azurin accumulated in the top zone of the matrix as before. Upon continued washing with chromatography buffer (under dim light), the two colored proteins gradually separated and the column was completely cleared from mScarlet (after a total washing volume of 1.5 mL), whereas Azurin was fully retained in the upper part of the column. Then, 355 nm UV light was applied and Azurin eluted instantaneously with the chromatography buffer flow. This experiment demonstrated that the Azo-tag as part of a biosynthetic POI allows its efficient separation from an untagged recombinant protein via affinity chromatography on an α-CD matrix under physiological buffer conditions, simply controlled by

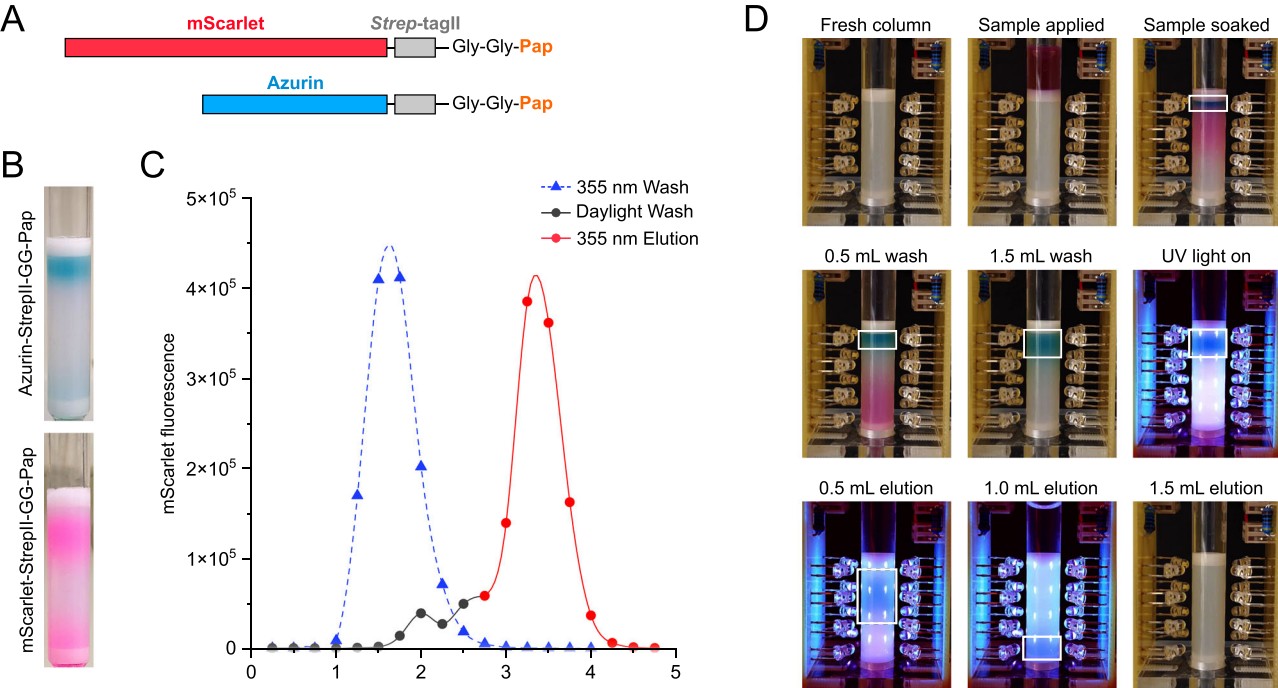

**Fig. 4 | α-CD affinity chromatography of Azo-tagged proteins. A** Scheme of the recombinant (mature) protein constructs for Azurin and mScarlet, both carrying a C-terminal Azo-tag with the Pap residue. **B** Retention of (pre-purified) Azurin-GG-Pap (top) and mScarlet-GG-Pap (bottom) to an α-CD affinity column (1 mL bed volume) after washing with 1 mL buffer. The minor fraction of protein that carried the Azo-tag in the *cis*-state (see Supplementary Fig. S14) or that had not fully incorporated the C-terminal Pap residue via amber suppression (see Supplementary Fig. S12) is already partially washed out of the column with the buffer flow. **C** The elution of mScarlet-*Strep*-GG-Pap under different illumination conditions

monitored via its specific fluorescence in the chromatography fractions. The column was either loaded with the protein and (i) washed directly under 355 nm UV light (blue dashed line) or (ii) washed under daylight (black solid line), followed by elution upon exposure to 355 nm UV light (red solid line). **D** Separation of a mixture of Azurin-*Strep*-GG-Pap and mScarlet-*Strep*, here without the Azo-tag, on the α-CD affinity column. While mScarlet (magenta) does not bind to the column and is quickly washed out, Azurin-*Strep*-GG-Pap (blue) is retained in the upper zone of the column and specifically eluted afterwards via exposure to UV light (see the Supplementary Movie 1).

light. The elution itself was typically complete within 1–1.5 column volumes, which is comparable to the conventional application of a competitive ligand, e.g. when using a biotin derivative in case of *Strep*-tag affinity purification[8] and, thus, keeps dilution of the finally purified protein sample at minimum.

## Biosynthesis of exemplary proteins and their one-step affinity purification from cell extracts

Aiming at practical applications in the life sciences, we investigated if the Azo-tag in combination with the α-CD affinity chromatography is suitable for the one-step purification of a recombinant POI from a complex host cell protein mixture. As the Azurin-*Strep*-GG-Pap from above constituted the vastly prevailing protein in the periplasmic protein fraction of the corresponding transformed *E. coli* strain its isolation from this cellular extract was easy to accomplish (Supplementary Fig. S12). Hence, we chose human cystatin C[43] as a more challenging and biomedically relevant recombinant POI, which also carries disulfide bonds. Again, this protein was fused with the C-terminal *Strep*-tag II followed by the GG-Pap sequence, in addition to the N-terminal OmpA signal peptide for periplasmic secretion.

The periplasmic extract of *E. coli* cells expressing this biosynthetic protein was prepared by osmotic shock and directly applied to the α-CD affinity column. After that, the column was washed with 3 column volumes of buffer under daylight until the host cell proteins were removed. When then applying 355 nm UV light, the cystatin C specifically eluted from the column as detected by SDS-PAGE analysis of the corresponding fractions (Fig. 5). Notably, this analysis revealed that essentially all host cell proteins were quickly washed out of the column using a physiological buffer (25 mM Tris/Cl pH 8.0, 150 mM NaCl). On

the other hand, the cystatin C was obtained in high purity – with just one minor contamination. This demonstrated the successful light-controlled purification of a POI via the Azo-tag from the periplasmic bacterial extract, also indicating high specificity of the immobilized α-CD groups towards the Pap side chain.

The only contaminating protein species that was detectable as a very weak band migrating at around 40 kDa in the SDS-PAGE was subsequently identified as the maltose binding protein (MBP) of *E. coli* using tandem ESI-MS of peptide fragments prepared by trypsin digestion (see Supplementary Fig. S18). This was also in agreement with its measured intact mass of 40706 Da. The same contaminating band, whose (retarded) elution was independent of light, was detected in experiments with other POIs. While this phenomenon was unexpected, since MBP is not a commonly known host cell contaminant in the widely applied affinity purification of MBP fusion proteins from *E. coli* extracts on an amylose resin[44], it is supported by previous investigations on the ligand spectrum of MBP which includes cyclic maltodextrins[45]. However, adsorption of the endogenous MBP to the α-CD affinity column was effectively prevented here when supplementing the bacterial extract with 5 mM maltose, thus saturating its ligand pocket ($K_D = 3.5\,\mu M$). As an alternative, we constructed a derivative of the expression strain using the λ-Red system as above, NEBExpress(lowRF1/ΔMBP), which had the *mal*E gene deleted (Fig. 3C). While this strain showed slightly slower growth, it reached similar cell densities in the stationary phase and appeared as another suitable expression host for POIs carrying an Azo-tag, as demonstrated here for sfGFP (Supplementary Fig. S18E, F).

In the next step, we set out to purify Azurin-*Strep*-GG-Pap – still secreted into the bacterial periplasm utilizing the N-terminal OmpA

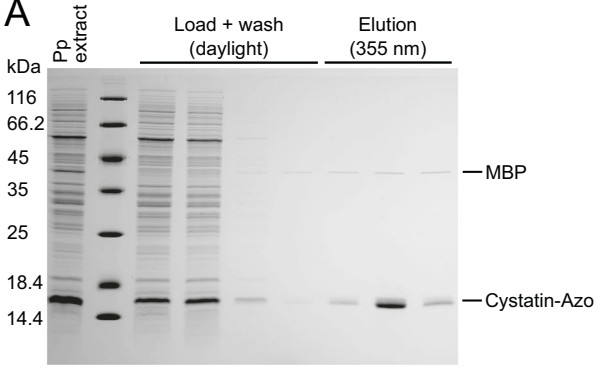

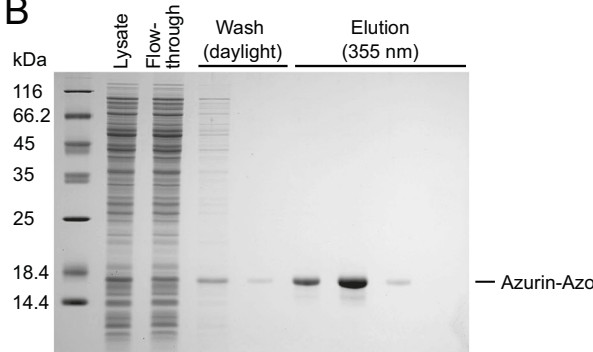

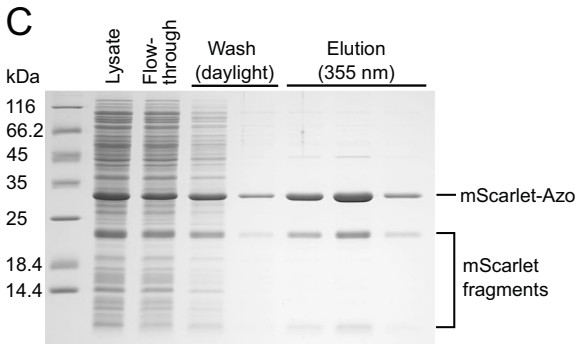

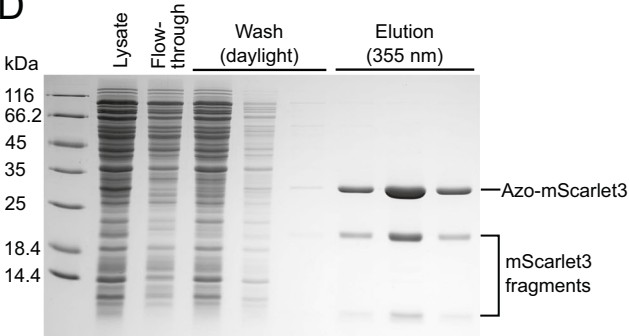

**Fig. 5 | One-step purification of different Azo-tagged POIs from complex protein mixtures via light-controlled α-CD affinity chromatography.** Protein solutions were applied to a column with 1 mL bed volume under daylight and, after washing with buffer, the bound protein was eluted via illumination at 355 nm (in the dark lab). **A** Isolation of cystatin C carrying the C-terminal GG-Pap sequence from the periplasmic cell extract of NEBExpress(lowRF1). **B** Isolation of Azurin carrying the C-terminal GG-Pap sequence from the total cell extract of NEBExpress(lowRF1), with 5 mM maltose added to the lysis buffer. **C** Isolation of mScarlet carrying the C-terminal GG-Pap sequence from the total cell extract under similar conditions. **D** Isolation of mScarlet3 carrying the N-terminal Ala-Pap-Gly sequence from the total cell extract, again under the same conditions. Note that the two additional lower bands in the elution fractions are obviously due to a specific backbone cleavage within mScarlet3 (cf. Supplementary Fig. S17A). This prominent post-maturation hydrolysis, which has also been described by others[68], was observed in all mScarlet(3) preparations, even when using other purification methods and in the absence of the Azo-tag, but not for sfGFP (cf. Supplementary Fig. S18E).

signal sequence as before – from the total cell lysate of *E. coli* (Fig. 5B). In the SDS-PAGE of this cell-free extract, Azurin no longer gave rise to a prominent protein band; nevertheless, upon elution under illumination at 355 nm it was recovered from the α-CD affinity column as a homogeneous protein without detectable impurities. Notably, analogous purification experiments performed with a version of this POI lacking the *Strep*-tag II, i.e. carrying the Azo-tag alone, led to similar results, thus excluding any contribution from the second affinity tag (Supplementary Fig. S13). Likewise, mScarlet equipped with the C-terminal *Strep*-GG-Pap tag, which was expressed in the bacterial cytoplasm, was isolated from the whole cell extract using the same procedure by light-controlled α-CD affinity chromatography in a highly pure state (Fig. 5C).

**Sequence optimization of the Azo-tag and α-CD affinity column performance**

While the experiments up to now clearly demonstrated the efficient purification of POIs carrying a C-terminal Azo-tag on an α-CD affinity column, even from a whole cell extract, there was always a proportion of the recombinant protein that did not adsorb to the affinity column (cf. Supplementary Fig. S12 and Fig. 5A, C) and appeared in the flow-through. As in these cases washing was performed under daylight, the premature elution of some proportion of the Azo-tagged POI in the *cis*-state, as mentioned further above, should have been largely prevented (see the effect of continuous illumination during the washing step as illustrated in Supplementary Fig. S14). Moreover, determination of the dynamic column capacity (Supplementary Fig. S15) did not provide an indication of column overloading, with a binding capacity of >10 mg

Azo-tagged POI per 1 mL bed volume. Likewise, the α-CD affinity matrix appeared highly robust and tolerant against widely applied biochemical reagents (such as 10 mM DTT or TCEP, 6 M GdnHCl, 8 M urea or 1 M NaOH), allowing 10–50 repeated chromatography runs without noticeable loss in performance (Supplementary Fig. S15).

However, UV/Vis absorption spectra as well as ESI-MS analyses indicated that the observed effect was due to a fraction of the biosynthetic protein that lacked the C-terminal Pap amino acid (Supplementary Fig. S12), whereas only the protein that had Pap incorporated remained bound to the α-CD matrix and was eluted upon UV exposure. Obviously, this phenomenon was caused by the incomplete suppression of the amber stop codon during the cotranslational incorporation of the ncAA using the engineered PylRS/tRNA^UAG system, due to the merely partial deactivation of RF-1 in the engineered *E. coli* host strain used, NEBExpress(lowRF1), as described above. This led to the translation of a protein with essentially identical biochemical properties but just one residue shorter, thus lacking the azobenzene side chain needed for the complex formation with the immobilized α-CD groups. While this unbound side product may less matter in practice, we sought to avoid its formation for demonstration purposes.

To ensure that only the full-length gene product carrying the Pap residue is expressed, the amber stop codon was moved from the C- to the N-terminus of the POI. Notably, in this setting care had to be taken that the azobenzene side chain was not obstructed by a bulky neighboring amino acid such as, in particular, the start Met residue in the case of direct cytoplasmic expression (e.g. for mScarlet). To promote adequate accessibility of the N-terminal Pap group, it was preceded by

an Ala residue, right after the start Met residue, based on the knowledge that the endogenous methionine aminopeptidase cleaves the start Met if followed by a residue with small side chain[46]. On the other hand, with regard to the secretory expression strategy, it was questionable if the signal sequence would get efficiently processed by the bacterial signal peptidase I[47] if directly followed by a bulky hydrophobic side chain such as the one of Pap. Therefore, an Ala residue was also inserted on the N-terminal side of Pap for those genetic constructs effecting periplasmic secretion (e.g. for Azurin). In both cases, Pap was followed by two Gly residues, with the POI sequence arranged downstream and still carrying the *Strep*-tag II at its C-terminus (for convenience). Using these constructs, the N-terminally Azo-tagged versions of both Azurin and mScarlet3 were both successfully purified from the whole cell extract of *E. coli* without detectable recombinant protein in the flow-through or wash fractions (Fig. 5D and Supplementary Fig. S18A).

Nevertheless, it appeared that the two N-terminally tagged POIs did not adsorb as tightly to the α-CD affinity column as their versions with the C-terminal Azo-tag. Hence, we investigated if the amino acids directly adjacent to Pap have an impact on its affinity to α-CD. In order to screen a large set of sequence combinations, a SPOT assay[48] was performed. To this end, an array of 18×18 Xaa-Pap-Yaa tripeptides, C-terminally anchored on a hydrophilic carrier membrane, was synthesized, with Xaa and Yaa representing all amino acids except for the bulky Trp and the chemically labile Cys. Binding activity was probed by incubation with bacterial alkaline phosphatase[27] which had been chemically conjugated with α-CD groups, followed by washing with buffer and detection of enzyme activity using a precipitating chromogenic substrate (Supplementary Fig. S19). In this assay, the SPOT signal obtained for the original tripeptide sequence, Ala-Pap-Gly, was only slightly above the median. However, among others, there was in particular one tripeptide that gave rise to a significantly elevated signal: $H_2N$-Gly-Pap-Gly-. In fact, the emergence of an N-terminal Gly residue, which lacks any side chain, appeared plausible in the light of further reduced sterical requirements in the chemical neighborhood of the Pap side chain as contemplated above.

To test the performance of the different N-terminal peptide sequences in the α-CD affinity purification, the corresponding versions of mScarlet3 were generated as recombinant proteins and equivalent amounts of the bacterial total cell lysate were loaded onto the column. As expected, moderate but consistent differences in binding of the colored Azo-tagged POI to the α-CD affinity resin were observed during the washing steps (see Supplementary Fig. S20). After washing with three bed volumes of chromatography buffer, the Gly-Pap-Gly variant still remained concentrated within the upper half of the column, whereas the initial Ala-Pap-Gly variant was already mobilized to some extent, visibly moving towards the bottom of the column. Apart from that, the purity of both protein preparations obtained after UV-induced elution was comparable.

Finally, we investigated if the exposure to UV light at 355 nm had an effect on the integrity of the Azo-tagged POI purified via α-CD affinity chromatography. To this end, we used (i) an ordinary protein lacking chromophores, human cystatin C (which only absorbs light in the farther UV region, around 280 nm, due to the Tyr and Trp side chains), and (ii) a colored/fluorescent protein that also shows absorption in the relevant spectral region, sfGFP (both POIs already employed above). Remarkably, even after extended illumination up to 60 min using the same setup of LEDs as applied for the affinity chromatography, no signs of deterioration apparent from SDS-PAGE, ESI-MS or the fluorescence spectrum were detectable (Supplementary Fig. S16). This is in line with the mild form of near UV radiation at 355 nm and the well known resistance of proteins against exposure even to 280 nm UV light, as it generally happens in common UV detectors used for chromatographic purification.

## Application of the Azo-tag in a high-throughput setup and to antibodies or their fragments

The experiments so far demonstrate that the light-controlled α-CD affinity chromatography is suitable for Azo-tagged protein purification at the bench top scale. While upscaling to bulk separation or to industrial application may require more elaborate column design and process development, Excitography directly appears attractive for screening applications at the microscale, compatible with laboratory automation. To this end, we investigated the isolation of proteins using 96-well microtiter plates. POIs belonging to two classes with biomedical relevance were chosen as examples: (i) enzymes, such as the monomeric β-lactamase AmpC[49] (Fig. 6A, B) as well as the homodimeric glutathione-*S*-transferase[50], GST (Supplementary Fig. S23A, B), and (ii) antibody fragments, i.e. the single-chain variable fragment (scFv) of the monoclonal antibody (MAb) T84.66 directed against the tumor-associated carcinoembryonic antigen (CEA)[51] (Fig. 6C) as well as the nanobodies (NBs) EgA1 directed against human epidermal growth factor receptor (EGFR)[52] (Supplementary Fig. 22) and 2Rs15d directed against human epidermal growth factor receptor 2 (HER2)[53] (Supplementary Fig. S23C). These representative POIs can be produced in small cultures and handled in functional assays using standard laboratory robotics, such as in the frame of biopharmaceutical protein engineering or industrial enzyme screening campaigns.

Here we used lysates of *E. coli* cultures expressing each of these POIs equipped either at the N-terminus or at the C-terminus with the Azo-tag (see Supplementary Table S1) and encoded on the pSB19-PapRS#34 vector for recombinant gene expression, followed by purification on the α-CD affinity matrix, this time hosted in the wells of a 96 well column plate (200 µl bed size). After the application of a 500 µl bacterial extract and washing under gravity flow, until the host cell proteins had been removed, the columns were illuminated from above at 355 nm by an array of UV LEDs (one per well; see Supplementary Fig. S1H), which led to the immediate elution of the bound POI simply in the applied buffer (chromatography buffer A or GST assay buffer). The collected protein eluate showed high purity and uniform protein concentration, with no significant well-to-well difference (Fig. 6A and Supplementary Fig. S23A). The resulting purified protein solutions were directly suitable for subsequent assays, without the need for buffer exchange or removal of elution agents, as here demonstrated with enzyme activity assays and antigen binding assays, respectively (Fig. 6B, C and Supplementary Fig. S23B, C).

In this context we also investigated if the presence of the Azo-tag may have an influence on the structural or functional properties of the POI. To this end, we prepared GST both with and without a C-terminal Azo-tag (GST-*Strep*-GG-Pap *versus* GST-*Strep*-GG, respectively) and performed various assays (Supplementary Fig. S21). While the presence of the azobenzene side chain (in the *trans*-state) was clearly visible as a distinct band at ~326 nm in the UV-Vis spectrum, there was no change in the protein secondary structure, as assessed from far-UV circular dichroism (CD) spectroscopy, and the enzyme activity of both protein versions was identical within experimental error, as measured by Michaelis-Menten kinetics. Most notably, the elution behavior in an analytical hydrophobic interaction chromatography was indistinguishable, which indicates that the observed poor solubility of the isolated Pap amino acid does not cause a measurable increase in overall hydrophobicity if incorporated into a folded protein. Likewise, a comparison of the anti-EGFR Nb with or without the C-terminal Pap residue revealed an unchanged high antigen-binding activity (Supplementary Fig. S22). Moreover, the azobenzene side chain proved to be stable against common (intracellular) biochemical reducing agents such as glutathione (GSH), NADPH and TCEP (Supplementary Fig. S13).

Finally, to further extend the utility of the light-controlled α-CD affinity chromatography also to full-size antibodies produced in mammalian cell culture, we designed a small adapter molecule based

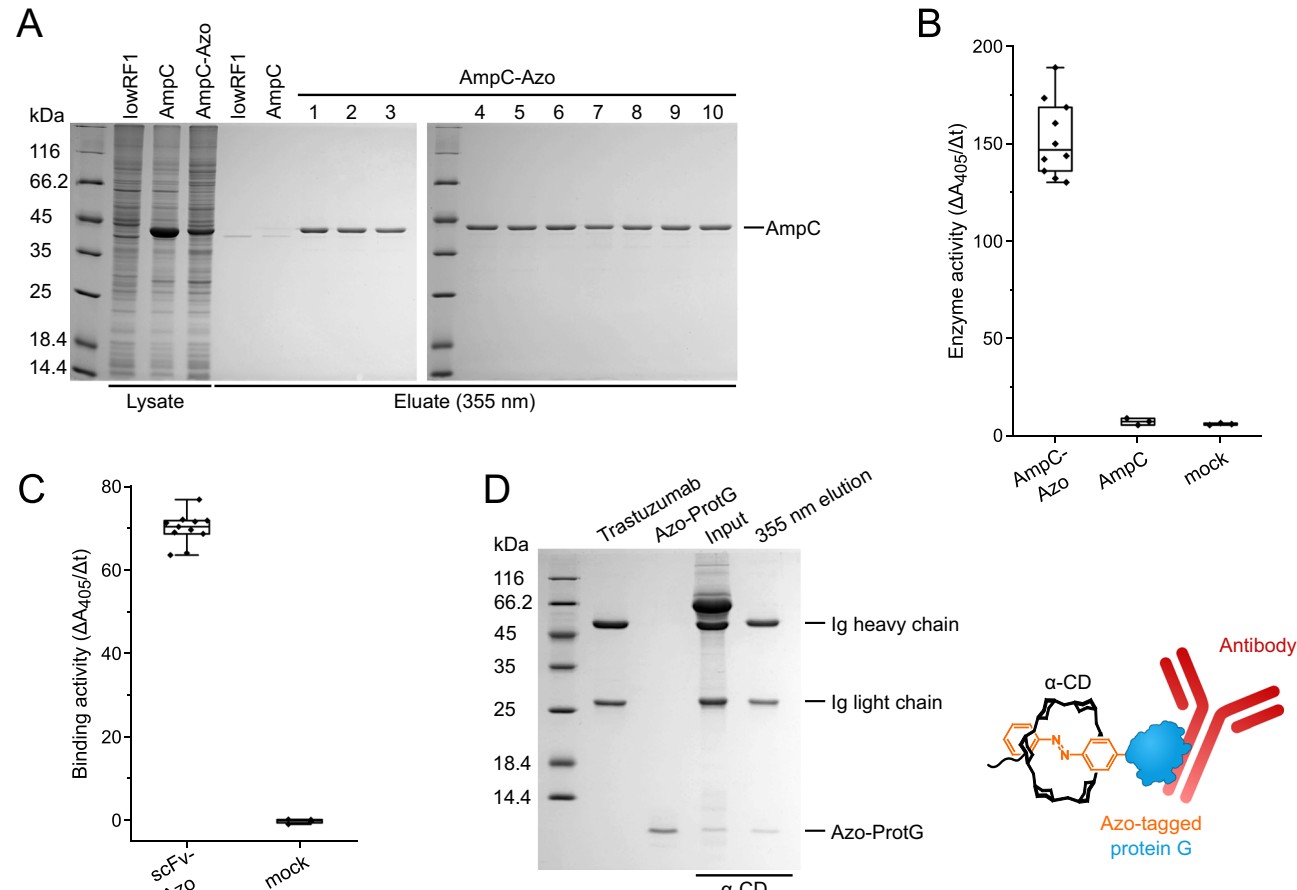

**Fig. 6 | One-step purification of Azo-tagged POIs in parallel using the 96-well format. A** Isolation of the β-lactamase AmpC carrying the C-terminal GG-Pap sequence from the total cell extract of NEBExpress(lowRF1) using 10 wells of a 96-well receiver plate filled with 200 μl α-CD affinity resin each. AmpC was purified as essentially homogeneous protein without significant well-to-well difference. AmpC without an Azo-tag or untransformed NEBExpress(lowRF1) cells served as negative controls. **B** Colorimetric assay to measure the AmpC enzyme activity in aliquots of the eluate from (**A**) after mixing with CENTA substrate in a 96-well microtest plate. **C** Affinity purification of the anti-CEA scFv antibody fragment T84.66 carrying the C-terminal GG-Pap sequence from the periplasmic extract of NEBExpress(lowRF1), followed by ELISA to assess antigen-binding activity in 10 separately eluted aliquots.

scFv bound to a CEA-coated microtiter plate was detected via an antibody-HRP conjugate directed against the *Strep*-tag II using ABTS substrate. Untransformed NEBExpress(lowRF1) cells served as negative controls (3 wells). Data points in box plots shown in (**B**) and (**C**) each correspond to a single measurement per well; median (center line), upper/lower quartiles (box limits) and min/max values (whiskers) are indicated. **D** Light-controlled affinity purification of trastuzumab from cell culture medium using an adapter protein, as schematically illustrated to the right. Trastuzumab and the Azo-tagged protein G C2 fragment (Azo-ProtG) were mixed in DMEM, 5 % (v/v) FCS (input). The antibody was specifically isolated by UV light-controlled elution while all serum proteins, in particular albumin (~66 kDa), were efficiently removed.

on the C2 fragment of protein G which forms complexes with the Fc portion of a wide range of immunoglobulins (Igs)[54]. This adapter molecule (Azo-ProtG), equipped with an N-terminal Azo-tag (see Supplementary Table S1) and a C-terminal *Strep*-tag II, was expressed with high yield in NEBExpress(lowRF1), purified and added to cell culture medium containing the anti-HER2 IgG1/κ humanized antibody trastuzumab (Herceptin®)[55] (Fig. 6D). Trastuzumab was isolated from this solution via Excitography in one step in the presence of phosphate-buffered saline (PBS) at physiological pH. Notably, albumin, which is a major protein constituent in many cell culture media, was quickly washed out of the α-CD affinity column, and thus efficiently removed, prior to the selective UV light-induced elution of the intact antibody.

## Discussion

We present a generic affinity chromatography for recombinant proteins that allows elution in a physiological buffer solely under the control of light, based on the considerable change in structure and, consequently, affinity between the azobenzene moiety of Pap and an α-CD matrix. Apart from presenting a genetic system for the efficient

incorporation of Pap into diverse proteins, including enzymes and antibody fragments, we have demonstrated their light-controlled purification – termed Excitography – under native buffer conditions in a functional state and, optionally, in a high-throughput fashion. Use of a tiny Azo-tagged adapter molecule even allowed the quick isolation of a clinically approved antibody from cell culture medium. The Azo-tag enables the one-step purification of POIs from complex biological samples or extracts to homogeneity in a quality comparable to established systems such as the His6-tag or the *Strep*-tag II, or even better.

One explanation for the high separation efficiency is the absence of background binding activity of the α-CD matrix towards most bacterial host cell proteins (or common metabolic cofactors) or cell culture proteins, which are quickly washed out of the column under mild buffer flow. A β-CD affinity column was previously used for the two-step purification, after anion-exchange chromatography, of pullulanase from the cell-free extract of a hyperthermophilic bacterium[28]. However, α-CD groups with their more narrow hydrophobic cavity likely exhibit higher selectivity. The only contaminant detectable in *E. coli* cell extracts was the MBP, which could be abolished by using a

deletion strain or by supplementing the cell extract with maltose, its physiological sugar ligand.

Another aspect is the effective elution of all Azo-tagged POIs investigated here upon irradiation at 355 nm via *trans/cis*-isomerization of the Pap side chain. From data available in the literature is was not expected that exposure to LED UV light at 355 nm specifically induces the almost quantitative formation of the *cis*-state of Pap (~95 % as measured by ${}^{1}$H-NMR spectroscopy) and that this isomer virtually shows no measurable affinity towards the α-CD group. Together, both effects enable the fast and selective elution of an Azo-tagged POI from the affinity matrix upon illumination at the chosen wavelength in the near UV region. In this context, sterical accessibility governed by the linker chemistry as well as the surface density of the α-CD groups on the stationary phase particles offer obvious parameters for the further optimization of Excitography.

Attempts to engage certain fusion proteins which undergo light-dependent conformational change for purposes of biomolecular separation at analytical scale have been described before. For example, the mem-iLID strategy[56] made use of a two-component system involving the light-oxygen-voltage-sensing domain, LOV2, of phototropin 1 from *Avena sativa* to selectively adsorb a fusion protein in the presence of blue light to a plasma membrane of *Xenopus laevis* oocytes or of *E. coli* with the anchored binding partner, followed by its slow release in the dark. Similarly, utilizing the light-dependent association between phytochrome B (PhyB) and its phytochrome-interacting factor 6 (PIF6) from *Arabidopsis thaliana*, interaction partners of the tyrosine kinase ZAP70, applied as a fusion protein, from ZAP70-deficient Jurkat T cells were enriched via selective binding to agarose beads functionalized with PhyB under illumination and, after light-induced desorption (at a different wavelength), analyzed by quantitative mass spectrometry[57]. However, both approaches employed photoresponsive plant proteins associated with chromophores (flavin mononucleotide and phycocyanobilin, respectively). In contrast, our system is based on the very short Azo-tag peptide comprising just 2–4 residues, which is cotranslationally incorporated into the POI, and its light-dependent interaction with the robust α-CD Sepharose matrix (without an immobilized protein as binding partner). Due to its simplicity, this approach offers unique advantages and enables efficient chromatographic purification, as demonstrated here for more than 12 different representative POIs (Supplementary Table S1).

Implementation of the Azo-tag into a POI is easy to accomplish by gene cloning (Supplementary Fig. S25) and involves just moderate alterations to the protein primary structure, with the photo-switchable Pap residue itself and one or two small Gly (or Pro) residues as spacer. This added peptide segment is shorter than all other recombinant affinity tags described to date. According to our analyses, the azo-benzene side chain is chemically stable and a putative increase in hydrophobicity is not detectable for the Azo-tagged POI, which seems plausible when comparing a single Pap residue with both an exposed hydrophobic Phe and Trp side chain in the case of the *Strep*-tag (II), for example. Nevertheless, the optimal choice of N-terminal *versus* C-terminal positioning of the Azo-tag should be tested empirically for each POI as such regional effects are also known for other established affinity tags for protein purification.

In case of mScarlet and Azurin, α-CD affinity chromatography appeared similar with both the N-terminal and the C-terminal Azo-tag, the latter showing just slightly better performance. On the other hand, the light-induced purification of the N-terminally tagged sfGFP was comparable to other POIs whereas retention on the α-CD affinity column with a C-terminal Azo-tag was strongly reduced. Interestingly, for the monomeric enzyme AmpC the opposite was the case. Apart from that, a direct comparison of the performance of Azo-tagged Azurin and mScarlet in Excitography demonstrated less tight retention of the latter, independent of the arrangement of the Azo-tag, thus indicating

some context-dependence of the complex stability between the *trans*-azobenzene side chain and the α-CD groups and/or influence of sterical effects. Nevertheless, all other POIs tested in our hands (see Supplementary Table S1) showed satisfactory retention on the α-CD affinity column, comparable to the one of Azurin, also irrespective of differences in protein size.

Apart from slightly varying performance in the α-CD affinity chromatography, the Azo-tag might potentially interfere with the biochemical activity of a POI. Usually, this can be avoided by appropriate molecular design, such as with the choice of the C-terminus (which is opposite to the antigen-binding site) in the case of the Ig fragments investigated here. Alternatively, a protease recognition site may be inserted between the Azo-tag and the POI in order to liberate the unmodified protein via site-specific proteolysis for further study. Based on the results of the SPOT Pap-peptide screen and the circumstance that the Azo-tag, in principle, works both at the N- and the C-terminus of a POI, one can even envisage applications where the Pap side chain may be exposed somewhere else on the protein surface, or as part of a linker in a fusion protein, provided that there is sufficient sterical accessibility.

Beyond our present proof of concept that the photo-induced configurational isomerization of the ncAA Pap enables chromatographic separation of an Azo-tagged POI under kinetic as well as quantitative aspects, its simple technical implementation using a cheap LED device should pave the way to even smarter applications – such as an affinity matrix exerting light-switchable binding properties by itself. As demonstrated here, practical applications for Excitography reach from the facile one-step protein purification for research purposes at the bench-top scale to automated high-throughput screening campaigns in microtiter plates and may even extend to industrial scale if using column setups that allow internal illumination, for example via light conductors, from within the chromatography bed.

## Methods
### Spectroscopic measurements

Home-made devices for the photo-induced *trans/cis*-isomerization of Pap are depicted in Supplementary Fig. S1. Absorption spectra of Pap were recorded on a S-3100 flash photometer (Scinco, Seoul, Korea) operated with LabPro Plus software (Build 410.1) and thermostated at 25 °C. The raw spectra (from 50 scans) were smoothed with 5th order polynomials where applicable and further evaluated with OriginPro 2021b software (OriginLab, Northampton, MA). Absorption spectra of proteins were measured with an Ultrospec 2100 pro photometer (Amersham Biosciences, Amersham, UK) with Datrys Life Science software 2.4 (GE Healthcare, Uppsala, Sweden). Ligand titration and Pap *cis/trans*-isomerization experiments were performed under a tiny light source that fits on top of a 4 mL (1×1 cm²) quartz cuvette (VWR/Avantor, Ismaning, Germany) equipped with two LEDs for different wavelengths that could be individually operated: 355 nm (XSL-355-5E-R6, 5 mm diameter, 10–20 nm spectrum half width, 1.2–2.4 mW, 15° viewing angle; Roithner LaserTechnik, Vienna, Austria) and 430 nm (LED430-01, 5 mm diameter, 16 nm half-width, 27 mW, 16° viewing angle; Roithner LaserTechnik). For irradiation of the α-CD affinity matrix (1 mL bed volume), packed into a Pierce disposable 2 mL column (Thermo Fisher Scientific, St. Leon-Rot, Germany), at 355 nm an illumination device carrying 8 or 16 LEDs (XSL-355-3E-R6, 3 mm diameter, 1.2–2.4 mW, 30° viewing angle; Roithner LaserTechnik), each placed at a distance approximately 5 mm to the column housing, was used. Sample illumination for the ${}^{1}$H-NMR measurements was performed with LEDs having peak wavelengths of 355, 365, 400, 410, 420, 430, 440, 450 or 465 nm (XSL-355-3E-R6, XSL-365-3E, LLS-UV400, VL410-5-15, LED420-33, LED430-33, VL440-5-15, LED450-01, RLS-B465, respectively; all from Roithner LaserTechnik). The spectra of LEDs – and also of environmental light sources – were checked using a handheld spectrometer (OHSP-350UV; Hangzhou Hopoo Light&Color

Technology, Zhejiang, China). In the course of such measurements it was found that the LED type LED430-33 showed a rather broad distribution of peak wavelengths, which allowed us to separate them into two groups with maxima around 430 nm and around 435 nm, respectively. Light-sensitive operations were generally performed in a shaded laboratory or in the dark room under monochromatic red LED light (642 nm peak wavelength, 57 nm halfwidth), if necessary. Fluorescence spectra of proteins were recorded on a FluoroMax 3 spectrofluorimeter (Horiba Jobin Yvon, Oberursel, Germany) with Fluor-Essence 3.5 software (Jobin Yvon).

**Titration of Pap with CD compounds**

A 4 mL (1×1 cm²) quartz cuvette was filled with 1980 or 1960 μL 100 mM Tris/HCl pH 8.0, and a blank spectrum was recorded. 20 or 40 μL, respectively, of a 5 mM Pap solution (in the predominant *trans*-state, after equilibration under daylight) in 10 mM NaOH was added and thoroughly mixed by pipetting. After 5 min equilibration, the first spectrum was collected, followed by the successive addition of appropriate volumes (from 1.0 to 20 μL) of α-CD or β-CD (50 mM in 100 mM Tris/HCl pH 8.0). Mixing was performed by pipetting or by using a small magnetic stirring bar, and the absorption spectrum was each time measured after a 30 s incubation period. For measurements with Pap in the *cis*-configuration, the 2 mL sample in the quartz cuvette was initially irradiated at 355 nm for 30 min from the top. Then, the first spectrum was collected immediately, followed by further measurements after the addition of the α-CD or β-CD aliquots as above. The degree of complex formation was monitored via absorbance measurement at 326 or 426 nm, respectively, utilizing the change in absorption amplitude for the Pap side chain when bound to the CD (see Supplementary Fig. S3C and D). Of note, complex formation between *trans*-Pap and both α-CD and β-CD leads to a red shift of the band for the π→π* transition around 326 nm, in line with measurements for other previously investigated azobenzene derivatives and indicating a less polar environment of the azo-dye in the complex[58]. This was accompanied by a small decreasing amplitude in the case of α-CD but a slightly increasing amplitude for β-CD.

In a typical titration experiment, 200 μL CD solution was added in total, leading to a final molar ratio Pap:CD of 1:100 and a dilution of the initial Pap solution by 10 %. For the titration curves shown in Fig. 2I, K and L, the absorbance values were fitted by non-linear regression using KaleidaGraph software (Synergy, Reading. PA) and the following equation based on the Law of Mass Action for bimolecular complex formation[59], explicitly accounting for the dilution effect:

vector pSB19 (see Fig. 3A) was assembled, using the one-step SLIC method[60], from five PCR fragments generated from suitable plasmid templates with 15–25 bp mutual overlaps, thus introducing the following changes compared with the preceding pSB15 vector: (i) replacement of the constitutive *pro*S promoter for the engineered PylRS gene by the inducible *ara*BAD promoter, including the *ara*C repressor gene, (ii) replacement of the *tet*^p/o for the POI-encoding region, here sfGFPa39, by the *lac*UV5^p/o, including the *lac*I repressor gene, and (iii) removal of the *tet*R repressor cistron downstream of the *bla* (*amp*^r) gene and replacement by the coding region for chloramphenicol acetyltransferase (*cam*^r) further harboring an amber stop codon at amino acid position 112. The purified PCR fragments were mixed in approximately equimolar amounts together with T4 DNA polymerase (Thermo Fisher Scientific) in the buffer supplied by the manufacturer. After incubation at room temperature for 3 min, the mix was directly used to transform chemically competent *E. coli* XL1-Blue[61].

To introduce the C-terminal Azo-tag into the coding region for the sfGFP reporter protein, two Gly codons and the amber stop codon were inserted between the C-terminal *Strep*-tag II and the final ochre stop codon, thus placing Pap as the last encoded residue in the reading frame, and this gene cassette was subcloned on pSB19 between the *Xba*I and *Hin*dIII restriction sites. An *Afe*I restriction site directly upstream of the *Strep*-tag II, together with the unique *Xba*I restriction site in front of the whole reading frame, was subsequently used to exchange the entire coding region for sfGFP by the ones for different other POIs (see Supplementary Table S1). For example, the coding region for mScarlet[41] was obtained as a synthetic gene whereas the one for Azurin (including the OmpA signal sequence for periplasmic secretion) was amplified by PCR from a published plasmid[7], followed by cloning on pSB19-PapRS#34.

Alternatively, Azurin was subcloned on the same vector with the C-terminal *Strep*-tag II without amber stop codon, and the Azo-tag was introduced at the N-terminus of the mature polypeptide, directly downstream of the OmpA signal sequence, via QuikChange site-directed mutagenesis (Agilent, Santa Clara, CA). Likewise, a synthetic gene for mScarlet3[42] carrying the N-terminal Azo-tag directly after the start Met residue (for cytoplasmic expression) was subcloned via the *Nde*I and *Afe*I restriction sites. mScarlet3 variants with different versions of the N-terminal Azo-tag were generated by PCR amplification with forward primers containing the corresponding mutations and cloning via the *Nde*I and *Hin*dIII restriction sites.

$$
\begin{aligned}
A(V_{CD}) \cdot (V_0 + V_{CD}) \\
= (V_0 \cdot C^0_{Pap} - V_{CD} \cdot C^0_{CD} - (V_0 + V_{CD}) \cdot K_D) \frac{\varepsilon_{Pap}}{2} + (V_0 \cdot C^0_{Pap} + V_{CD} \cdot C^0_{CD} + (V_0 + V_{CD}) \cdot K_D) \frac{\varepsilon_{Pap \cdot CD}}{2} \\
+ (\varepsilon_{Pap} - \varepsilon_{Pap \cdot CD}) \sqrt{(V_0 + V_{CD})^2 \cdot \left( \frac{V_0 \cdot C^0_{Pap} + V_{CD} \cdot C^0_{CD}}{2 \cdot (V_0 + V_{CD})} + \frac{K_D}{2} \right)^2 - V_0 \cdot V_{CD} \cdot C^0_{Pap} \cdot C^0_{CD}}
\end{aligned}
\tag{1}
$$

$A(V_{CD})$: Absorbance as a function of the added volume of the CD solution, $V_{CD}$; $V_0$: initial volume (2 mL) of the Pap solution with concentration $C^0_{Pap}$; $C^0_{CD}$: concentration of the CD solution; $\varepsilon_{Pap}$: absorption coefficient of Pap; $\varepsilon_{Pap \cdot CD}$: absorption coefficient of the Pap•CD complex; $K_D$: dissociation constant.

**Construction of expression plasmids for reporter proteins carrying the Azo-tag**

PCR was performed using the Q5 Hot Start High-Fidelity Polymerase (New England Biolabs) according to the manufacturer's protocol. The

**Protein expression and Pap incorporation**

*E. coli* NEBExpress(lowRF1), or NEBExpress(lowRF1/ΔMBP), was transformed with pSB19-PapRS#34 encoding the appropriate POI equipped with the Azo-tag (see Supplementary Table S1). Depending on the experiment, a culture of 100 mL or of 2 L Luria-Bertani (LB) medium supplemented with 100 μg/mL ampicillin (LB/Amp) was inoculated with 1/100 volume of an overnight culture and grown at 30 °C to OD₆₀₀ ≈ 0.4. At this point in time, 0.2 mM Pap (in case of the optimized genetic system), from a stock solution comprising 50 mM Pap and 200 mM β-CD in 0.2 M NaOH, as well as 0.2 % (w/v) L(+)-arabinose

(from a 20 % (w/v) stock solution in ddH$_2$O) were added. At OD$_{600}$ ≈ 0.8, gene expression for the POI was induced by further adding 0.5 mM IPTG (from a 0.5 M stock solution in ddH$_2$O) and continued for 16 h (cytoplasmic expression) or 3 h (periplasmic expression) at 30 °C.

### Preparation of bacterial cell extracts and Strep-Tactin affinity chromatography (SAC)

Preparation of periplasmic or cytoplasmic extracts as well as Strep-Tactin affinity chromatography were essentially performed as previously described[8,62]. Briefly, pellets from the bacterial culture were resuspended in periplasmic extraction buffer (100 mM Tris/Cl pH 8.0, 0.5 M sucrose, 1 mM EDTA), incubated for 30 min at 4 °C, and centrifuged repeatedly to remove the spheroplasts. The supernatant was recovered as the periplasmic cell fraction and dialyzed against SAC buffer (100 mM Tris/Cl pH 8.0, 150 mM NaCl, optionally with 1 mM EDTA). For cytoplasmic proteins, the bacterial pellet was resuspended in SAC buffer and the suspension was passed four times through a PANDA Plus 2000 homogenizer (GEA Niro Soavi, Lübeck, Germany). The total cell lysate was cleared by centrifugation and supplemented with 25 μL per liter culture × OD$_{600}$ of BioLock avidin (IBA Lifesciences, Göttingen, Germany) to mask D-biotin from the culture medium and/or endogenous biotinylated proteins from the host cell, followed by sterile-filtration. After application of the protein sample to the Strep-Tactin Superflow column (8 or 16 mL bed volume; IBA Lifesciences), the column was washed with SAC buffer until the baseline was reached, as monitored at 280 nm. The POI was eluted with SAC buffer supplemented with 2.5 mM D-desthiobiotin. In the case of Azurin, EDTA was omitted from the elution buffer. To reconstitute its blue-colored copper complex, the Azurin concentration in the eluate was determined from its absorbance at 280 nm, and 100 mM CuSO$_4$ was slowly added at a tenfold molar ratio.

### Protein analytics

SDS-PAGE was performed using a published buffer system[63] and staining with Coomassie brilliant blue. Protein concentrations were determined by measuring the absorbance at 280 nm, A$_{280}$, with calculated extinction coefficients as provided by the ProtParam server (https://www.expasy.org) according to the formula ε$_{280}$ = 1490 × n$_{Tyr}$ + 5500 × n$_{Trp}$ + 125 × n$_{Cystine}$ M$^{-1}$ cm$^{-1}$, plus a further increment of 7600 M$^{-1}$ cm$^{-1}$ to account for the contribution of the Pap side chain in this spectral region, if applicable.

Protein secondary structure was analyzed via circular dichroism (CD) spectroscopy on a J-1500 spectropolarimeter (Jasco, Groß-Umstadt, Germany) equipped with a quartz cuvette 106-QS (0.1 mm path length; Hellma, Müllheim, Germany) using Spectra Manager 2.15 (Jasco). Spectra were recorded at 20 °C from 190 to 250 nm by accumulating 16 runs (bandwidth 1 nm, scan speed 100 nm/min) using 16 μM protein solution in 0.1 M K-P$_i$ pH 6.5. After correction for the buffer blank, spectra were smoothed using the instrument software. The molar ellipticity was calculated according to the equation Θ$_M$ = Θ$_{obs}$ / (c·d), wherein Θ$_{obs}$ denotes the measured ellipticity, c the protein concentration [mol/L], and d the path length of the quartz cuvette [cm], and plotted against the wavelength.

For analytical hydrophobic interaction chromatography, a purified solution of GST-*Strep*-GG or GST-*Strep*-GG-Pap in 0.1 M K-P$_i$ pH 6.5 was adjusted to 2 % (v/v) acetonitrile, 0.1 % (v/v) formic acid. 1.9 nmol of protein in a total volume of 1 mL was applied to a Resource RPC column with 1 mL bed volume (Cytiva, Freiburg, Germany) equilibrated with buffer A (2 % (v/v) acetonitrile, 0.1 % (v/v) formic acid). For elution, a 0–100 % gradient of buffer B (80 % (v/v) acetonitrile, 0.1 % (v/v) formic acid) was applied over 10 bed volumes at a flow rate of 1 mL/min while monitoring the protein elution at 280 nm.

Mass spectra of intact proteins were measured on an impact II time-of-flight (ToF) mass spectrometer with an electrospray ionization (ESI) source (Bruker Daltonics, Bremen, Germany) in the positive ion mode. To this end, the purified protein was dialyzed against 10 mM ammonium acetate pH 6.6, followed by the addition of 50 % (v/v) methanol and 1 % (v/v) formic acid, and directly applied via a syringe pump at 90 μL/h. The following conditions for the ion transfer were used: 4500 V capillary voltage, 500 V endplate offset, 4 L/min dry gas at 200 °C, 1.8 bar nebulizer pressure and 10 eV collision energy. Raw spectra were deconvoluted with the Bruker Compass Data Analysis Software version 6.1 using the MaxEnt algorithm[64]. Mass spectra of chemical compounds were measured on a maXis ToF ESI-MS instrument (Bruker Daltonics) in the positive or negative ion mode, as appropriate. Average-isotopic molecular weights were calculated using the ProtParam server (https://www.expasy.org) or ChemDraw Professional 19.1.0.8 software (PerkinElmer, Waltham, MA).

### Protein functional studies

GST-*Strep*-GG and GST-*Strep*-GG-Pap were each expressed in NEBExpress(lowRF1) – with the ncAA omitted from the bacterial culture during expression of GST-*Strep*-GG – and purified from the total cell extract via Strep-Tactin or α-CD affinity chromatography, respectively. Purified proteins were dialyzed against GST assay buffer (0.1 M K-P$_i$ pH 6.5). A dilution series of 0–5 mM GSH, together with 2.5 mM CDNB (from a 62.5 mM stock solution in ethanol), in GST assay buffer (25 μL/ well) was mixed with an equal volume of the protein solution (1 μM GST, 2.5 mM CDNB in GST assay buffer) in a 96-well flat bottom microtest plate (Sarstedt, Nümbrecht, Germany) at 30 °C and the absorbance at 352 nm was measured every 20 s for 10 min in a Synergy 2 microtiter plate reader (BioTek, Winooski, VT) using Gen5 version 1.9 software (BioTek). K$_M$ values were determined with OriginPro 2023 10.0 (OriginLab) using a Michelis-Menten fit.

Real-time surface plasmon resonance (SPR) spectroscopy was performed on a BIAcore X100 instrument (GE Healthcare) with HEPES-buffered saline supplemented with Tween (HBST; 20 mM HEPES/NaOH pH 7.4, 150 mM NaCl, 0.005 % (v/v) Tween 20) as running buffer. A 5 μg/mL dilution of AffiniPure Mouse Anti-Human IgG, Fcγ fragment specific antibody (Jackson ImmunoResearch, West Grove, PA) in 10 mM Na-acetate pH 5.0 was immobilized on a CM5 sensorchip (Cytiva) via NHS/EDC amine coupling. A recombinant hEGFR extracellular domain fused to human Fc (SinoBiological, Eschborn, Germany) was diluted in HBST to 20 μg/mL and immobilized at a surface density ΔRU ≈ 150. Nb(EgA1)-*Strep*-GG-Pap, expressed in NEBExpress(lowRF1) using a 100 mL culture and purified from the whole cell extract via α-CD affinity chromatography, was dialyzed against HBST, and a dilution series (0.5–8 nM) was injected at a flow rate of 25 μL/min. Binding parameters were determined using BIAcore X100 evaluation software version 2.0 with a 1:1 Langmuir fit.

### Light-controlled protein affinity chromatography on an α-CD matrix

A Pierce disposable 2 mL column was packed with 1 mL bed volume of α-CD-conjugated Sepharose (see Suppl. Methods) and equilibrated with chromatography buffer A (25 mM Tris/Cl pH 8.0, 150 mM NaCl) or chromatography buffer B (50 mM Tris/Cl pH 8.0, 100 mM K$_2$SO$_4$). A periplasmic extract of E. coli was prepared (see above) and dialyzed against the chromatography buffer prior to application. Alternatively, for a whole cell lysate, the bacterial pellet was resuspended in the chromatography buffer, lysed by sonication with a Digital Sonifier (Branson Ultrasonics, Danbury, CT) and cleared by centrifugation. To prevent transient binding of MBP from E. coli to the α-CD matrix, the cell lysate was supplemented with 5 mM maltose as a competing ligand. The protein extract/lysate was passed through a sterile filter (0.2 μm, Filtropur S; Sarstedt) and applied to the α-CD affinity column, followed by washing (under ambient light) with the chromatography buffer. Elution of the Azo-tagged protein was achieved in the same buffer (in the dark laboratory) by illuminating the affinity matrix at 355 nm with a set of 8 or 16 LEDs arranged laterally around the column

(see Supplementary Fig. S1). Optionally, to ensure complete regeneration of the column, 1 mL of 0.5 M NaOH was applied, followed by rinsing with water and reequilibration with chromatography buffer.

For the exemplary separation of a mixture of two different intensely colored reporter proteins, Azurin and mScarlet, protein preparations previously purified with the help of the *Strep*-tag II were applied. To this end, Azurin-*Strep*-GG-Pap was first purified via SAC and then also passed through the α-CD affinity column in order to remove that fraction of Azurin which had not incorporated the C-terminal Pap residue via amber suppression (see Supplementary Fig. S12). The Azo-tagged Azurin was reverted to the *trans*-state by exposure to 430 nm light and then mixed with the SAC-purified mScarlet protein lacking the Azo-tag. Finally, the mixture of both proteins was applied to the α-CD affinity column and the chromatography was conducted as described above, including the UV illumination step at 355 nm for the elution of the Azo-tagged Azurin after the mScarlet protein had been washed out. To quantify the mScarlet fluorescence in individual chromatography fractions, 100 μL samples were each transferred to a 96-well Nunc black microwell plate (Thermo Fisher Scientific) and the fluorescence intensities were measured in a Synergy 2 microtiter plate reader with 530/25 nm and 590/35 nm filters for excitation and emission, respectively.

Detailed assessment of the effects of light exposure during washing/elution as well as further characterization of the chromatographic performance were accomplished using a new column device that allows individual illumination at 355 nm or 430 nm (see Supplementary Fig. S1E to G). mScarlet-*Strep*-GG-Pap and Azurin-*Strep*-GG-Pap were purified via SAC, followed by α-CD affinity chromatography to remove untagged protein, as described above. Samples were exposed to daylight for 2 h before performing the experiments shown in Supplementary Figs. S13 to S15. Chromatography runs were carried out with a sample volume of 100 μL mScarlet-*Strep*-GG-Pap (3 mg/mL), followed by washing with 2 mL chromatography buffer B and elution with 2–4 mL of the same buffer, after exposure to 355 nm UV light. 200 μL fractions were collected and 100 μL of each fraction was transferred to a black microtiter plate to measure the fluorescence as described above. A sample volume of 300 μL Azurin-*Strep*-GG-Pap (1.5 mg/mL) was applied in the analogous experiments with chromatography buffer A as running buffer. To assess the dynamic column capacity, a 1 mg/mL solution of the purified Azurin-*Strep*-GG-Pap in chromatography buffer A was continuously applied to the column in the dark. 200 μL fractions were collected in a flat-bottom 96-well microtest plate (Sarstedt) and the absorbance at 620 nm was measured in a Synergy 2 microtiter plate reader.

To test the effect of reducing agents on an Azo-tagged POI, 90 μL mScarlet-*Strep*-GG-Pap was mixed with 10 μL of 100 mM reduced L-glutathione (GSH), 50 mM tris(2-carboxyethyl)phosphine hydrochloride (TCEP) or 100 mM NADPH tetrasodium salt, each dissolved in chromatography buffer B. The samples were incubated for 16 h at room temperature in the dark and α-CD affinity chromatography was then performed as described above.

### α-CD affinity chromatography in the 96-well microtiter format
Two rows (2 × 8 wells) of a 96-well receiver plate with inserted filter frits (20 μm pore size; Macherey-Nagel, Düren, Germany) were filled with α-CD-conjugated Sepharose to yield a bed volume of 200 μL per well. The β-lactamase AmpC from *Enterobacter cloacae* WS1293 (UniProt ID: C0M550) was produced with and without a C-terminal Azo-tag in NEBExpress(lowRF1) using a 50 mL LB/Amp culture (see above). Untransformed NEBExpress(lowRF1) cells were grown in 50 mL LB medium as a negative control. Each cell pellet was lysed in 10 mL chromatography buffer A by sonication, and 0.5 mL aliquots of the cleared lysate were applied per well on top of the α-CD affinity matrix (ten wells with AmpC-GG-Pap and three wells each with untagged AmpC and the lysate from untransformed NEB Express(lowRF1)). Each of these

wells was washed four times with 200 μL chromatography buffer A under daylight. The wells were then exposed to 355 nm UV light from above (see Supplementary Fig. S1H) for 20 s and, directly thereafter, 200 μL chromatography buffer A was applied while continuing with UV illumination to elute the bound POI. The eluate was either analyzed by SDS-PAGE or diluted 1:100 in chromatography buffer A for a subsequent enzymatic assay. To this end, 30 μL of the diluted eluate was mixed with 20 μL of 1.5 mM CENTA β-lactamase substrate in a 96-well flat bottom microtest plate (Sarstedt). The absorbance at 405 nm was monitored at 20 °C in a Synergy 2 microtiter plate reader, and the initial slope of the resulting curve was used as a measure of enzyme activity.

Expression of GST[50] with an N-terminal Azo-tag (Gly-Pap-GG) and purification in the 96-well format was performed as described above for AmpC, this time replacing the chromatography buffer by GST assay buffer (0.1 M K-$P_i$ pH 6.5). For the enzyme assay, a similar setup was used as described above. 25 μL of the eluate from the α-CD affinity chromatography was mixed in a microtest plate with 25 μL substrate solution comprising 2 mM GSH and 2 mM 1-chloro-2,4-dinitrobenzene (CDNB) in GST assay buffer, and the change in the absorbance at 430 nm was monitored.

The nanobody (Nb) 2Rs15d[53] with an N-terminal Gly-Pap-GG tag was expressed in the cytoplasm of NEBExpress(lowRF1) and purified as described above for AmpC. The coding region for the scFv T84.66[51] was cloned on the pSB19-PapRS#34 plasmid with the OmpA signal sequence for periplasmic secretion as well as a C-terminal Azo-tag (GG-Pap) and produced in NEBExpress(lowRF1) in 500 mL LB/Amp medium. scFv gene expression was induced at $OD_{550} \approx 2.0$ and cells were cultivated for 3 h at 22 °C. A periplasmic extract was prepared (see above), yielding a final volume of 15 mL. α-CD affinity chromatography was performed as described for AmpC, this time applying 1 mL periplasmic extract per well. The activities of the eluted Nb and the scFv were subsequently assessed by ELISA. To this end, a 96-well Nunc Maxisorp plate (Thermo Fisher Scientific) was coated with 5 μg/mL human CEA-Fc chimera (Genscript, Piscataway, NJ) or HER2 extracellular domain (EMP Genetech, Ingolstadt, Germany) in PBS (4 mM $KH_2PO_4$, 16 mM $Na_2HPO_4$, 115 mM NaCl, pH 7.4) at 4 °C overnight and blocked with 3 % (w/v) bovine serum albumin (BSA) in PBS/T (PBS supplemented with 0.1 % (v/v) Tween-20) for 2 h at room temperature. The plate was washed once with PBS and then incubated with 50 μL/well of each of the collected eluates for 1 h. After washing three times with PBS/T, the plate was incubated for 1 h with StrepMAB-Classic HRP (mouse anti-*Strep*-tag HRP conjugate; IBA Lifesciences) diluted 1:2500 in PBS/T, followed by washing twice each with PBS/T and PBS. The ELISA was developed by addition of 50 μL/well ABTS HRP substrate (1 mg/mL in ABTS buffer) and the change in absorbance at 405 nm was monitored at 30 °C.

### Light-controlled α-CD affinity purification of an antibody from cell culture medium using an adapter molecule
The C2 domain of *streptococcal* protein G[54] was expressed with an N-terminal Azo-tag (Gly-Pap-GP) and a C-terminal *Strep*-tag II in the cytoplasm of NEBExpress(lowRF1). The Azo-tagged protein fragment (Azo-ProtG) was purified by Strep-Tactin affinity chromatography and, to remove residual impurities, the eluate was applied to a 1 mL α-CD affinity column. Azo-ProtG was eluted by irradiation with 355 nm UV light, followed by exposure of the eluate to 430 nm light for 15 min to completely revert Pap to the *trans*-configuration. 1 mg trastuzumab (Herceptin®; Roche, Grenzach-Wyhlen, Germany) was mixed with 1 mL Dulbecco's Modified Eagle's Medium (DMEM; gibco / Thermo Fisher Scientific) containing 5 % (v/v) fetal calf serum (FCS; PAN-Biotech, Aidenbach, Germany), supplemented with an equimolar amount of Azo-ProtG, incubated on ice for 10 min and centrifuged (3 min, 20,000 × *g*). The clear protein solution was applied to the α-CD affinity column (1 mL bed size), followed by washing with 3 mL PBS. Bound protein was eluted in 1.5 mL PBS under exposure to 355 nm UV light

and finally concentrated using an Amicon Ultra-4 centrifugal filter (30 kDa MWCO; Merck Millipore, Darmstadt, Germany).

## Generation of *E. coli* gene knock-out strains

Genomic deletions in the *E. coli* B strain NEBExpress (New England Biolabs) were performed using the pKD46 plasmid encoding the three structural genes, γ, β and exo, for the λ Red recombination system[65]. ~500 bp regions upstream and downstream of the target reading frame were amplified by PCR from the genomic DNA using suitable oligodeoxynucleotide primers (Eurofins Genomics, Ebersberg, Germany). In parallel, the neomycin phosphotransferase gene from transposon Tn5, which encodes aminoglycoside 3'-phosphotransferase (APH) and confers kanamycin resistance (*kan*[r]; UniProt ID: P00552) and the gene for aminoglycoside/streptomycin adenylyltransferase (AadA1) from Tn21 (*str*[r]; UniProt ID: P0AG05) were amplified from available plasmids. To this end, extended primers were used that exhibited 5'-sequence overlaps with each one of those primer pairs that were previously employed for the amplification of the genomic sequences flanking the target gene on both sides. In this manner, the reading frame of the target gene was precisely substituted by the one of the enzymes conferring antibiotic resistance. Assembly of the three PCR products – two from both flanking regions of the target gene, one from the antibiotic resistance gene – was accomplished using overlap extension PCR with primers hybridizing at both extreme ends, and the resulting DNA fragment was purified by agarose gel electrophoresis.

*E. coli* NEBExpress was first transformed with the pKD46 plasmid and then typically grown in 2 L Terrific Broth (TB) medium supplemented with 100 μg/mL Amp at 30 °C in the presence of 0.2 % (w/v) arabinose to induce expression of the λ Red genes. Bacteria were harvested at $OD_{550} \approx 0.5$ by centrifugation, washed three times in cold 10 % (v/v) glycerol, and resuspended in 200 μl GYT medium to yield approximately 1 mL cell suspension[65,66]. 100 μl of this cell suspension was mixed with 500 ng of the assembled DNA fragment from above in 10 μl ddH$_2$O and subjected to electroporation (5 ms, 12.5 kV/ cm; MicroPulser; Bio-Rad, Munich, Germany). After that, the bacteria were resuspended in 2 mL SOC medium and incubated at 37 °C under agitation for 1.5 h. Serial dilutions of this culture were plated on agar with LB medium containing different concentrations of the appropriate antibiotic (10–35 μg/mL kanamycin or 30–100 μg/mL spectinomycin) and incubated at 40 °C for 24 h. Single colonies obtained at the highest possible antibiotic concentration were picked and tested for successful deletion of the target gene by colony PCR using suitable flanking primers, followed by double-stranded DNA sequencing of the amplified region (Eurofins Genomics). Positive clones were used for inoculating agar culture plates containing the corresponding antibiotic and, in parallel, on plates containing Amp; the absence of duplicate colonies on the latter indicated the successful curing from the pKD46 plasmid.

## Optimization of PylRS for Pap substrates via directed evolution

The structural gene encoding the *M. mazei* PylRS variant with the previously introduced mutations A302T, L309S, N346V, and C348G[34] was synthesized with optimized codon usage for *E. coli* (Twist Biosciences, South San Francisco, CA), also including the conservative mutation K192R to incorporate a second *Bsa*I cleavage site. The resulting coding region for the so-called PylRS variant MmPSCaaRS(BsaI) was cloned, via the *Avr*II and *Sac*I restriction sites, on the vector pSB15, which carries the genes for *M. barkeri* PylRS[67] together with its cognate suppressor tRNA[Pyl] as well as the sfGFP reporter protein[35] harboring an amber stop codon at the permissive amino acid position 39. The published mutation Y384F[39] was introduced into MmPSCaaRS(BsaI) via QuikChange mutagenesis, yielding PapRS#0.

For further optimization, the central gene fragment of PapRS#0, flanked by the two *Bsa*I restriction sites, was subjected to error-prone PCR using the GeneMorph II random mutagenesis kit (Agilent), adjusting a mutagenesis frequency of ≤4 nucleotide exchanges. The PCR fragment comprising the pooled population of mutants was purified by agarose gel electrophoresis and cloned again on pSB15-PapRS#0-sfGFPa39 via the pair of *Bsa*I sites. Electroporation of *E. coli* NEBExpress(lowRF1) with 4 μg ligated DNA yielded a library of $5 \times 10^8$ transformants, which was plated on four square LB/Amp agar plates (12×12 cm$^2$). Colonies were scraped from these plates and resuspended in 50 mL LB/Amp medium. The bacterial suspension was used to inoculate 200 mL LB/Amp medium, followed by cultivation at 30 °C to $OD_{600} \approx 0.4$. 20 mL of this culture was supplemented with 1 mM NH$_2$-Pap (from a 50 mM stock solution in 100 mM HCl) and, after 1 h, expression of the sfGFP gene (under control of the *tet*[p/o]) was induced by adding 200 ng/mL anhydrotetracycline (aTc), followed by incubation at 30 °C for 16 h. Finally, the bacteria were collected by centrifugation, and the pellet from 1 mL culture was washed once in 1 mL PBS and then resuspended in the same volume of PBS.

This sample was applied to a FACS Aria IIu instrument (BD Bioscience, Heidelberg, Germany) for fluorescence analysis and sorting of the bacteria with FACSDiva 8.0 software (BD Biosciences). The instrument was operated with PBS as sheath fluid using a 488 nm Laser for excitation as well as a 502 nm long-pass filter with a 530/30 band-pass filter for specific detection of the sfGFP fluorescence. Live bacteria were selected in the "yield" mode with an appropriate FSC/SSC gate while setting a sorting gate to collect 25 % of those cells with the highest sfGFP fluorescence ($4 \times 10^7$ events in total). Bacteria were directly collected in LB/Amp medium, plated on LB/Amp agar and incubated at 30 °C overnight. Colonies were collectively resuspended in 200 mL LB/ Amp medium, and 50 mL of this suspension was subjected to plasmid DNA extraction (Midiprep Kit; Qiagen, Hilden, Germany). This mixed plasmid DNA was used as template for a second cycle of error-prone PCR and FACS as described above. The resulting library was subjected to several positive and negative selection cycles following a published procedure[31]. For each cycle, the collected bacteria were scraped from the agar plates and used to inoculate 100 mL LB/Amp medium with an initial $OD_{600} = 0.05$. When the culture had reached $OD_{600} \approx 0.4$, 10 mL were each transferred to two 25 mL shake flasks, one of which was supplemented with 0.5–1 mM NH$_2$-Pap. sfGFP expression and preparation of FACS samples were carried out as above. This time, sorting gates were set to collect 1–10 % of the population with the highest sfGFP fluorescence in the presence of NH$_2$-Pap for positive selection, while raising the gate after each cycle. For negative selection, the gate was set to sort cells with a low fluorescence signal, comparable to the initial PapRS#0, in the absence of the ncAA. Collected bacteria were grown again on LB/Amp agar. In total, three positive selection cycles were carried out, followed by two negative selection cycles and a final positive selection. Sample preparation for FACS analysis of individually cultured clones was performed essentially as described above, applying both Pap and NH$_2$-Pap at different concentrations. After selecting an appropriate FSC/SSC gate, sfGFP fluorescence of $10^5$ cells was recorded and the median of the fluorescence intensity (MFI) was determined. FACS data were analyzed with FlowJo software 10.10 (BD Bioscience). Sanger sequence analysis of PylRS mutants was performed using plasmid DNA isolated from individual colonies (Eurofins Genomics). Individual favorable mutations were combined as described in the text via subcloning using a *Bam*HI restriction site at the amino acid positions P317/D318 in conjunction with the *Sac*I restriction site.

## Statistics & reproducibility

Dots in graphs correspond to single measurements from individual experiments (biological replicates) unless stated otherwise. The number of replicates (N) is indicated in the graphs and Figure legends

(for N > 1). For FACS measurements, dots in bar graphs correspond to the MFI of $10^5$ recorded events. Bars show the mean, error bars indicate standard deviation; statistics in box plots are based on measurements of single wells from the same plate (here serving to document the well-to-well variability). All protein purification experiments documented by SDS-PAGE were performed at least in duplicate.

## Reporting summary

Further information on research design is available in the Nature Portfolio Reporting Summary linked to this article.

## Data availability

All data supporting the findings of this study are available within the paper and its Supplementary Information as well as the Source Data file. The genomic sequence of NEBExpress(lowRF1), including the raw NGS data, has been deposited at Genbank under accession code PRJNA1190201, and the cloning vector pSB19-PapRS#34-OmpA, including its full nucleotide sequence, is available from Addgene as Plasmid #231105 (https://www.addgene.org/231105). The previously published PylRS crystal structure used in this work can be found under accession number 2ZCE. Source data are provided with this paper.

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

## Acknowledgements

This work was financially supported through the Federal State of Bavaria and the Deutsche Forschungsgemeinschaft by providing the ESI Q-TOF LC-MS instrument in frame of their Major Research Instrumentation Programme (Grant No. INST 95/1734–1). The authors are grateful to Prof. Dr. Thomas Hofmann and Dr. Oliver Frank for support in the ¹H-NMR measurements.

## Author contributions

P.M. conducted the aaRS evolution and plasmid and strain modifications and carried out affinity chromatography and biochemical experiments. M.R.A. performed the chemical synthesis of Pap, its chemical and spectroscopic characterization as well as CD interaction analyses. K.S. and C.A.S. generated the NEBExpress(lowRF1) strain. I.T. conducted molecular cloning and protein expression as well as purification. M.S. performed the SPOT assay, RPC, and CD spectroscopy. S.A. constructed the illumination devices and conducted mass spectrometry. A.S. conceived and coordinated the project, designed the illumination devices and analyzed the bacterial genome sequence. A.S., P.M., and M.R.A. analyzed the data and wrote the manuscript with input from all authors.

## Funding

## Competing interests

P.M., M.R.A., S.A., and A.S. are inventors on a patent application related to this work. The remaining authors declare no competing interests.
