## [Transparent Peer Review file · Nature Communications]

Protein purification with light via a genetically encoded azobenzene side chain

Corresponding Author: Professor Arne Skerra

Version 0:

Reviewer comments:

Reviewer #1

(Remarks to the Author)

The manuscript "Protein purification with light" by P. Mayrhofer and co-authors demonstrates a novel method of affinity chromatography based on photomodulated adherence of azobenzene (installed in form of Azo-tag on the protein of interest) and immobilized alpha-cyclodextrin. This interaction, known and applied in the area of smart materials and light-triggered gelator systems, has been creatively implemented here into the novel domain - protein purification. The work is original, significant, and creative, as it demonstrates a complementary approach to the established protein affinity purification techniques like NiNTA/His6 or FLAG tag. The obvious advantage over these methods is lack of chemical modification of the protein's environment in order to detach it from the column (which may have potentially denaturing effect on that protein). The methodology is sound, and it may find interesting applications, particularly in regular purification of selected proteins otherwise sensitive to denaturation.

Yet, although I share the authors' enthusiasm towards their new method, I seriously doubt that it would become competitive e.g. to the NiNTA/hexahistidine tag, or any other tag which requires simple insertion or mutation of proteinogenic aminoacids. Therefore, I enthusiastically recommend publication of the manuscript, providing that the points listed below will be sufficiently addressed by the authors, and that they will provide a critical discussion about objective limitations of their new methodology in the conclusions of their manuscript, in order to make the less-versed readers aware e.g. of challenges associated with the introduction of unnatural aminoacids into the protein structure, being far from the routine operation until now.

Please provide explicit and detailed comments on the following issues, that may have substantial meaning to the readers that would eventually like to apply your method in purification of their protein systems:

- 1) how difficult is in reality to implement the demonstrated system into a random protein of interest? What most common limitations can be expected upon decoration of a protein with the Azo-tag? Please make a separate paragraph with systematic discussion on this issue, based on all the results collected in your manuscript.
- 2) location of the Azo-tag. N-terminus vs. C-terminus: you already discusses some limitations on the pages 9-10, lines 408-451. Are there some other details that one has to keep in mind once deciding if the Azo-tag should be located at the N- or C-terminus? Is it also possible to insert the Azo-tag (e.g. the SPOT-optimized Gly-Baf-Gly) in the middle of the protein sequence, e.g. in a loop of a protein that points outwards? Would this be of any benefit regarding the binding strength?
- 3) The system for amber-codon-suppression installment of the azobenzene-aminoacid (tRNA and the tRNA synthetase) has been optimized - a specific expression strain with the removed RF-1-encoding gene has been created, and the PyIRS has been optimized using directed evolution. Which elements of such systems are commercially available, which can be obtained (e.g. as plasmids) from your group, and which have to be optimized or re-designed in situ? Importantly, is the tRNA/tRNA-synthetase set optimized by you universally applicable for installation of the Azo-tag on every protein of interest? Or do both the components need to be optimized (e.g. by directed evolution) in each case anew, in order to achieve satisfactory amber codon suppression level with the Azo-aminoacid?

4) maybe I haven't found that information, but what is the amber codon suppression level in the demonstrated cases of proteins with C-terminal Azo-tag (page 9, lines 394-406)? That is, what is the ratio of the truncated protein with the sequence correctly stopped at amber, versus the full-length protein with the Azo-tag? Is this at the standard 10-20% of the engineered protein, or was the optimization combined with the RF-1 deletion giving more favorable yields of the suppression process in this case?

Obviously, with the N-terminal Azo-tag (described below, page 9, lines 409-424), truncated protein would not be produced at all (eventually the yield for the whole protein type would drop down)

5) insertion of the Azo-tag adds a significant element of hydrophobicity (particularly in the thermally stable E-isomer) to the protein's terminus. Even if a short spacer has been added into the Azo-tag to somehow moderate it, this might influence the folding of the tagged protein, and possibly its e.g. catalytic properties. Is there some recent literature that analyzes the influence of protein tagging on their properties? If so, you should include a short discussion on it as well.

Of course, a perfect experiment in the future (which I recommend for follow-up papers) would be to express an Azo-tagged mutant of an enzyme and compare its kinetic parameters with the wt-version. Yet, this would be an excessive requirement for the current paper demonstrating proof-of-principle. However, what can be done is taking the currently demonstrated Azo-tagged proteins (mScarlett and Azurin) and compare them with their wild-type untagged counterparts e.g. by circular dichroism, (which should indicate the eventual conformational changes of the tagged proteins, if blanked with the equimolar solution of the Azo-aminoacids to minimize the influence of the Azo chromophore) or by NMR techniques sensitive to conformational changes - to demonstrate that the overall protein structure before and after tagging is not heavily affected

6) I hope I did not overlook something, but in all demonstrated protein examples isolated by the Azo-tag in the manuscript I have seen also decoration with a Strep-tag. Most of them were also pre-purified by Strep-Tactin affinity chromatography prior to further assessing (page 7, lines 292-295).

The isolation from crude cell extract was demonstrated on page 8, lines 344 and following. There, the only significant impurity seemed to be the maltose-binding protein MBP (understandable for the type of column used). Yet, it would be important for this manuscript to demonstrate, how efficiently can the Azo-tagged protein be purified from the crude cell extract without the Strep-tag, just as a negative control that would exclude the possibility that the Strep-tag has some (selective or unselective) interaction with the CD-exposing column material. For that, the authors should express one of the proteins (eg. the already used cystatin C, now fused with the C-terminal GG-Baf sequence only, without the preceding Strep-tag II), isolate it again - but this time in one experiment from the periplasmic extract as described for cystatin C in the manuscript, and in the parallel experiment from the whole crude cell extract - using the a-CD-containing column to catch the Azo-tagged species, and see how this compares with the already demonstrated experiment both in terms of purity of the isolated cystatin C, and its yield (if its binding is equally efficient under all aforementioned conditions).

7) Another technical limitation, that needs to be determined prior to publication, is the efficiency of agarose penetration with the used UV-light source (355 nm). UV light, apart from being absorbed on many cellular components, can be also scattered by numerous materials. The proposed technology requires efficient penetration of the whole agarose deposit with 355 nm in order to detach the protein of interest. The authors have to measure in the quantitative manner the transmittance of 355 nm light through their material, e.g. with an actinometer and cuvettes (e.g. the typical UV-Vis quartz cuvettes) of increasing thickness filled with the used agarose beads in the same buffer, as demonstrated in the manuscript. The detector has to be isolated from the light source (so that there is no irradiation passing on the sides of the cuvette or the vial) and the amount of 355 nm light that can actually penetrate the beads without being scattered should be determined (some sort of calibration curve: transmittance of 355 nm light vs. the thickness of the agarose gel). You can probably use the classical UV-Vis spectrophotometer for that as well, providing that you measure the light transmittance. But it would be better to use your particular light source and a separate detector/actinometer, to get realistic assessment for your demonstrated setup. This will tell us, what the maximal thickness of the agarose column could be, in order to effectively apply the UV light-triggered release method in its whole volume.

8) one commonly known limitation of azobenzene applications in vivo is its susceptibility to intracellular reducing agents, in particular glutathione. There, the azo bond is reduced to the non-photochromic hydrazine. This property has been regularly monitored by the community, especially for photopharmacology applications. This might also be a serious concern for the Azo-tag installation protocol on the recombinant proteins. The standard model for in vitro testing the said susceptibility is incubation of the azobenzene of interest with 10 mM aqueous solution of reduced glutathione stabilized with 5 mM TCEP (using organic co-solvents like acetonitrile in case, if the azobenzene cannot be homogeneously dissolved in water). The authors have to incubate the azobenzene aminoacid used for feeding their cells with the aforementioned mixture, and determine the lifetime of the Azo-compound under these conditions, using spectrophotometry (UV-Vis spectrum, follow the absorption maximum frequency of the azobenzene, and plot its decay over time).

9) in the other direction - UV light may have negative influence on proteins, sometimes causing degradation. Please, provide experimental evidences that your proteins are resistant to the used wavelength of 355 nm at least over the time needed for isolation. Please prepare solutions of discussed proteins (mScarlett, Azurin, cystatin C) in regular buffers, expose the solutions to the irradiation setup used previously with your column, and take samples upon regular intervals - you can measure by HPLC-MS and gel electrophoresis, if the proteins' integration is kept, and for how long (I would suggest at least 60 minutes constant exposure, and sampling every 10 minutes).

10) the authors claim quantitative switching of the (E)-azobenzene to the "excited" (?- maybe they mean thermally less stable) (Z)-isomer based on UV-Vis spectra - this is not a proof. It only shows that the switching occurs between two photostationary states, which may both contain various E/Z isomer ratio. The experience and literature data show that photoisomerization of azobenzenes without strongly donating or accepting substituents typically produces up to 80% of the

Z-isomer in the photostationary state (although often in aqueous media the result of 30-40% may be observed). The authors should necessarily repeat the irradiation experiments and isomer ratio determination with NMR (identify at least three well-separated peaks that belong to each photoisomer, and compare their averaged integrations between the E- and Z-isomer peaks) and with HPLC (find the isosbestic point, where the absorbance of E and Z isomer are equal, then measure HPLC chromatograms before and after irradiation - there, the comparison of peak integration for the old (E)-isomer and the newly formed (Z)-isomer will indicate the isomer ratio).

Reviewer #2

(Remarks to the Author)

Mayerhofer et al. present an interesting, and to the best of my knowledge novel, strategy for the chromatographic purification of proteins using a light. The system relies on the light-dependent host-guest interaction between azobenzene and cyclodextrin, which is well documented in literature (e.g. Nie et al. 2021). For the tagging of proteins of interest (POIs), the authors rely on the amber stop-codon/non-canonical amino acid (nc-AA) strategy to incorporate L-benzazo-phenylalanine (Baf; also known as AzoPhe in the literature), which is incorporated into a short so-called Azo-tag (2-4 amino acids) attached to the N- or C-terminus of the protein during heterologous production in a specifically adapted E. coli strain. As shown by the authors for the isolated nc-AA, Baf can be switched between trans- and cis-states by illumination with 355 nm UV-A light, with the trans-form strongly interacting with α -cyclodextrin (KD approx. 91 μ M), with no detectable affinity between α -cyclodextrin and the cis-form of the nc-AA. This essentially allows the use of column materials with covalently attached α -cyclodextrin, which were synthesized by the authors. For the incorporation of Baf into the Azo-tag of a given POI, a one-plasmid system was developed carrying the Azo-tagged POI as well as the needed heterologous tRNA and the cognate aminoacyl-tRNA synthase. The authors went to great lengths to optimize both the Azo-tag sequence, the used aminoacyl-tRNA synthase as well as the expression host strain for optimal incorporation of Baf and selective and contaminant free elution by illumination. The Azo-tagged POI (with Baf in the trans-form) can be bound to the column in the dark (or dim daylight), unspecifically bound proteins are removed by washing with the same buffer and elution is carried out under buffer flow and lateral illumination of the column by UV-A light using LEDs. Proof of concept of the strategy was shown for two model proteins mScarlet (a fluorescent reporter) and Azurin (a blue, copper-binding protein of *Pseudomonas aeruginosa*). Purification was demonstrated from cytoplasmic crude cell extracts (mScarlet) and periplasmic extracts (Azurin targeted to the periplasm). In addition to these experiments with model proteins, applicability was verified for high-throughput, microscale purification of more realistic POIs: i) two enzymes: the monomeric beta-lactamase AmpC and the homo-dimeric glutathione S-transferase GST, ii) antibody fragments (single-chain variable fragment of a monoclonal antibody as well as a nanobody produced in E. coli. Finally, using an azo-tagged adaptor protein that binds to full-size antibodies, trastuzumab, a monoclonal antibody used to treat breast and gastric cancer, was isolated from mammalian cell culture medium.

The presented work is certainly novel and original, the results sound and the manuscript in most parts clearly written and illustrated (for some exceptions see below). If indeed as widely applicable as the authors propose, the strategy could be of general interest to a wide range of researchers and thus deserves publication in Nature Communications. However, the manuscript in its current form leaves several potential issues unaddressed, which would be essential for wide-spread application. However, if the authors can address these issues, I would wholeheartedly recommend publication.

Major comments

1. An important issue for any protein purification strategy is the achievable yields (which depend on the POI but also on the binding capacity of the column) and purities (that depend on the selectivity of the column-POI interaction). The authors have sufficiently demonstrated that their strategy can yield pure protein in a single step, but they leave other important questions unanswered, such as e.g. the dynamic binding capacity of their column material. How much protein can be bound to e.g. 1 mL of resin at a given flow rate before 10% of the target protein accumulates in the flow-through fraction ("breakthrough"). Similarly, what about reuseability and chemical stability of the material? Please note that I am not advocating for a complete characterization of the column material which would rather be needed if commercialization of the strategy is attempted. The authors should however provide some basic characteristics to address dynamic binding capacity, reuseability and chemical stability with regard to common components used for protein stabilization (e.g. DTT, TCEP). Likewise, it would be interesting if purification under denaturing conditions (e.g. in the presence of Urea) is possible.

For a good overview see: For a good overview see e.g.

<https://www.thermofisher.com/de/de/home/life-science/protein-biology/protein-biology-learning-center/protein-biology-resource-library/protein-biology-application-notes/performance-characterization-his-pur-nickel-superflow-agarose.html>

2. Another important issue to consider when using UV-A light for protein elution is whether the light will cause oxidative modifications of the POI, which in turn could affect function. UV-A light is known to cause oxidative damage to proteins either directly or via the generation of reactive oxygen species such as singlet oxygen. See e.g. Hawkins et al 2019 for an overview. Some efforts should be made to rule out that the light-dependent elution scheme used by the authors does not result in oxidative damage to the POI.

3. Along the same lines, how much light is needed to trigger elution from the column? Can this be quantified?

Minor comments:

The strategy presented by the authors is hardly the first that allows purification of proteins by light. The authors should cite and discuss other strategies for light dependent protein purification. What are advantages/disadvantages of their method relative to previously presented studies? Examples include the mem-iLID strategy (Tang et al 2021), light-switchable

nanobodies (Gil et al. 2020) and the use of Phytochrome-PIF6 interaction for affinity-based POI purification (Hörner et al. 2019).

Line 109: "Quantitative switch to the excited state, i.e. the cis-configuration ..." As far as I know, the cis-configuration of the azobenzene is not an electronic excited state. Both the trans and the cis-state are electronic ground states, while isomerization proceeds via excited state formation.

Figure 1, F: Kinetic stability of the cis-state: Kinetics have not been recorded sufficiently long so that an exponential fit would yield meaningful values. The data shown in the Figure could be equally well fitted with a linear function. Measurements would need to be conducted much longer, e.g. three times the lifetime of the exponential decay. I am aware, that this might experimentally not be feasible here. So either, carry out longer measurements mention this caveat.

In Figure 3: Please schematically illustrate the strategy for the adaptor-protein based purification of trastuzumab.

Line 219: "Initially, we utilized superfolder GFP (sfGFP) in order to select E. coli cells carrying a mutated PyIRS for incorporation of Baf at its structurally permissible sequence position 39 via fluorescence-activated cell sorting (FACS)" It is unclear to me, if this sentence describes work that was carried out as part of this study (then some Figure should be referenced) or if the authors refer to some of their previous work. Please clarify!

Line 278: "which also made the use of beta-CD for solubilizing Baf less important in our hands" This doesn't say clearly if the authors used this strategy for the next experiments or not. Please clarify!

References

C. Nie, C. Liu, S. Sun, S. Wu, Visible-Light-Controlled Azobenzene-Cyclodextrin Host-Guest Interactions for Biomedical Applications and Surface Functionalization. 2021. ChemPhotoChem 5: 893.

Hawkins CL, Davies MJ. Detection, identification, and quantification of oxidative protein modifications. J Biol Chem. 2019. 294:19683-19708.

Gil AA, Carrasco-López C, Zhu L, Zhao EM, Ravindran PT, Wilson MZ, Goglia AG, Avalos JL, Toettcher JE. Optogenetic control of protein binding using light-switchable nanobodies. Nat Commun. 2020. 11: 4044

Hörner M, Eble J, Yousefi OS, Schwarz J, Warscheid B, Weber W, Schamel WWA. Light-Controlled Affinity Purification of Protein Complexes Exemplified by the Resting ZAP70 Interactome. Front Immunol. 2019. 10: 226.

Tang R, Yang S, Nagel G, Gao S. mem-iLID, a fast and economic protein purification method. Biosci Rep. 2021. 41:BSR20210800

Reviewer #3

(Remarks to the Author)

The authors presented a light-switchable affinity tag, Baf, that binds to CD with high affinity (91 μ M) in the trans-form (ground state) and low affinity in the cis form (excited state). The dissociate constant is comparable to Strep-tag (13 μ M), developed by the same lab, and enables protein purification, with a linked tag, under the stimulation of mild UV light. Compared to commonly used affinity chromatography, which requires additional incubation of elution buffer, this "excitography" uses light to trigger the dissociation of Baf-labeled POI and CD-containing matrix.

The data presented in this work is clear and organized. One of the major concerns, however, is the requirement for non-natural amino acids and tRNA-synthase (e.g., PyIRS) to be incorporated into the target protein. As demonstrated by the author, PyIRS needs to be optimized, special strains of bacterial expression system need to be used, and the location of Baf infusion and even the neighboring amino acids around Baf should be optimized. A general question is how these optimized conditions apply to other proteins not demonstrated in this work and if there are any limitations for the POI in terms of size, pKa, and other physicochemical properties.

It seems that Baf is not commercially available and must be synthesized, impeding its wide applicability to labs without experience or equipment necessary to carry out the required chemistry. Please include a brief discussion in the text about its availability (or lack thereof).

Some technical questions:

What is the purpose of including the Strep-tag in the fusion POI? Is this simply a convenience for cloning or is it somehow required for Baf to work?

Minor concern:

It would be nice to comment on the different trends of A326 when titrating trans-Baf with α - (Fig. 1I) or β -CD (Fig. 1K).

Fig. 2G. The initial distribution of mScarlet-Streptag-Baf seems to be much broader than Aruzin-Streptag-Baf. In this case, what would be the effect of POI size on the effect of binding between trans-Baf and CD?

There is no mention in the text about the availability of any of the plasmid variants required to produce recombinant proteins that incorporate Baf. If plasmids are not deposited in Addgene or another suitable distributor to make these materials

available, please ensure that readers can access or obtain the full sequences of the most relevant/effective plasmid backbone that contains the machinery to incorporate Baf, for example the one incorporating the sfGFP construct. This will help other investigators to apply this technique to their own systems.

On lines 106-7 of the supplementary methods, the text reads “purified PCR fragments were mixed in approximately equimolar amounts together with T4 DNA polymerase (Thermo Fisher Scientific) in the buffer supplied by the manufacturer” – this may be a typo referring to T4 DNA Ligase?

Reviewer #4

(Remarks to the Author)

Key results

The manuscript entitled “Protein purification with light” presents a strategy to purify protein samples using a light-sensitive tag. The tag comprises the azobenzene based unnatural amino acid known as AzoF (here named Baf), which allows the binding of the protein in the dark or under day-light to an α -cyclodextrin resin. Elution of the protein of interest is then facilitated by irradiation with UV light. The presented research appears to be original and innovative, and a lot of work and thought has been put into the success of this strategy from a biochemical point of view. I am particularly convinced that this method will have an impact on diverse fields of protein chemistry due to the extremely high binding efficiency and purity of the eluted samples. I therefore very much like to see this manuscript published. However, I have two major concerns regarding the publication in a high-impact journal such as Nature Communications, which should be addressed appropriately.

Key concerns

First, I have to point out that its photochemical/physical behavior is misrepresented (as I will explain in more detail below). As a result thereof, data were misinterpreted and other data that are crucial to show the reproducibility of this technique are missing. Second, although they nicely emphasized on the benefits of this strategy and minimized some limitations of it, the authors missed to discuss the (remaining) major limitations of using an unnatural amino acid in a protein tag. I have provided detailed arguments in the attached file sorted into major and minor points. These illustrate how the manuscript should be revised to entirely cover the state-of-the-art and how it should be complemented with further experiments to further analyze the principle and improve the reproducibility of this purification strategy (which in my opinion might also be necessary for the pending patent).

Reviewer #5

(Remarks to the Author)

I co-reviewed this manuscript with one of the reviewers who provided the listed reports as part of the Nature Communications initiative to facilitate training in peer review and appropriate recognition for co-reviewers.

Version 1:

Reviewer comments:

Reviewer #1

(Remarks to the Author)

I highly appreciate the effort of authors to improve the quality of the manuscript and to address the comments of me and other reviewers. From my perspective, my numerous comments have been sufficiently addressed by the authors, and thus I recommend the acceptance of the manuscript in its current form. I particularly appreciate the effort to determine the isomeric composition via NMR. I agree that the overall effect (binding vs. release) occurs with satisfactory selectivity level. Yet, in light of the multiple problems known to the photoswitch community and often successfully addressed in the last two decades, it was important to understand the behavior of Azo label at the molecular level. The results obtained from the quantification are not impossible, but pretty uncommon for the type of photoswitch selected for this research. Therefore, this quantification was needed for understanding and justification of the conclusions drawn by authors.

Yet, as the demonstrated experiments do not occur solely in solution, but in interactions with the solid phase, it is not only the photoconversion, but (apparently predominantly) the sequestering of the trans-isomer (whenever it is generated in the photodynamic equilibrium) to the solid phase, which determines the macroscopic outcome.

While the authors should not be criticized for not addressing all problems known to the photoswitch and photopharmacology community, careful analysis of the comments from me and other reviewers obviously helped to increase the mutual understanding.

Minor (successfully addressed) points:

- penetration of the solid phase of the column with light - while this is probably irrelevant for the diameter demonstrated in the publication, the scattering and attenuation might have still be a potential concern for larger columns. Therefore it is good that authors determined the transmittance in the used setup, which will clearly indicate what level of limitation would that issue eventually be able to cause (or not).

- reductive conditions and azobenzenes - this is a serious concern for azobenzene applications, especially in vivo or with physiological fluids. This was carefully addressed in many publications from e.g. Andrew Woolley lab (e.g. Samanta et al. J. Am. Chem. Soc. 2013, 135, 26, 9777–9784 <https://doi.org/10.1021/ja402220t>). The fact that this issue was not addressed in

previous publications concerning amber codon-type biosynthetic incorporation of PAP or its derivatives, may rather refer to the overall inefficiency of the whole chain of operations, where this issue somehow might have been overshadowed by other bottlenecks

- careful determination of the photoconversions - looking at the earliest reports, like Bose et al JACS paper from 2006 (Pete Schultz group, J. Am. Chem. Soc. 2006, 128, 2, 388–389, <https://doi.org/10.1021/ja055467u>), the photoconversion of PAP was 55% to the cis isomer, return to trans-isomer was quantitative. These values are more typical for that chromophore, therefore the authors should not be surprised on the reviewers' sceptical comments on the photoconversion issue in your manuscript. It is good that you performed the quantification experiments.

Reviewer #2

(Remarks to the Author)

This is the revised version of a manuscript that I have previously reviewed. I thank the authors for carefully addressing all my concerns. The inclusion of performance data such as the determination of the dynamic binding capacity of the chromatographic material and data on its chemical stability is crucial for wider application. Regarding the kinetic measurement of cis-state stability, I still believe that increasing the measurement time would give a more robust fit, but since this is not an experiment that is crucial to the validity of the developed method, setting an asymptotic limit in the fitting routine and performing a single exponential decay fit will suffice. The manuscript has been greatly improved and from my side I gladly support its publication.

Reviewer #3

(Remarks to the Author)

In this revision, the authors added new experimental results and discussion and commented on the effort to make the materials accessible to the community. My concerns have been addressed.

Reviewer #5

(Remarks to the Author)

AUTHORS' RESPONSE TO REVIEWERS' COMMENTS

Reviewer #1 (Remarks to the Author):

The manuscript "Protein purification with light" by P. Mayrhofer and co-authors demonstrates a novel method of affinity chromatography based on photomodulated adherence of azobenzene (installed in form of Azo-tag on the protein of interest) and immobilized alpha-cyclodextrin. This interaction, known and applied in the area of smart materials and light-triggered gelator systems, has been creatively implemented here into the novel domain - protein purification. The work is original, significant, and creative, as it demonstrates a complementary approach to the established protein affinity purification techniques like NiNTA/His6 or FLAG tag. The obvious advantage over these methods is lack of chemical modification of the protein's environment in order to detach it from the column (which may have potentially denaturing effect on that protein). The methodology is sound, and it may find interesting applications, particularly in regular purification of selected proteins otherwise sensitive to denaturation.

Yet, although I share the authors' enthusiasm towards their new method, I seriously doubt that it would become competitive e.g. to the NiNTA/hexahistidine tag, or any other tag which requires simple insertion or mutation of proteinogenic aminoacids. Therefore, I enthusiastically recommend publication of the manuscript, providing that the points listed below will be sufficiently addressed by the authors, and that they will provide a critical discussion about objective limitations of their new methodology in the conclusions of their manuscript, in order to make the less-versed readers aware e.g. of challenges associated with the introduction of unnatural aminoacids into the protein structure, being far from the routine operation until now.

Au: Thanks for the generally positive assessment of our work. As a humble remark, the comparison with a technology, such as His-tag/IMAC purification, which exists since more than 30 years and, as most users know, is by far not successful in all cases, is not entirely fair. As we have exemplified in our manuscript, and will also point out in the following (including complementary experiments), the introduction of the Azo-amino acid for the preparative production of recombinant proteins is less complicated and more efficient than the Reviewer may think.

Please provide explicit and detailed comments on the following issues, that may have substantial meaning to the readers that would eventually like to apply your method in purification of their protein systems:

1) how difficult is in reality to implement the demonstrated system into a random protein of interest? What most common limitations can be expected upon decoration of a protein with the Azo-tag? Please make a separate paragraph with systematic discussion on this issue, based on all the results collected in your manuscript.

Au: We are not aware of any general limitations regarding the decoration of a protein with the Azo-tag. In our manuscript we have described the purification of in total 12 different proteins, some of these additionally with alternative arrangements of the Azo-tag; for an overview see the new Suppl. Table S1. During the last year we have furthermore purified almost the same number of additional proteins and, as most of these experiments were done by undergraduate students in our laboratory, we believe that our technique is pretty robust as it stands. One potential caveat may be that, like with many established affinity tags (and obviously due to sterical reasons), the optimal choice of N-terminal versus C-terminal positioning should be tested empirically for each POI, both with regard to efficient affinity purification and biochemical functionality. We have critically discussed this now as part of the amended Conclusions section (see lines 649–683 of our revised manuscript, referring to the red-line version). While questions for potential issues with hydrophobicity and chemical stability of the Azo-side chain have been clarified in response

to this Reviewer and others (see below) with additional experiments, now described in the revised manuscript (see, e.g. Suppl. Figs. S13, S16, S21, S22), we have briefly discussed these aspects, too, in the same context of the main manuscript.

2) location of the Azo-tag. N-terminus vs. C-terminus: you already discusses some limitations on the pages 9-10, lines 408-451. Are there some other details that one has to keep in mind once deciding if the Azo-tag should be located at the N- or C-terminus? Is it also possible to insert the Azo-tag (e.g. the SPOT-optimized Gly-Baf-Gly) in the middle of the protein sequence, e.g. in a loop of a protein that points outwards? Would this be of any benefit regarding the binding strength?

Au: Please see the amended Conclusions section mentioned above. So far, we have no experience with the suitability of the Azo-amino acid for affinity purification if positioned in the middle of a polypeptide chain, as we believe that this is a specialized application which requires individual engineering of a POI. However, the fact that the Azo-amino acid can be exposed directly on the surface of a folded protein is exemplified with our reporter protein for the FACS experiments, sfGFP, which carries the azobenzene side chain at position 39 (see Suppl. Results B).

3) The system for amber-codon-suppression installment of the azobenzene-aminoacid (tRNA and the tRNA synthetase) has been optimized - a specific expression strain with the removed RF-1-encoding gene has been created, and the PylRS has been optimized using directed evolution. Which elements of such systems are commercially available, which can be obtained (e.g. as plasmids) from your group, and which have to be optimized or re-designed in situ? Importantly, is the tRNA/tRNA-synthetase set optimized by you universally applicable for installation of the Azo-tag on every protein of interest? Or do both the components need to be optimized (e.g. by directed evolution) in each case anew, in order to achieve satisfactory amber codon suppression level with the Azo-aminoacid?

Au: We have now amended a new schematic figure in the Supplement which in detail describes how to clone (in a single step) the coding region of a POI on our optimized expression vector using different arrangements of the Azo-tag and optionally employing cytoplasmic expression or periplasmic secretion (see Suppl. Fig. S25). In our hands, among these strategies at least one has worked for all recombinant proteins tested so far (see Suppl. Table S1) without any need of reoptimization of the engineered PylRS or the bacterial expression strain. Our expression vector and the genetically modified host strain will be made generally accessible. The sequence of the vector will be submitted to AddGene for general distribution after acceptance of our manuscript. Likewise, we will submit our expression strain, NEBExpress(lowRF1), to a public repository. As an alternative, researchers may use the published pEVOL double vector system (see Suppl. Results C and references therein) encoding the previously described AzoPheRS for incorporation of the Azo-amino acid, even though with lower efficiency, and the synthetic E. coli strains C321.ΔA or B-95.ΔA, which were extensively engineered by different groups to lack most of the amber stop codons as well as RF-1 (see PMID: 24136966 and PMID: 25982672, respectively). The Azo-amino acid itself has been made commercially available meanwhile from several vendors, as explained in our response to Reviewer #3 further below.

4) maybe I haven't found that information, but what is the amber codon suppression level in the demonstrated cases of proteins with C-terminal Azo-tag (page 9, lines 394-406)? That is, what is the ratio of the truncated protein with the sequence correctly stopped at amber, versus the full-length protein with the Azo-tag? Is this at the standard 10-20% of the engineered protein, or was the optimization combined with the RF-1 deletion giving more favorable yields of the suppression process in this case?

Obviously, with the N-terminal Azo-tag (described below, page 9, lines 409-424), truncated protein would not be produced at all (eventually the yield for the whole protein type would drop down)

Au: The efficiency of amber stop codon suppression with our optimized system is much higher than the value mentioned (10-20 %, as a rule of thumb for natural bacterial suppressor tRNAs) and in fact amounts to on average 50 % (somewhat depending on the sequence context). This has been quantified by ESI mass spectrometry for the C-terminal Azo-tag (see Suppl. Fig. S12) from a comparison of the two species differing just by the mass increment for the azo-amino acid (due to not fully complete amber suppression) which had been copurified via the Strep-tag II. In addition, we have now quantified the yield of purified mScarlet3 protein equipped with the N-terminal Azo-tag and compared it with a version carrying a Phe residue instead of the azo-amino acid (see the amended Suppl. Results C), demonstrating that translational efficiency of the amber stop codon with our optimized system falls within the range for canonical codons.

5) insertion of the Azo-tag adds a significant element of hydrophobicity (particularly in the thermally stable E-isomer) to the protein's terminus. Even if a short spacer has been added into the Azo-tag to somehow moderate it, this might influence the folding of the tagged protein, and possibly its e.g. catalytic properties. Is there some recent literature that analyzes the influence of protein tagging on their properties? If so, you should include a short discussion on it as well.

Of course, a perfect experiment in the future (which I recommend for follow-up papers) would be to express an Azo-tagged mutant of an enzyme and compare its kinetic parameters with the wt-version. Yet, this would be an excessive requirement for the current paper demonstrating proof-of-principle. However, what can be done is taking the currently demonstrated Azo-tagged proteins (mScarlett and Azurin) and compare them with their wild-type untagged counterparts e.g. by circular dichroism, (which should indicate the eventual conformational changes of the tagged proteins, if blanked with the equimolar solution of the Azo-aminoacids to minimize the influence of the Azo chromophore) or by NMR techniques sensitive to conformational changes - to demonstrate that the overall protein structure before and after tagging is not heavily affected

Au: Indeed, in principle, the addition of a peptide tag might influence the biochemical properties of a recombinant protein. However, initial concerns in this regard at the time when the concept of affinity tags was invented (see our Introduction and the review articles quoted there) were largely mitigated over the years. This is reflected by the fact, for example, that meanwhile crystal structures of thousands of proteins carrying the His-tag or the Strep-tag (II) and the like, mostly with proven bioactivity, have been determined and deposited at the PDB. In this context, the influence of a single Pap residue, flanked by hydrophilic Gly residues, should be compared with both an exposed hydrophobic Phe and a Trp side chain in the case of the *Strep-tag* (II), which should let this appear less worrying. Nevertheless, we have now included functional characterisation both for an enzyme and an Ig fragment carrying the Azo-tag (see Suppl. Figs. S21 and S22) without signs of loss in activity. In addition, we have investigated the overall hydrophobicity of GST carrying either the Azo-tag or lacking this amino acid by analytical reverse phase chromatography (RPC) and not found detectable differences (see Suppl. Fig. S21D). Finally, we have measured CD spectra in the far UV region to analyse the secondary structure and, again, not found any influence of the Azo-tag (see Suppl. Fig. S21C). This has now also been mentioned in the main manuscript (lines 566–580).

6) I hope I did not overlook something, but in all demonstrated protein examples isolated by the Azo-tag in the manuscript I have seen also decoration with a Strep-tag. Most of them were also pre-purified by Strep-Tactin affinity chromatography prior to further assessing (page 7, lines 292-295).

The isolation from crude cell extract was demonstrated on page 8, lines 344 and following. There, the only significant impurity seemed to be the maltose-binding protein MBP

(understandable for the type of column used). Yet, it would be important for this manuscript to demonstrate, how efficiently can the Azo-tagged protein be purified from the crude cell extract without the Strep-tag, just as a negative control that would exclude the possibility that the Strep-tag has some (selective or unselective) interaction with the CD-exposing column material. For that, the authors should express one of the proteins (eg. the already used cystatin C, now fused with the C-terminal GG-Baf sequence only, without the preceding Strep-tag II), isolate it again - but this time in one experiment from the periplasmic extract as described for cystatin C in the manuscript, and in the parallel experiment from the whole crude cell extract - using the α -CD-containing column to catch the Azo-tagged species, and see how this compares with the already demonstrated experiment both in terms of purity of the isolated cystatin C, and its yield (if its binding is equally efficient under all aforementioned conditions).

Au: The theoretical possibility that the Strep-tag peptide sequence shows relevant affinity towards α -CD appears somewhat remote and, in fact, is already excluded with our experiment documented in the Suppl. Video, where mScarlet equipped with the Strep-tag II, but lacking the Azo-tag, was immediately eluted from the α -CD affinity column already during the washing step, without noticeable retention. Nevertheless, we admit that the recurring presence of the Strep-tag in the protein constructs with the Azo-tag may appear confusing and we have performed corresponding control experiments. To this end, we have expressed Azurin with the Azo-tag alone (see also the compilation of all Azo-tagged protein constructs that have been used in our study in Suppl. Table S1) and purified it both from the periplasmic extract and the whole cell lysate of *E. coli* under the same conditions as the previous protein version, which additionally carried the Strep-tag II. As result, the purity and yield were identical, whereas the same POI carrying the Strep-tag II alone did not show any binding to the α -CD affinity column (see the amended Suppl. Fig. S13), now also mentioned in the main manuscript (lines 441–443).

7) Another technical limitation, that needs to be determined prior to publication, is the efficiency of agarose penetration with the used UV-light source (355 nm). UV light, apart from being absorbed on many cellular components, can be also scattered by numerous materials. The proposed technology requires efficient penetration of the whole agarose deposit with 355 nm in order to detach the protein of interest. The authors have to measure in the quantitative manner the transmittance of 355 nm light through their material, e.g. with an actinometer and cuvettes (e.g. the typical UV-Vis quartz cuvettes) of increasing thickness filled with the used agarose beads in the same buffer, as demonstrated in the manuscript. The detector has to be isolated from the light source (so that there is no irradiation passing on the sides of the cuvette or the vial) and the amount of 355 nm light that can actually penetrate the beads without being scattered should be determined (some sort of calibration curve: transmittance of 355 nm light vs. the thickness of the agarose gel). You can probably use the classical UV-Vis spectrophotometer for that as well, providing that you measure the light transmittance. But it would be better to use your particular light source and a separate detector/actinometer, to get realistic assessment for your demonstrated setup. This will tell us, what the maximal thickness of the agarose column could be, in order to effectively apply the UV light-triggered release method in its whole volume.

Au: We do not quite understand the sceptical view here as our data clearly demonstrate (in particular, when looking at the Suppl. Video) that with the described experimental setup the light penetration of the chromatography matrix is sufficient to trigger the quantitative release of the Azo-tagged protein upon irradiation with LED UV light. Nevertheless, we have now invested some effort and examined this technical aspect more quantitatively. Of note, measuring transmission with a conventional UV-Vis spectrophotometer, as suggested, is not suitable due to the typically very small subtended central viewing angle of the radiance detector, which leads to extremely small measured values when investigating a strongly scattering medium. The reason is that after passage through the turbid chromatography matrix the incoming light beam has lost its focus and is scattered isotropically into all directions. As the "viewing angle" of ordinary UV-

Vis spectrophotometers used in a biochemical laboratory is not precisely documented, a geometric correction for this effect is not feasible. Hence, we devised our own setup that fulfills the relevant technical criteria mentioned. To this end, we used a commercially available Optical Power Meter to measure the transmitted light (through layers of increasing thickness in appropriate cuvettes) essentially independent of the angle. This setup and the resulting data are depicted in Suppl. Fig. S8, demonstrating that for a standard bench scale chromatography column (7 mm inner diameter) the light intensity at the center of our affinity matrix still amounts to approximately 50 % when illuminated from opposite sides.

8) one commonly known limitation of azobenzene applications in vivo is its susceptibility to intracellular reducing agents, in particular glutathione. There, the azo bond is reduced to the non-photochromic hydrazine. This property has been regularly monitored by the community, especially for photopharmacology applications. This might also be a serious concern for the Azo-tag installation protocol on the recombinant proteins. The standard model for in vitro testing the said susceptibility is incubation of the azobenzene of interest with 10 mM aqueous solution of reduced glutathione stabilized with 5 mM TCEP (using organic co-solvents like acetonitrile in case, if the azobenzene cannot be homogeneously dissolved in water). The authors have to incubate the azobenzene amino acid used for feeding their cells with the aforementioned mixture, and determined the lifetime of the Azo-compound under these conditions, using spectrophotometry (UV-Vis spectrum, follow the absorption maximum frequency of the azobenzene, and plot its decay over time).

Au: Unfortunately, we do not know which studies from the somewhat remote area of photopharmacology the Reviewer is referring to. However, in previous publications that reported the use of the same amino acid with a plain azobenzene side chain as in our application there was no mention of such a reduction phenomenon. Likewise, Reviewer #4, who seems to have hands-on experience with this non-canonical amino acid, did not raise this point. The proposed investigation of the UV-Vis spectrum of this compound is not feasible due to considerable spectroscopic overlap with common intracellular reducing agents. Hence, instead of looking at the spectral decay of the azo-amino acid substrate, we decided to investigate the function and structural integrity of the recombinant protein (POI) carrying the incorporated azobenzene side chain, as we believe that this is the more relevant experiment. To this end, we incubated the Azo-tagged POI mScarlet with 10 mM GSH, 5 mM TCEP or 10 mM NADPH over night and investigated if it can still bind to the alpha-CD affinity column, followed by light-triggered elution, without observing any deteriorating effect. This additional experiment has now been described in the new Suppl. Fig. S13B and also mentioned in the main manuscript (lines 578–580).

9) in the other direction - UV light may have negative influence on proteins, sometimes causing degradation. Please, provide experimental evidences that your proteins are resistant to the used wavelength of 355 nm at least over the time needed for isolation. Please prepare solutions of discussed proteins (mScarlett, Azurin, cystatin C) in regular buffers, expose the solutions to the irradiation setup used previously with your column, and take samples upon regular intervals - you can measure by HPLC-MS and gel electrophoresis, if the proteins' integration is kept, and for how long (I would suggest at least 60 minutes constant exposure, and sampling every 10 minutes).

Au: Normal proteins do not absorb UV light around 355 nm as they lack corresponding chromophores (as a reminder, the longest wavelength at which the aromatic Tyr and Trp side chains absorb light is around 280 nm). In so far, a hypothetical chemical damage induced by 355 nm light is not expected. Admittedly, the situation is different for colored proteins such as GFP, mScarlet and Azurin. To exclude photochemical decay in these cases we have performed the proposed experiment, which is now included as the new Suppl. Fig. S16. In essence, we could neither detect changes in the ESI mass spectrum or SDS-PAGE for human cystatin C nor alterations in the characteristic fluorescence spectrum of sfGFP after irradiation for up to 60 min

using similar light intensity as applied in the alpha-CD affinity chromatography, now also mentioned in the main manuscript (lines 521–531). For comparison, in our light-controlled purification experiments, elution of the Azo-tagged protein under UV light exposure is usually accomplished within a much shorter period of 15 min (in this context see also the new Suppl. Fig. S14C).

10) the authors claim quantitative switching of the (E)-azobenzene to the "excited" (?- maybe they mean thermally less stable) (Z)-isomer based on UV-Vis spectra - this is not a proof. It only shows that the switching occurs between two photostationary states, which may both contain various E/Z isomer ratio. The experience and literature data show that photoisomerization of azobenzenes without strongly donating or accepting substituents typically produces up to 80% of the Z-isomer in the photostationary state (although often in aqueous media the result of 30-40% may be observed).

The authors should necessarily repeat the irradiation experiments and isomer ratio determination with NMR (identify at least three well-separated peaks that belong to each photoisomer, and compare their averaged integrations between the E- and Z-isomer peaks) and with HPLC (find the isosbestic point, where the absorbance of E and Z isomer are equal, then measure HPLC chromatograms before and after irradiation - there, the comparison of peak integration for the old (E)-isomer and the newly formed (Z)-isomer will indicate the isomer ratio).

Au: First of all, we have corrected the nomenclature and no longer call the cis-isomer an "excited state" but a "higher-energy state". Second, we have now precisely quantified the cis/trans ratio obtained after irradiation at various wavelengths using ¹H-NMR spectroscopy. These elaborate experiments have been described in the new section Suppl. Results A as well as Suppl. Figs. S4 to S7, leading to meaningful insights which now have also been mentioned in the main manuscript (lines 616-619). Unexpectedly, the cis-state of the azo-amino acid in aqueous solution can almost be quantitatively populated (around 95 %) upon irradiation by 355 nm LED light. Conversely, only 70-80 % of the energetically favored trans-isomer is obtained in the photostationary state upon illumination at daylight, or with monochromatic blue LED light at around 430 nm, in line with published values. Nevertheless, while these data are certainly of academic interest, also in the light of the limited photochemical characterisation of p-(phenylazo)-L-phenylalanine so far, they do not change our experimental proof of concept for the light-controlled affinity chromatography. Still, knowledge of the wavelength-dependent photostationary composition may inspire changes in the way how the chromatography is carried out (such as performing the column washing step, after adsorption of the Azo-tagged POI, in the dark or under blue light illumination). Also in response to the questions by other Reviewers below, this has now been investigated and described in the new Suppl. Fig. S14. The results of our detailed NMR analyses are also consistent with UV-Vis spectroscopic measurements (see Suppl. Fig. S7) and should be convincing. Hence, we refrained from performing additional HPLC experiments, also considering the fact that transfer of the azo-amino acid from aqueous solution to an organic mobile phase (acetonitrile for RP-HPLC) is known to affect the cis-/trans-isomer ratio.

Reviewer #2 (Remarks to the Author):

Mayerhofer et al. present an interesting, and to the best of my knowledge novel, strategy for the chromatographic purification of proteins using a light. The system relies on the light-dependent host-guest interaction between azobenzene and cyclodextrin, which is well documented in literature (e.g. Nie et al. 2021). For the tagging of proteins of interest (POIs), the authors rely on the amber stop-codon/non-canonical amino acid (nc-AA) strategy to incorporate L-benzazophenylalanine (Baf; also known as AzoPhe in the literature), which is incorporated into a short so-called Azo-tag (2-4 amino acids) attached to the N- or C-terminus of the protein during

heterologous production in a specifically adapted *E. coli* strain. As shown by the authors for the isolated nc-AA, Baf can be switched between trans- and cis-states by illumination with 355 nm UV-A light, with the trans-form strongly interacting with α -cyclodextrin (KD approx. 91 μ M), with no detectable affinity between α -cyclodextrin and the cis-form of the nc-AA. This essentially allows the use of column materials with covalently attached α -cyclodextrin, which were synthesized by the authors. For the incorporation of Baf into the Azo-tag of a given POI, a one-plasmid system was developed carrying the Azo-tagged POI as well as the needed heterologous tRNA and the cognate aminoacyl-tRNA synthase. The authors went to great lengths to optimize both the Azo-tag sequence, the used aminoacyl-tRNA synthase as well as the expression host strain for optimal incorporation of Baf and selective and contaminant free elution by illumination. The Azo-tagged POI (with Baf in the trans-form) can be bound to the column in the dark (or dim daylight), unspecifically bound proteins are removed by washing with the same buffer and elution is carried out under buffer flow and lateral illumination of the column by UV-A light using LEDs. Proof of concept of the strategy was shown for two model proteins mScarlet (a fluorescent reporter) and Azurin (a blue, copper-binding protein of *Pseudomonas aeruginosa*). Purification was demonstrated from cytoplasmic crude cell extracts (mScarlet) and periplasmic extracts (Azurin targeted to the periplasm). In addition to these experiments with model proteins, applicability was verified for high-throughput, microscale purification of more realistic POIs: i) two enzymes: the monomeric beta-lactamase AmpC and the homo-dimeric glutathione S-transferase GST, ii) antibody fragments (single-chain variable fragment of a monoclonal antibody as well as a nanobody produced in *E. coli*). Finally, using an azo-tagged adaptor protein that binds to full-size antibodies, trastuzumab, a monoclonal antibody used to treat breast and gastric cancer, was isolated from mammalian cell culture medium.

The presented work is certainly novel and original, the results sound and the manuscript in most parts clearly written and illustrated (for some exceptions see below). If indeed as widely applicable as the authors propose, the strategy could be of general interest to a wide range of researchers and thus deserves publication in *Nature Communications*. However, the manuscript in its current form leaves several potential issues unaddressed, which would be essential for wide-spread application. However, if the authors can address these issues, I would wholeheartedly recommend publication.

Au: Thanks for the perfect summary and generally positive assessment of our results. Besides, for completion we have now also quoted the review article by Nie et al., 2021 (Ref. 26 in the revised main manuscript) as part of the background literature (even though we are not convinced that this particular reference is crucial).

Major comments

1. An important issue for any protein purification strategy is the achievable yields (which depend on the POI but also on the binding capacity of the column) and purities (that depend on the selectivity of the column-POI interaction). The authors have sufficiently demonstrated that their strategy can yield pure protein in a single step, but they leave other important questions unanswered, such as e.g. the dynamic binding capacity of their column material. How much protein can be bound to e.g. 1 mL of resin at a given flow rate before 10% of the target protein accumulates in the flow-through fraction ("breakthrough"). Similarly, what about reusability and chemical stability of the material? Please note that I am not advocating for a complete characterization of the column material which would rather be needed if commercialization of the strategy is attempted. The authors should however provide some basic characteristics to address dynamic binding capacity, reusability and chemical stability with regard to common components used for protein stabilization (e.g. DTT, TCEP). Likewise, it would be interesting if purification under denaturing conditions (e.g. in the presence of Urea) is possible.

For a good overview see: For a good overview see e.g.

<https://www.thermofisher.com/de/de/home/life-science/protein-biology/protein-biology-learning-center/protein-biology-resource-library/protein-biology-application-notes/performance-characterization-hispur-nickel-superflow-agarose.html>

Au: We agree that some of these aspects could be relevant with regard to broader use of the light-controlled alpha-CD affinity chromatography. Therefore, both the dynamic binding capacity has been determined for our standard column and reusability as well as chemical stability of the affinity matrix have been tested and described in the new Suppl. Fig. S15. In brief, the binding capacity was approximately 12 mg per mL bed volume for the Azo-tagged Azurin protein while the affinity matrix was shown to be fully stable against DTT, TCEP, GdnHCl, urea as well as NaOH (which we had used before for regeneration/sanitizing purposes). Furthermore, 10 consecutive chromatography runs did not indicate any deterioration of the binding capacity towards an Azo-tagged POI. Of note, all the experiments described in our manuscript were carried out with just two specimen of the alpha-CD affinity column, which further proves its robust performance. Besides, both Sepharose and cyclodextrin are materials used for bulk industrial applications without known sensitivities against conventional biochemical reagents. On the other hand, the Reviewer should keep in mind that a particular advantage of our light-controlled affinity chromatography is the protein elution under native conditions, using a mild or physiological buffer. Methods for affinity purification under denaturing conditions are long established in the field, starting with the initial application of the His-tag in the presence of GdnHCl (see, e.g., Ref. 4). To address the Reviewer's last question, we have tried to subject the Azo-tagged Nanobody EgA1 (Nb-StrepII-GG-Pap, see Suppl. Table S1) to alpha-CD affinity chromatography in the presence of 6 M urea both in the sample and the chromatography buffer and, as a result, the Azo-tagged POI appeared in the flow through. However, this is not surprising in the light of the well known solubilizing effect of concentrated urea solutions on hydrophobic groups (see PMID: 19123816), thus competing with the formation of a stable host-guest complex with alpha-CD.

2. Another important issue to consider when using UV-A light for protein elution is whether the light will cause oxidative modifications of the POI, which in turn could affect function. UV-A light is known to cause oxidative damage to proteins either directly or via the generation of reactive oxygen species such as singlet oxygen. See e.g. Hawkins et al 2019 for an overview. Some efforts should be made to rule out that the light-dependent elution scheme used by the authors does not result in oxidative damage to the POI.

Au: This topic was also brought up by Reviewer #1 above. Of note, normal proteins do not absorb UV light around 355 nm as they lack corresponding chromophores. In so far, chemical damage induced by 355 nm light is not expected, even though the situation might be different for colored proteins such as GFP, mScarlet and Azurin. Furthermore, the generation of reactive oxygen species (ROS) in oxygen-containing aqueous solution with such mild UV light is not an issue unless photosensitizers (certain organic dyes or redox-active proteins) or organelles responsible for aerobic respiration or photosynthesis (chloroplasts, mitochondria, among others) are present. The latter is addressed by the review article by Hawkins et al. (2019) mentioned, which mainly discusses well known protein modifications that occur within biological systems. In contrast, proteins in cell-free solution are investigated in our study. If ROS generation for such a kind of clarified extracts were a relevant problem in general, one should never purify a protein on a liquid chromatography station equipped with a UV flow-through detector operating at the much harder wavelength of 280 nm, as it is common practice in biochemistry and biotechnology. Nevertheless, to exclude the possibility of photochemical decay during elution from the affinity column with our system, we have performed extended illumination experiments at 355 nm, now described in the Suppl. Fig. S16. In essence, we could neither detect additional molecular species in the ESI mass spectrum or changes in the SDS-PAGE band pattern for human cystatin C nor alterations in the characteristic fluorescence spectrum of sfGFP after irradiation for up to 60 min using similar light intensity as applied in the alpha-CD affinity chromatography.

For comparison, in our light-controlled purification experiments, elution of the Azo-tagged protein is usually accomplished at a much shorter time scale of 15 min.

3. Along the same lines, how much light is needed to trigger elution from the column? Can this be quantified?

Au: Again, this topic was addressed by Reviewer #1, too. First, the combined light intensity of in total 16 LEDs with peak maximum at 355 nm arranged around the alpha-CD affinity column was 20–40 mW (as explained in Suppl. Method A, see also Suppl. Fig. S1). Besides, this is by orders of magnitude less power than the LED UV light source applied in the studies quoted by Reviewer #4, with 2900 mW at 365 nm (Hiefinger et al., 2023). Considering also the suggestions by Reviewers #1 and #4, we have now applied initial light pulses of 1 or 3 min to trigger the trans-to-cis isomerisation of the Azo-tag, thus effecting elution (see the new Suppl. Fig. S14). Alternatively, we applied intermittent illumination with the set of 355 nm LEDs (using a 0.1 s pulse per 1 s, corresponding to 10 % light intensity on time average), indicating that the amount of light (or its energy quantity = power x time) needed to achieve quantitative elution of the azo-tagged POI from the alpha-CD affinity column is modest in practice. In addition, as suggested by Reviewer #1, we measured the effective light intensity that emerges after the LED light has passed a layer with defined thickness of the settled chromatography matrix in our preferred chromatography buffer (as light scattering depends on the differences in the refractive indices), which has now been described in the new Suppl. Fig. S8.

Minor comments:

The strategy presented by the authors is hardly the first that allows purification of proteins by light. The authors should cite and discuss other strategies for light dependent protein purification. What are advantages disadvantages of their method relative to previously presented studies? Examples include the mem-iLID strategy (Tang et al 2021), light-switchable nanobodies (Gil et al. 2020) and the use of Phytochrome-PIF6 interaction for affinity-based POI purification (Hörner et al. 2019).

Au: After reading the papers mentioned, it appears that only those by Tang et al. (2021) and Hörner et al. (2019) actually deal with some kind of bioseparation mediated by light-dependent adsorption (either to functionalized biological membranes or to functionalized bead suspensions), even though not demonstrating protein purification by affinity chromatography. For completeness, we have now discussed these publications in the amended Conclusions section (see lines 630–647).

Line 109: “Quantitative switch to the excited state, i.e. the cis-configuration ...” As far as I know, the cis-configuration of the azobenzene is not an electronic excited state. Both the trans and the cis-state are electronic ground states, while isomerization proceeds via excited state formation.

Au: To avoid confusion, we have now used the term "higher-energy state" instead, as also suggested by Reviewer #1.

Figure 1, F: Kinetic stability of the cis-state: Kinetics have not been recorded sufficiently long so that an exponential fit would yield meaningful values. The data shown in the Figure could be equally well fitted with a linear function. Measurements would need to be conducted much longer, e.g. three times the lifetime of the exponential decay. I am aware, that this might experimentally not be feasible here. So either, carry out longer measurements mention this caveat.

Au: It is physically plausible and has been experimentally verified in many instances (see, e.g., Ref. 34 in the revised Supplement) that the thermal reversion of the higher-energy cis-state of azobenzene (and its derivatives) to the trans-state follows a simple 1st order kinetics. As we know the asymptotic end value, which is the absorbance prior to induction of the trans-to-cis isomerisation with 355 nm UV light (see Fig. 1E), within the same solution and cuvette, we

believe that an exponential curve fit as presented is justified. This is further supported by the full reversibility of this process as we have proven in our experiments. We have now amended the relevant description in Suppl. Results A (see lines 660–663 in the revised Suppl. Information).

In Figure 3: Please schematically illustrate the strategy for the adaptor-protein based purification of trastuzumab.

Au: This strategy has now been schematically illustrated in Fig. 3H.

Line 219: “Initially, we utilized superfolder GFP (sfGFP) in order to select E. coli cells carrying a mutated PylRS for incorporation of Baf at its structurally permissible sequence position 39 via fluorescence-activated cell sorting (FACS)” It is unclear to me, if this sentence describes work that was carried out as part of this study (then some Figure should be referenced) or if the authors refer to some of their previous work. Please clarify!

Au: This work is part of the present study and was actually described in greater detail two paras. further below (as well as Fig. 2D, E). For clarification, we have now also inserted a reference to Suppl. Results B (see lines 249–250 in the revised red-line manuscript version).

Line 278: “which also made the use of beta-CD for solubilizing Baf less important in our hands” This doesn’t say clearly if the authors used this strategy for the next experiments or not. Please clarify!

Au: While we experimentally observed that beta-CD can actually be omitted as a solubilization enhancer at these lower concentrations of the azo-amino acid in the culture medium, we routinely applied this supplement throughout the experiments shown (see Suppl. Method J). For clarification we have now added "(data not shown)" to line 319.

References

C. Nie, C. Liu, S. Sun, S. Wu, Visible-Light-Controlled Azobenzene-Cyclodextrin Host-Guest Interactions for Biomedical Applications and Surface Functionalization. 2021. ChemPhotoChem 5: 893.

Hawkins CL, Davies MJ. Detection, identification, and quantification of oxidative protein modifications. J Biol Chem. 2019. 294:19683-19708.

Gil AA, Carrasco-López C, Zhu L, Zhao EM, Ravindran PT, Wilson MZ, Goglia AG, Avalos JL, Toettcher JE. Optogenetic control of protein binding using light-switchable nanobodies. Nat Commun. 2020. 11: 4044

Hörner M, Eble J, Yousefi OS, Schwarz J, Warscheid B, Weber W, Schamel WWA. Light-Controlled Affinity Purification of Protein Complexes Exemplified by the Resting ZAP70 Interactome. Front Immunol. 2019. 10: 226.

Tang R, Yang S, Nagel G, Gao S. mem-iLID, a fast and economic protein purification method. Biosci Rep. 2021. 41:BSR20210800

Reviewer #3 (Remarks to the Author):

The authors presented a light-switchable affinity tag, Baf, that binds to CD with high affinity (91 uM) in the trans-form (ground state) and low affinity in the cis form (excited state). The dissociate constant is comparable to Strep-tag (13 uM), developed by the same lab, and enables protein purification, with a linked tag, under the stimulation of mild UV light. Compared to commonly used affinity chromatography, which requires additional incubation of elution buffer, this “excitography” uses light to trigger the dissociation of Baf-labeled POI and CD-containing matrix.

The data presented in this work is clear and organized. One of the major concerns, however, is the requirement for non-natural amino acids and tRNA-synthase (e.g., PylRS) to be incorporated into the target protein. As demonstrated by the author, PylRS needs to be optimized, special strains of bacterial expression system need to be used, and the location of Baf infusion and even the neighboring amino acids around Baf should be optimized. A general question is how these optimized conditions apply to other proteins not demonstrated in this work and if there are any limitations for the POI in terms of size, pKa, and other physicochemical properties.

Au: Thanks for the overall positive assessment. To make it clear, the optimization work has been fully accomplished and there is no need to repeat it. Our work resulted in an engineered PylRS that enables suppression of the amber stop codon in general (essentially regardless of the sequence context) and leads to the efficient cotranslational incorporation of the azo-amino acid, as has been demonstrated for various different representative POIs. Likewise, the optimization of the Azo-tag has been completed in our study, with sterically non-demanding Gly residues on both sides and a preferred GGC or GGA codon preceding the amber codon (see the new Suppl. Fig. S25). As we have demonstrated in our study, the Azo-tag works nicely at the N-terminus or the C-terminus, or both, for 12 different POIs, and even more distinct protein constructs (now summarized in the Suppl. Table S1), plus for a similar number of other proteins that have been tested in our laboratory during the last months. In particular, we see no limitations with regard to the biochemical protein properties mentioned by the Reviewer. To facilitate the wider use of "Excitography" (also in response to a similar question by Reviewer #1), we have now included the new Suppl. Fig. S25 depicting a scheme which in detail describes the methodology how to clone the coding region of a POI on our optimized expression vector in a single step, optionally using different arrangements of the Azo-tag and employing cytoplasmic expression or periplasmic secretion. In addition, our expression vector and the genetically modified host strain will be made generally accessible. The vector will be submitted to AddGene for general distribution after acceptance of our manuscript. Likewise, we will submit our expression strain, NEBExpress(lowRF1), to a public repository. As an alternative, researchers may use the pEVOL double vector system encoding the previously published AzoPheRS (see Suppl. Ref. 28) for incorporation of the Azo-amino acid, even though with lower efficiency (see the amended Suppl. Results C), and the E. coli strains C321.ΔA or B-95.ΔA, which were extensively engineered to lack most of the amber stop codons as well as RF-1 (see PMID: 24136966 and PMID: 25982672, respectively).

It seems that Baf is not commercially available and must be synthesized, impeding its wide applicability to labs without experience or equipment necessary to carry out the required chemistry. Please include a brief discussion in the text about its availability (or lack thereof).

Au: In fact, these concerns are unjustified as the azo-amino acid has been made commercially available from several vendors recently, for example from Sigma-Aldrich/Merck under the compound name "phenylalanine-4'-azobenzene HCl" (Order no.: 914584) and from Iris Biotech (Marktredwitz, Germany) under the compound name "H-L-Phe(4-N=NPh)-OH*HCl" (Order no.: HAA3710); in this context, the Reviewer may also have a look at their topical newsletter: <https://iris-biotech.de/global/blog/azobenzene-photoswitches>

Some technical questions:

What is the purpose of including the Strep-tag in the fusion POI? Is this simply a convenience for cloning or is it somehow required for Baf to work?

Au: Indeed, the Strep-tag was used by us for convenience, especially during the early stage of this project, where we prepurified some of the Azo-tagged proteins before studying their chromatographic behaviour on an alpha-CD affinity column. However, the Strep-tag has no

functional relevance in the context of the light-responsive Azo-tag. To demonstrate this (also in response to a similar question by Reviewer #1), we have expressed Azurin with the Azo-tag alone (see Azurin-GG-Pap in Suppl. Table S1) and purified it both from the periplasmic extract and the whole cell lysate of *E. coli* under the same conditions as the previous protein version which also carried the Strep-tag II. As a result, the purity and yield were identical, whereas the same POI carrying the Strep-tag II alone did not show any binding to the alpha-CD affinity column (see the new Suppl. Fig. S13A).

Minor concern:

It would be nice to comment on the different trends of A326 when titrating trans-Baf with α - (Fig. 1I) or β -CD (Fig. 1K).

Au: Complex formation between the trans-form of the azo-amino acid and both alpha-CD and beta-CD leads to a red shift of the band around 326 nm, in line with measurements for other azobenzene derivatives reported in the literature, thus indicating a less polar environment of the azo-dye in the complex (see Ref. 58 and Suppl. Ref. 33 in our revised manuscript). The only difference in our titration experiments was that this was accompanied by a small decreasing amplitude in the case of alpha-CD but a slightly increasing amplitude for beta-CD, as now shown in the new panel D of Suppl. Fig. S3. Similar deviations have been documented for other azobenzene derivatives when forming complexes with different cyclodextrins (cf. Suppl. Ref. 33). This is likely explained by distinct intermolecular interactions within the host-guest complex, in line with the fact that beta-CD can also accommodate the more bulky cis-isomer whereas alpha-CD cannot. We have added a corresponding remark and explanation at the end of the legends to Fig. 1 and Suppl. Fig. S3, respectively.

Fig. 2G. The initial distribution of mScarlet-Streptag-Baf seems to be much broader than Aruzin-Streptag-Baf. In this case, what would be the effect of POI size on the effect of binding between trans-Baf and CD?

Au: The binding to the affinity matrix is less dependent on the protein size – see, for example, the efficient purification of a full-size antibody in complex with the Azo-tagged protein G shown in Fig. 3H – however, there seems to be some context-dependence regarding the surface properties of the POI. This is a common phenomenon for affinity tags and has also been seen for the His-tag and the Strep-tag in many instances. For example, sfGFP, which has almost exactly the same size as mScarlet, shows a more narrow elution (see Suppl. Fig. 18F). Likewise, the enzyme AmpC showed a rather sharp elution, as indicated by SDS-PAGE analysis as this protein is not colored. We have now briefly mentioned this aspect in the amended Conclusions section (see lines 665–672 in the revised red-line version), also referring to the new Suppl. Table S1 where we have summarized all the different Azo-tagged POIs investigated in our study.

There is no mention in the text about the availability of any of the plasmid variants required to produce recombinant proteins that incorporate Baf. If plasmids are not deposited in Addgene or another suitable distributor to make these materials available, please ensure that readers can access or obtain the full sequences of the most relevant/effective plasmid backbone that contains the machinery to incorporate Baf, for example the one incorporating the sfGFP construct. This will help other investigators to apply this technique to their own systems.

Au: We will make our expression system publicly available, as already explained in our response to this Reviewer above; see also the new Suppl. Fig. S25 with a description of the cloning procedure.

On lines 106-7 of the supplementary methods, the text reads “purified PCR fragments were mixed in approximately equimolar amounts together with T4 DNA polymerase (Thermo Fisher Scientific) in the buffer supplied by the manufacturer” – this may be a typo referring to T4 DNA Ligase?

Au: As mentioned a few lines further up within the same para. of the Supplement referred to we used the one-step SLIC method (see Suppl. Ref. 2) in order to assemble the vector pSB19. With this technique, efficient and directional cloning is achieved by direct bacterial transformation after mixing a linearized vector with insert(s) prepared by PCR and T4 DNA polymerase, which yields hybridizing single-stranded overhangs that are filled-in and ligated intracellularly.

Reviewer #4 (Remarks to the Author):

Key results

The manuscript entitled "Protein purification with light" presents a strategy to purify protein samples using a light-sensitive tag. The tag comprises the azobenzene based unnatural amino acid known as AzoF (here named Baf), which allows the binding of the protein in the dark or under day-light to an α -cyclodextrin resin. Elution of the protein of interest is then facilitated by irradiation with UV light. The presented research appears to be original and innovative, and a lot of work and thought has been put into the success of this strategy from a biochemical point of view. I am particularly convinced that this method will have an impact on diverse fields of protein chemistry due to the extremely high binding efficiency and purity of the eluted samples. I therefore very much like to see this manuscript published. However, I have two major concerns regarding the publication in a high-impact journal such as Nature Communications, which should be addressed appropriately.

Au: We are glad that this Reviewer appreciates the main message of our work.

Key concerns

First, I have to point out that its photochemical/physical behavior is misrepresented (as I will explain in more detail below). As a result thereof, data were misinterpreted and other data that are crucial to show the reproducibility of this technique are missing. Second, although they nicely emphasized on the benefits of this strategy and minimized some limitations of it, the authors missed to discuss the (remaining) major limitations of using an unnatural amino acid in a protein tag. I have provided detailed arguments below sorted into major and minor points. These illustrate how the manuscript should be revised to entirely cover the state-of-the-art and how it should be complemented with further experiments to further analyze the principle and improve the reproducibility of this purification strategy (which in my opinion might also be necessary for the pending patent).

Au: We believe that this is an overly critical statement. While some constructive suggestions are acceptable, other arguments appear unjustified, as will be addressed point by point in our detailed rebuttal below.

Major points

Misrepresentation of the photochemical/physical behavior of azobenzenes [throughout the manuscript (particularly p. 3)].

- The trans and cis configurations of azobenzenes (including AzoF) do not exhibit distinct absorption bands that allow their individual spectroscopic detection. The two signals in the UV and visible range originate from the excitation of the $\pi \rightarrow \pi^*$ and $n \rightarrow \pi^*$ transition of the azobond, respectively. The azobond is the central moiety in trans and cis. If you follow the isomerization over time you will not follow the appearance of a second peak for cis and the decline of the first peak for trans as for chemical reactions. Instead both peaks change their intensity and maximum smoothly with the isomerization.

- Azobenzenes (including AzoF) cannot switch quantitatively because of their overlapping spectra. As a result, a photo-induced equilibrium named photostationary state (PSS) is established. Each irradiation step switches between a trans-enriched and cis-enriched PSS. This means that during irradiation with any wavelength that falls in the range of AzoF absorption, AzoF will switch back and forth constantly keeping up the equilibrium. The intensity and wavelength of light thereby defines the rate of this back-and-forth switching.
- The composition between trans and cis in the PSS does not only depend on the extinction coefficients ϵ but also the quantum yield Φ . Importantly, both factors are dependent on the wavelength in azobenzenes. Since the quantum efficiency is often difficult to measure, the best wavelength for the photoswitch is usually determined by trying different LEDs with different wavelength maxima.
- Moreover, both ϵ and Φ are extremely dependent on the environment, such as buffer, ions, acids/bases. This means AzoF will exhibit a different trans/cis composition at the PSS and a different rate of isomerization when incorporated into proteins compared to its isolated form (and between different proteins).
- Due to the overlap of the trans and cis spectra, spectral evaluation only allows for an estimation of the trans/cis composition at a defined state.
- Finally, owing to the wavelength dependence of ϵ and Φ , different LEDs result in different isomerization rates and trans/cis compositions at the PSS. This can even be the case for two LEDs with the same peak wavelength. The reason lies within the different width of the Gaussian wavelength spectrum of the LEDs. Hence, using different LEDs can have a major impact on the reproducibility of any experiment with photoswitches.

⇒ In addition to my explanations, I suggest to go back to literature and research the photochemistry of azobenzenes. Works of Viktor Szymanski, Ben Feringa, Dirk Trauner, Alex Deiters, and Andrea Hupfeld (née Kneutinger) will help.

⇒ Subsequently, I find it important to update the state-of-the-art on azobenzenes in the introduction (currently missing) and the results section.

Au: We are grateful to the Reviewer for her/his comprehensive summary of the photophysics/chemistry of azobenzene. We agree that this is common state of the art (see, e.g. the excellent and comprehensive review by Bandara & Burdette, 2012, which we had quoted as ref. no. 18 on p. 3, now Ref. 19 in our revised manuscript) and this was nowhere put into question by us. Most importantly, the partially overlapping spectra of the trans- and cis-isomers were explicitly documented in Fig. 1D (now amended with a more detailed analysis to illustrate the influence of the irradiation wavelength on the degree of isomerization as shown in Suppl. Fig. S7) and correctly interpreted in the text of our original manuscript. In so far we cannot see any "Misrepresentation".

Furthermore, in our brief discussion on p. 3 we had actually referred to the dynamic light-dependent equilibrium between cis- and trans-azobenzene which she/he explains so elegantly here. Admittedly, the photodynamic aspects may inspire some deeper investigation; therefore, also motivated by the comments of Reviewer #2, we compared both the retention and the elution of an Azo-tagged protein under constant or pulsed illumination and/or in the dark. As expected, while the light-induced affinity purification in principle worked under all these conditions, there were noticeable differences in the retention/elution behavior, which could be contributed to a frozen ratio between cis/trans states in the dark or a dynamic reequilibration under illumination (approaching the PSS). These experiments are now described in detail in the Suppl. Fig. S14.

Of note, the careful choice of LEDs with suitable wavelengths was part of our arduous technical optimization process and, indeed, a handheld spectrophotometer which measures both peak wavelength and distribution played a crucial role in this context (see Suppl. Method A). In the

course of this technical assessment, we also found that the emission spectrum of commercially available LEDs can even be asymmetric (and not Gaussian type, as assumed by the Reviewer); however, those types listed in our methods description worked reliably throughout all our experiments.

The Reviewer should furthermore understand that our manuscript is not a review article about the photophysics of azo-benzene derivatives (or their use in photopharmacology) but an experimental study on an application for affinity purification in the field of protein biochemistry. In those instances where our use of the terminology may not have been sufficiently precise, we have now corrected this, also in response to the more specific suggestions by Reviewers #1–3 (see the red-line corrections in our revised manuscript).

Misinterpretation of data.

p.3, l. 99-142: “However, the spectroscopic properties of Baf have been only partially characterized up to now ... Therefore, we prepared Baf following the published procedure and investigated its absorption spectra as well as its isomerization between the trans- and cis-state in greater detail...”

- Owing to the sensitivity of ϵ and Φ on the environment, the trans/cis composition, isomerization rate and thermal stability of isolated AzoF is only limitedly representative to AzoF incorporated into proteins

⇒ Please provide spectral analyses for at least two of the chosen proteins with incorporated AzoF to compare with isolated AzoF.

Au: Such spectra are now depicted for the uncolored enzyme GST (Suppl. Fig. S21A) and the nanobody EgA1 (Suppl. Fig. S22B). As foreseeable, there were no surprises since the spectral contributions by the natural aromatic side chains of the protein and the single azobenzene group are practically additive. However, we have taken this opportunity to report the incremental contribution of the azo-amino acid to the absorbance at 280 nm (see Suppl Method L), which is helpful to correctly determine the concentration of an Azo-tagged POI by UV spectroscopy (as it is common practice in protein biochemistry today).

p. 3, l. 109-111: “Quantitative switch to the excited state, i.e. the cis-configuration, was easily achieved by illuminating the solution in a quartz cuvette from the top (Suppl. Fig. S2) with mild UV light at 355 nm using a 1.2–2.4 mW LED for =30 min (Fig. 1E)”

- The cis configuration is not the excited state, it is instead the product of excitation (to S2) and return to the ground state via isomerization. Excited azobenzenes also fall back to the ground state without isomerization due to their low quantum yield.

⇒ Please correct this statement.

Au: Thanks for spotting this. Of course, we agree that the cis-configuration does not represent an excited state in the physicochemical sense. We have now used the term "higher-energy state" instead.

- As explained above, azobenzenes cannot switch quantitatively. Instead, irradiation results in a trans- or cis-enriched PSS.

⇒ Determine the composition of trans and cis of the state shown in Fig.1D(red), e.g. by using spectral analysis (<https://www.biorxiv.org/content/10.1101/2024.07.04.602025v1>, <https://doi.org/10.1016/bs.mie.2022.12.003>, <https://chemistry-europe.onlinelibrary.wiley.com/doi/10.1002/chem.202004061>)

Au: We have now used a more precise wording to account for the incomplete light-induced isomerisation and included additional experiments to quantify the proportion of cis/trans isomers

of the azo-amino acid under different conditions (see Suppl. Results A). Motivated by the comment of Reviewer #1 we have accurately measured the cis/trans ratio obtained after irradiation at various wavelengths using $^1\text{H-NMR}$ spectroscopy (see Suppl. Figs. S4 to S6) and correlated this with a UV-Vis spectral analysis (Suppl. Fig. S7). The key results were that (i) the cis-state is populated to a rather high proportion of 95 % after irradiation with LED UV light at 355 nm whereas (ii) the lower-energy trans-state is only populated at approximately 75 % under daylight conditions. In contrast, previous publications on azobenzene derivatives suggested a more quantitative trans-isomerisation under daylight but just partial cis-isomerisation during UV illumination. Only when using enforced thermal equilibration in the dark, after transient acidification as described by Rickhoff et al., 2022 (Suppl. Ref. 34), an almost complete trans-state of the azo-amino acid was obtained. This allowed us to present the UV-Vis spectra of essentially pure (around 95 %) cis- and trans-isomers, as now shown in the new Suppl. Fig. S7A. In addition, upon calibration against the quantitative $^1\text{H-NMR}$ data, we were also able to determine precise extinction coefficients for the individual cis- and trans-isomers at the relevant wavelengths of 326 and 426 nm (Suppl. Fig. S7B).

We believe that this NMR-based quantification is more accurate than the kind of "spectral analysis" presented in the Methods in Enzymology chapter by Hiefinger et al. (2023) (the second reference quoted above) which, in order to determine the cis/trans equilibrium in the PSS, proposes a "simulation of the 100% Z-isomer spectrum" under the assumption that the "absorbance value of the $\pi\text{-}\pi^*$ transition signal at 330 nm approximates zero". However, our data clearly show that, while the band maximum shifts from 326 nm to 293 nm upon isomerisation of the azo-amino acid from the trans- to the cis-state, there is still a detectable shoulder at 330 nm, which cannot be neglected (see Suppl. Fig. S7).

- Isomerization and trans/cis composition at the PSS depend on the intensity and wavelength of irradiation. Hence, it is more important for the reproducibility to provide the actual intensity of light that was used for irradiation and the wavelength spectrum of the LED than the technical specification.

⇒ Please provide the intensity of irradiation (to be exact measure with a sensor in the same distance to the LED as the sample; or at least provide the voltage and current that were applied) and the wavelength spectrum of the used LED.

Au: These technical data were fully disclosed in the Suppl. Method A (see lines 28–34 and also Suppl. Fig. S1). Detailed spectral characteristics of the LEDs used are freely available from the vendor as part of the technical documentation, see e.g.: http://www.roithner-laser.com/datasheets/led_div/xsl-355-3e-r6.pdf

p.3, l. 111-124: "The resulting solution of cis-Baf ... Baf shows distinct absorption bands that not only allow the individual spectroscopic detection of its trans- and cis states, ... However, to optimize the equilibrium in the photostationary state..., we chose a wavelength of 355 nm for efficiently switching trans→cis, corresponding to a high ratio $\epsilon_{\text{trans}}/\epsilon_{\text{cis}} = 8.3$ (Suppl. Fig. S3)"

- As explained the trans/cis composition depends not only on ϵ but also Φ making the analysis in Suppl. Fig. S3 redundant.

⇒ Either find a different logical reason for choosing 355 nm or provide a wavelength screen, in which the trans/cis composition at the PSS for different wavelengths is shown.

Au: We have now mentioned the role of the quantum yield as an additional factor (see line 140 in the revised manuscript). One reason that we chose the wavelength of 355 nm was that we experimentally observed a more complete trans-to-cis switch than with LED UV light at 365 nm, the wavelength previously used in other published studies (including the work by Hiefinger et al., 2023). This is now also documented with our quantitative $^1\text{H-NMR}$ measurements, indicating 94.3 % cis at 255 nm but only 86.5 % cis at 365 nm (see Suppl. Figs. S6 and S7). Unfortunately,

as we had mentioned briefly in our manuscript, the wavelength range of interest is only partially covered by commercially available LEDs. In fact, such LEDs are not accessible in the area below 355 nm, which prevents a broader wavelength screen in this spectral region.

p.4, l.185: "When a 5 mM solution of Baf was applied under daylight (i.e. in its trans-state) ..."

- Under daylight AzoF is also isomerizing. We have to cover all our AzoF samples since daylight or even our lab light is able to switch AzoF. The thereby established PSS might not be cis-enriched, however back- and forth isomerization is still taking place during light exposure.

⇒ Please correct this statement.

Au: We have now used the term "predominant" trans-state (see line 212). Of note, lab light is spectrally different from (dim) sun light (or "daylight"), especially when using energy-saving neon tubes or even LED-based lights. Furthermore, motivated by the suggestions of Reviewer #2, and as mentioned above, we have compared both the retention and the elution of an Azo-tagged protein under constant or pulsed illumination and/or in the dark (see lines 364–371 and Suppl. Fig. S14).

p.5, l.193-195: "Measurement of the absorption spectrum of the eluted fraction revealed the presence of Baf in its cis-state by showing only absorptions at 293 nm and 426 nm and no peak at 326 nm."

- The results are not provided.

⇒ Please provide the respective UV/Vis spectra and determine the trans/cis composition.

Au: See Suppl. Fig. S9B. There is no detectable trans-isomer immediately after the UV-triggered elution.

Consequences on the principle and reproducibility of the strategy.

p.2, l.87: "... (Scheme 1)."

- The description of the principle in the main text is quite brief. For example, the new name "Excitography" is missing.
- The authors only roughly describe the principle of their new chromatography. What is implied to the reader is that the proteins are loaded in a 100% (fixed) trans state of AzoF with high affinity to the resin. After washing (still 100% trans), the column is irradiated so that AzoF isomerizes to 100% (fixed) cis. Concomitantly, the protein loses its affinity and can be eluted concomitantly.
- As I have explained, AzoF cannot be switched quantitatively to 100% cis and exposure to daylight can lead to its isomerization resulting in a trans/cis composition of <100% trans.
- Following the implied principle, AzoF could neither be fully loaded nor fully eluted.
- However, the provided results indicate that AzoF-labeled proteins can at least be fully eluted (even though this has only been shown qualitatively with pictures of the columns). In my opinion, this can be explained by the methodical realization. Fig. 2I and Suppl. Fig. S8F show that the column was simultaneously irradiated during elution. Hence, AzoF was continuously switching between trans and cis during the elution step allowing the full elution of the AzoF-labelled protein. It appears to me that excitography works by regulating the retention strength with light. In the dark, only trans AzoF will bind to the resin, whereas cis AzoF will flow through. If the column is exposed to weak daylight, trans/cis switching of AzoF is initiated, though with a probably trans-enriched PSS and with probably slow isomerization rates. This results in high retention and probably slow elution. High intensity UV light instead

creates a cis-enriched PSS with high isomerization rates. This results in low retention and faster elution.

⇒ Please reconsider the principle of excitography.

⇒ In case of agreement, please explain the principle in greater detail in the main text, revise Scheme 1 and adjust the manuscript, e.g. the conclusion, in which the authors already point out some discrepancy with their hypothesis.

Au: Unfortunately, we do not quite get the point here. Our Scheme 1 was specifically included for the purpose to explain the principle of the light-controlled affinity chromatography in a graphical sketch, together with a comprehensible explanatory legend, which also introduced the term Excitography (see first line of the legend). And obviously, the Reviewer has grasped the principle of our technique (like the other Reviewers). Thus, we do not see a reason to repeat the description in the main text, also in the interest of scientific conciseness. However, we are happy to introduce the term "Excitography" now already in the Abstract (see line 30) and we have inserted a couple of additional references to Scheme 1 throughout the text.

On the other hand, there seems to be some misconception regarding a putative exclusive role of cis/trans-photoisomerization, and of its dynamics, for the chromatographic behavior (i.e., retention versus elution) of an Azo-tagged POI. We concur that in practice it is difficult to achieve a 100 % trans-state, whereas with our optimal illumination wavelength of 355 nm we are getting close to quantitative cis-isomerisation (95 %, as documented by ¹H-NMR analysis, see above). Agreeably, this aspect was not sufficiently addressed in our manuscript and, therefore, we have now also compared both the retention and the elution of an Azo-tagged protein under constant or pulsed illumination and/or in the dark, now discussed in lines 364–371 and described in Suppl. Fig. S14.

In summary and under practical considerations regarding protein affinity purification, it does not much matter if the sample is applied with just 75 % trans-state (instead of desirable 100 %), as done with a fixed photostationary composition (i.e. in the dark), or in a dynamical photostationary equilibrium under ongoing illumination with daylight (or at 430 nm). In the worst case, this may just lead to a loss of 25 % of the applied Azo-tagged POI, which is the proportion in a fixed cis-state that appears in the flow-through of the column when washing in the dark. In this context, one should not forget that, apart from the light-triggered elution, the other major advantage of our affinity chromatography is the extremely low background binding activity of the alpha-CD matrix, which allows the efficient removal of all host cell proteins!

However, elution is not only driven by a potentially dynamic cis/trans-PHOTOisomerization (in the presence of light at a given wavelength) as seems to be presumed by this Reviewer. In fact, the host/guest complex formation between the Azo-tagged POI and the alpha-CD groups of the affinity column is not a "yes-or-no event" but is dictated by the chemical Law of Mass Action, which itself has a dynamical nature. This means that even in a pure trans-state, the Azo-tagged POI should slowly elute with the chromatography buffer flow, even though in a retarded manner (generally called "retention"), depending both on the microscopic affinity and the relative concentration of the binding-competent isomer.

As we have demonstrated, the dissociation constant between the azo-amino acid (in its trans-state) and alpha-CD is just 91 uM in solution and corresponds to a moderate affinity. Thus, binding of the trans-form to the affinity column is not static but highly dynamic, in agreement with the well known relation $K_d = k_{off} / k_{on}$. Under such circumstances, retention of an analyte on the chromatography column is usually explained in chromatographic theory by the concept of "theoretical plates", initially proposed by the Nobel prize winners Martin and Synge in 1941, which assumes repeating binding/dissociation equilibria at consecutive infinitesimally thin sections of the column. Consequently, even if the equilibrium is only partially on the side of the bound state, its repeated occurrence leads to a macroscopic retention (but not complete fixation)

of the analyte or ligand (here the azo-amino acid or the Azo-tagged POI in the trans-state) under buffer flow.

Conversely, if the analyte/ligand is switched to the cis-configuration, with its practically undetectable affinity, the ligand is by far in the unbound state and rapidly elutes from the alpha-CD affinity column with the buffer flow. The phenomena we have seen both for the isolated azo-amino acid and for the Azo-tagged POIs perfectly fit into this generally accepted mechanistic concept, which also explains the immediate and efficient elution upon illumination at 355 nm. Thus, the Reviewer's assessment that "Following the implied principle, AzoF could neither be fully loaded nor fully eluted." is just not true.

In other words, if we have a mixture of two different molecules (i.e. in the trans and the cis configuration) of which only one species (trans) shows measurable affinity towards alpha-CD, a light-induced shift in the composition of the mixture (for simplicity, let us say from 90 % trans / 10 % cis to 9 % trans / 91 % cis) would lead to a reduction in the apparent affinity of the molecular ensemble as a whole by a factor 10 (in fact, in our setup it is even higher). From the (bio)chemical perspective, this is the driving force that effects the rapid elution of Azo-tagged proteins from the alpha-CD affinity column upon illumination at 355 nm.

For those readers who are not fully aware of the underlying fundamental principles, we have now illustrated the combined effects of the dynamical binding equilibrium and photo-isomerisation with a new graphical sketch added as panel (B) to our Scheme 1. This should complement an (admittedly) somewhat simplified view of our Excitography technique presented in panel (A). Nevertheless, there is no "discrepancy with our hypothesis" and there is no reason to "reconsider the principle of excitography".

p. 3, l. 138: "... at $t_{1/2} \sim 196$ h (Fig. 1F)."

- The thermal stability of the cis isomer of AzoF loses its importance for the strategy following the proposed principle
- Moreover, the thermal relaxation of the cis-enriched PSS of AzoF was measured for 100 h and the data show a nearly linear course. The exponential fit is therefore not reliable.

⇒ I suggest to keep the data as they are, but remove the $t_{1/2}$ because the exact value is not significant for the purification

Au: We believe that the surprisingly low speed of relaxation in the dark has practical relevance and deserves mention, also in view of the much shorter half-lives of the cis-state from the literature quoted (even though measured in organic solvents there). However, we take the point of better explaining the validity of our curve fit based on the knowledge of the asymptotic values (see the amended Suppl. Results A), thus also addressing the corresponding comment by Reviewer #2.

p.4, l.187-192: "The compound was largely retained in this zone, showing just modest movement with the flow of the mobile phase when washing the column..."

- No data are provided on the amount of AzoF in the flow-through or on the column.
- To substantiate the principle of the purification strategy, it would be important to know whether AzoF also moves modestly with the flow of the mobile phase in the dark.
- Suppl. Fig. S4 shows that here elution occurred after irradiation. Hence, some of the AzoF might still be retained on the column due to the non-quantitative switching.
- In regard to the fact that most purchasable LEDs vary either in their peak wavelength or the width of the wavelength spectrum, it would be beneficial for the reproducibility to test the effect of various LEDs (with different peaks and widths) on the rate of elution.

⇒ I suggest to test the rate and efficiency of elution by determining the absorbance of AzoF similar to Suppl. Fig. S10B for multiple light sources including daylight and different LEDs. Besides the change in wavelength, it is also important to measure the influence of the light intensity, since this may also affect the rate of elution.

⇒ Compare these data with the flow of AzoF in the dark.

⇒ If possible determine the amount of AzoF that is retained on the column after elution since this is important to exclude contaminants in the next purification procedure.

Au: We believe that these questions were already fully answered in our responses to this Reviewer above, drawing here the attention in particular to the new Suppl. Fig. S14. The UV light intensity provided by our LED setup (see Suppl. Fig. S1 and Suppl. Method A) obviously is sufficient to trigger the release of the Azo-tagged POIs (see, e.g., Fig. 1I and, in particular, the Suppl. Video). And this is achieved by illumination at 355 nm with much less power (20–40 mW) than the 365 nm LED UV light source applied in the studies quoted by this Reviewer #4 (2900 mW; Hiefinger et al., 2023).

Finally, we can confirm that no Azo-tagged POI was ever seen retained on the alpha-CD affinity column after UV-induced elution (as should actually be apparent from the many elution profiles documented by SDS-PAGE in our paper). By the way, carry-over by residual bound Azo-tagged protein to a following purification run can easily be ruled out by washing with NaOH as we had explained in Suppl. Method O (line 337).

p.8, l. 346-354, Suppl. Fig. S6 and

p.9, l. 394-400: „While these experiments clearly demonstrated the efficient purification of POIs carrying a C-terminal Azo-tag on an α -CD affinity column, even from a whole cell extract, there was always a proportion of the recombinant protein that did not absorb to the affinity column (cf. Suppl. Fig. S6 and Fig. 3A, C) and appeared in the flow-through. UV/Vis absorption spectra as well as ESI-MS analyses indicated that this was due to a fraction of the biosynthetic protein that lacked the C-terminal Baf amino acid.”

- While the UV/Vis data for the conclusion that the C-terminal AzoF is lacking appears to be provided in Suppl. Fig. S6, the ESI-MS data are missing.
- Even though this conclusion is logical, it is somewhat astounding since the authors expressed in a Δ RF1 strain, in which the termination at the amber stop codon is minimized.
- Suppl. Fig. S6B makes no sense to me. The eluate should not have a pronounced AzoF peak at ~320 nm, since most of the AzoF has switched to the cis isomer with irradiation. Moreover, the authors state that in the flow-through the trans-AzoF absorption was largely absent. This is true; however, the spectrum looks to me like a normal UV/Vis trace of the cis-enriched PSS. Are the authors sure that both spectra were not mixed up? It would make more sense for the flow-through to have a trans-enriched PSS, and for the eluate to have a cis-enriched PSS.

⇒ Please provide the ESI-MS data for the lack of AzoF in the C-terminal tag of the POIs.

⇒ Please discuss why AzoF could be missing despite expression in the Δ RF1 strain.

⇒ Please check the UV/Vis raw traces shown in Suppl. Fig. S6B or remeasure to make sure that they were not mixed up.

⇒ Determine the trans/cis composition in the two states shown in Suppl. Fig. S6B (see above).

Au:

- ESI-MS data for the lack of Pap in the C-terminal Azo-tag are now shown in Suppl. Fig. S12C.

- Suppl. Fig. S6B (now S12B) should "make sense" if reading our remark "following exposure to daylight (in order to revert Baf to the trans-configuration)" in the legend (lines 1092–1093). Nothing was mixed up here.

- In addition, the NMR measurements proposed by Reviewer #1 and discussed above have led to a better/quantitative understanding of the cis/trans photostationary composition under different illumination conditions. According to this, the azo-amino acid contains already approximately 25 % cis-form under diffuse daylight. This explains why we can detect a small proportion of the intact POI (carrying the full-length Azo-tag), which had been produced and applied under daylight, in the flow-through of the column when measuring the absorbance at 326 nm.

- Thanks for addressing the point of much improved, but not fully complete amber stop codon suppression in our engineered E. coli expression strain, which was also puzzling for us. Initially, we had confirmed the replacement of the prfA cistron by the coding region for the Kan resistance enzyme via analytical colony PCR using primers flanking the relevant genomic region and, subsequently, also by Bacterial Genome Sequencing employing Oxford Nanopore Technology (Eurofins Genomics), yielding contigs that exactly matched our planned genomic construct. Meanwhile, we had a deeper look into the raw data of the Nanopore reads using the Geneious software package and we found some evidence that during the genome engineering a gene duplication via homologous recombination must have occurred, apparently triggered by a hopping gene (insertion sequence IS1) in the neighborhood. While this remaining copy of the prfA gene likely explains the unaffected rapid growth behavior and high recombinant protein expression capacity compared with the parent strain, it is obviously less active in the new genomic environment due to the triplicate occurrence of the upstream hemA cistron including its regulated promoter, which is now described in greater detail in the amended Suppl. Method section R. This explains the much enhanced amber stop codon suppression (by a factor 5) as had been documented in the Suppl. Results section C. To account for these new findings we have changed the name of our engineered E. coli strain to NEBExpress(lowRF1).

UV-trace in Suppl. Fig. S10B and SDS-PAGE in Suppl. Fig S8E

- Both show that the POIs are still found in the flow-through and the wash fractions after switch to the N-terminal Azo-tag and optimization of the neighboring residues.
- As suggested above, this might be explained by loading and washing the column under daylight, which is able to switch AzoF.

⇒ Please present experiments, in which the loading and washing steps are performed in the dark to increase the retention of AzoF on the column.

Au: This point has already been addressed above, drawing again the attention to the new Suppl. Fig. S14.

Limitations of Excitography.

P. 1, l. 28: “ ...in a functional state ...”

- This might be true for most proteins of interest. However: i) the Azo-Tag may affect protein/enzyme activity, e.g. when the termini are involved in complex formation, ii) the Azo-Tag may drastically reduce the solubility and therefore expression yields (for example, my group was not able to produce AzoF-Ubiquitin despite one year of trying different approaches), iii) light exposure can affect the function of light-sensitive proteins, particularly if the protein has light-sensitive cofactors bound (e.g. PLP or FAD), iv) AzoF may be a critical epitope in in vivo studies.
- In part these limitations might be solved by a cleavable Azo-Tag.
- For light-sensitive proteins/cofactors, elution could be performed during visible light

irradiation. This probably reduces the rate of elution, however, complete elution of the POI should still be possible.

⇒ Please discuss the mentioned limitations.

⇒ If possible, include LEDs emitting visible light in the above suggested experiments.

Au: We have mentioned potential limitations and the (trivial) possibility of introducing a protease cleavage site in order to remove the Azo-tag in our amended Conclusions section (see lines 677–679), also in response to Reviewer #1. In addition, we have now included experiments demonstrating that enzymatic activity, antigen-binding function, protein folding and overall hydrophobicity are not affected by the presence of the Azo-tag (see the new Suppl. Figs. S21 and S22). Hence, in general there is no more limitation with the Azo-tag than with many other well established affinity tag systems and an overly critical view is not justified. Of course, we can only speculate about the problems with the preparation of AzoF-Ubiquitin in the Reviewer's laboratory. However, we will be happy to invite her/him to use our newly developed aaRS and provide the corresponding expression plasmid after acceptance of this manuscript.

p.10-11, l.479-482: „The resulting purified protein solutions were directly suitable for subsequent assays, without the need for buffer exchange or removal of elution agents, as here demonstrated with enzyme activity assays and antigen binding assays, respectively (Fig. 3F,G and Suppl. Fig. S11B, C).”

⇒ Please include a comparison to the activity of the wildtype enzyme (without Azo-Tag and purified using e.g. Strep-Tag) showing whether the Azo-Tag somehow influences the activity.

Au: See above.

Further limitations are: i) the expression yields are reduced with unnatural amino acids (rule-of-thumb: to 10%), ii) the chromatography is difficult to use for xenoproteins since incorporation of two different UAAs is still hard to accomplish.

⇒ Please discuss the mentioned limitations.

Au: The Reviewer's "10% rule-of-thumb" does not apply to our manifold optimized genetic system, as we had explained in our manuscript, including the extensive experimental effort to get there. We may also draw her/his attention to the Suppl. Results section C, where we have now additionally reported the yield of an (N-terminally) Azo-tagged POI to be essentially the same as the one obtained with a coding region where the amber stop codon had been replaced by a conventional codon for Phe. In fact, this yield is approximately 5-fold higher with our engineered E. coli strain compared with the parental strain NEBExpress (which carries the intact single gene for RF-1). This aspect was already addressed in our response a few paras. above.

Minor points

p.2, l.83: “non-natural amino acid (nnAa)”

⇒ Please use the established terminology to improve the visibility of this manuscript. It is either non-canonical amino acids (ncAA) or unnatural amino acids (UAA)

Au: When we first published in this field (see PMID: 20837025) the term "non-natural amino acid" (nnAA) was well established. This term currently yields 1135 hits in a PubMed search. The term "non-canonical amino acid" (ncAA) has become slightly more common in the recent years, yielding 1876 hits. "unnatural amino acid" (UAA) even results in 3285 hits, but may be misleading from the semantic perspective as not all non-canonical amino acid are "unnatural" in the true sense. Furthermore, its abbreviation is confusing as it may be mixed up with the nucleotide triplet of the ochre stop codon (UAA). Thus, after careful consideration, we have now used the term "ncAA" throughout our manuscript.

p. 2, l. 93: “L-benzazo-phenylalanine (Baf). Baf was previously incorporated (then dubbed AzoPhe)”

Established terminology serves to increase visibility. The described photoswitchable unnatural amino acid is named azobenzene-4'-phenylalanine and its common abbreviation is “AzoF”.

⇒ Please use the common term to improve the visibility of this manuscript.

Au: While AzoF may be common for this Reviewer, this is certainly not the case in the wider literature, and we believe that scientific accuracy should be the guideline. Bose et al. (2006) (Ref. 13), who introduced this ncAA into recombinant proteins, used the abbreviation "AzoPhe". However, they were not the first to synthesize this amino acid, and they were not even the first to incorporate it into a biosynthetic protein, which was initially achieved by Liu et al. (1997) (Ref. 12 in the revised manuscript). The first preparative synthesis of this non-canonical amino acid was described long before by Goodman & Kossoy in 1966 (JACS 88:21), who designated it "L-p-(phenylazo)phenylalanine", in line with chemical nomenclature (see also: <https://pubchem.ncbi.nlm.nih.gov/compound/10084432>). Hence, after careful reconsideration, we have decided to use the abbreviation "Pap" which (i) has been much earlier introduced into the literature/field with the pioneering work of Liu et al. (1997), (ii) is a true acronym and (iii) is compatible with the established three-letter code of amino acids, unlike AzoF or AzoPhe.

p. 2, l.94-96: “... protein-functional studies into the Escherichia coli catabolite activator protein CAP, into myoglobin from sperm whale and into the ImGPS glutaminase subunit HisH from *Thermotoga maritima*.”

It does not become clear to the reader for what purpose AzoF is generally employed. The field of photoxenoprotein engineering is growing, particularly due to the high interest to control protein activity with light in various research fields.

⇒ I therefore recommend to add a brief description of AzoF, how it works and some examples for how it is used in the introduction (in this regard, please note that there are two more publications on AzoF in ImGPS).

Au: We believe that we have already given sufficient account on previous applications of Pap in biochemistry in the Introduction section (see lines 97–101). The Reviewer should understand that this is not a generic review about photobiochemistry or light-modulation of enzyme activity.

p. 2, l.95: “...into the ImGPS glutaminase.”

⇒ Please provide an explanation for the acronym ImGPS

Au: In the spirit of scientific conciseness we have slightly rephrased this sentence and avoided this abbreviation while keeping the literature reference (see lines 100–101). Otherwise, when stating the rather long full name "Imidazole glycerol phosphate synthase" together with "the glutaminase subunit HisH from *Thermotoga maritima*" in the same sentence, we would fear to distract the reader from the actual message of our study, which focuses at the light-controlled affinity purification of proteins, but not the allosteric control of enzymes. Alternatively, we may suggest to delete this reference as whole since there is no direct relation to our study and it rather refers to the general scientific background.

p.5, l.207-211: “..., we chose the pyrrolysyl-tRNA synthetase (PylRS) from *Methanosarcina mazei* with altered substrate specificity and its cognate tRNA^{Pyl}. This naturally evolved orthogonal pair is known for its more robust amber suppression in *E. coli* and less cross-reactivity with endogenous canonical amino acids.”

- It is unclear to me why the authors chose to evolve a new aaRS for AzoF, when three different ones have already been available: MjTyrRS based from Schultz (DOI

10.1021/ja055467u), MjTyrRS based from Amiram (DOI 10.1002/adfm.202011276) and MbPylRS based from Deiters (DOI 10.1002/cbic.201800226).

- Although the mentioned advantages of PylRS are true, TyrRSs are generally more suitable for AzoF since they have a larger binding pocket that is better for bulky and hydrophobic UAAs than the one of PylRS. In this regard, it would be interesting to have a direct comparison of the efficiency of all four evolved systems (including the one here).
- Unfortunately, the only indication of efficiency for the newly evolved PylRS is given in form of the reduced amount of AzoF required for expression (l. 277). It would be good to supplement the yields of protein in mg per expression medium.
- A discussion comparing the difference of the newly evolved PylRS to the established aaRSs is missing.

⇒ The higher efficiency of the newly evolved PylRS for AzoF incorporation over the other three established aaRSs should be proved by a comparison of expression yields with each aaRS.

⇒ Please provide a discussion that compares the four aaRSs for AzoF incorporation, possibly including a structural comparison, e.g. provided with AlphaFold.

Au: We have experience with engineered aaRS enzymes derived from both MjTyrRS and MmPylRS in our lab for many years. Generally, the MmPylRS shows better performance with regard to biosynthetic protein yield, a trend also seen elsewhere in the literature and confirmed by the Reviewer in her/his statement above. That MmPylRS can provide a large binding pocket to accommodate a bulky and hydrophobic ncAA, in particular certain meta-substituted Pap derivatives, was already convincingly demonstrated by others (Hoppmann et al., 2014; see Ref. 14), which in fact provided the basis for our aaRS engineering effort, as described in Suppl. Results B. Following the suggestion by the Reviewer, we have now depicted a structural model of our mutant PapRS#34 generated with AlphaFold 2. This nicely illustrates how a few amino acid exchanges within the region of the substrate pocket that accommodates the side chain of the ncAA substrate create space for the azobenzene moiety, which is actually shorter than the natural substrate pyrrolysine but a bit more bulky (see Suppl. Fig. S11).

Yields of a typical Azo-tagged POI had been reported in Suppl. Results section C. To further address this question, we have now performed a side-by-side expression experiment employing the published vector pEVOL encoding AzoPheRS (Suppl. Refs. 28 and 31), which we had already in our laboratory. Using the same improved expression strain, NEBExpress(lowRF1), and the same concentration of Pap in the culture medium, this resulted in almost double yield (21.8 mg versus 12.0 mg per liter) of purified protein with our engineered MmPylRS than with the pEVOL vector (see the amended Suppl. Results C). At the same time, however, our system (based on an expression vector encoding both the Azo-tagged POI and the engineered aaRS) allows much easier genetic manipulation. As the Reviewer surely is aware, pEVOL carries two identical copies of the structural gene for the AzoFRS enzyme (which was described by the authors to be crucial for efficient incorporation of Pap via amber suppression, see Suppl. Ref. 28). This considerably complicates the cloning or further mutagenesis of the engineered aaRS, apart from the additional effort for handling a second expression plasmid that encodes the POI and, furthermore, the burden of double antibiotic selection.

Besides, the publication from the Deiters laboratory (Luo et al., 2018) mentioned by the Reviewer specifically deals with the incorporation of fluorinated azobenzene amino acids in eukaryotic cells (not E. coli), which is not relevant in our context. Furthermore, this work employed a homologous PylRS from a different organism, *Methanosarcina barkeri*, and most of the mutations used there to achieve incorporation of the fluoro-azo-amino acids are different from the equivalent residues in our engineered PylRS (in fact, apart from the generic mutation Y384F, see Suppl. Ref 23, there is only one mutated residue in common). Of note, this study

does not say anything about protein yields, let alone presenting a comparison with other Pap-specific aaRS.

On the other hand, the system based on MjTyrRS described by Amiram (Israeli et al., 2021) mentioned by the Reviewer is totally different from ours as it utilizes a chromosomally encoded engineered aaRS in the context of the extensively engineered *E. coli* strain C321.ΔRF1 to prepare light-responsive protein-polymers, and this bacterial strain is not available to us; here again, the authors do not provide a side by side comparison with other previously described systems for the incorporation of the azo-amino acid, such as pEVOL. Hence, we do not see it as our task to validate or reproduce these thematically remote studies from other laboratories aiming at different applications.

p.8, l.337-339: “Notably, exposure of the column to 355 nm UV light ...– in the absence of daylight – triggered the elution of both bound Azo-tagged POIs without delay, thus obviating the need for an additional incubation period.”

⇒ Please improve this sentence for clarity, particularly the “need for an additional incubation period”.

Au: We have deleted this potentially confusing statement. This point has become obsolete in some sense in the light of our new quantitative measurements of the light-induced affinity chromatography comparing continuous and pulsed illumination, as explained above.

Reviewer #5 (Remarks to the Author):

I co-reviewed this manuscript with one of the reviewers who provided the listed reports as part of the Nature Communications initiative to facilitate training in peer review and appropriate recognition for co-reviewers.

AUTHORS' RESPONSE TO REVIEWERS' COMMENTS

Reviewer #1 (Remarks to the Author):

I highly appreciate the effort of authors to improve the quality of the manuscript and to address the comments of me and other reviewers. From my perspective, my numerous comments have been sufficiently addressed by the authors, and thus I recommend the acceptance of the manuscript in its current form. I particularly appreciate the effort to determine the isomeric composition via NMR. I agree that the overall effect (binding vs. release) occurs with satisfactory selectivity level. Yet, in light of the multiple problems known to the photoswitch community and often successfully addressed in the last two decades, it was important to understand the behavior of Azo label at the molecular level. The results obtained from the quantification are not impossible, but pretty uncommon for the type of photoswitch selected for this research. Therefore, this quantification was needed for understanding and justification of the conclusions drawn by authors.

Yet, as the demonstrated experiments do not occur solely in solution, but in interactions with the solid phase, it is not only the photoconversion, but (apparently predominantly) the sequestering of the trans-isomer (whenever it is generated in the photodynamic equilibrium) to the solid phase, which determines the macroscopic outcome.

While the authors should not be criticized for not addressing all problems known to the photoswitch and photopharmacology community, careful analysis of the comments from me and other reviewers obviously helped to increase the mutual understanding.

Minor (successfully addressed) points:

- penetration of the solid phase of the column with light - while this is probably irrelevant for the diameter demonstrated in the publication, the scattering and attenuation might have still be a potential concern for larger columns. Therefore it is good that authors determined the transmittance in the used setup, which will clearly indicate what level of limitation would that issue eventually be able to cause (or not).

- reductive conditions and azobenzenes - this is a serious concern for azobenzene applications, especially in vivo or with physiological fluids. This was carefully addressed in many publications from e.g. Andrew Woolley lab (e.g. Samanta et al. J. Am. Chem. Soc. 2013, 135, 26, 9777–9784 <https://doi.org/10.1021/ja402220t>). The fact that this issue was not addressed in previous publications concerning amber codon-type biosynthetic incorporation of PAP or its derivatives, may rather refer to the overall inefficiency of the whole chain of operations, where this issue somehow might have been overshadowed by other bottlenecks

- careful determination of the photoconversions - looking at the earliest reports, like Bose et al JACS paper from 2006 (Pete Schultz group, J. Am. Chem. Soc. 2006, 128, 2, 388–389, <https://doi.org/10.1021/ja055467u>), the photoconversion of PAP was 55% to the cis isomer, return to trans-isomer was quantitative. These values are more typical for that chromophore, therefore the authors should not be surprized on the reviewers' sceptical comments on the photoconversion issue in your manuscript. It is good that you performed the quantification experiments.

Au: Thanks for acknowledging our efforts. To complete the data set regarding the NMR analysis of the light-dependent cis-trans isomerisation of Pap we have now also included the data point for the pure trans-Pap, which was procured as the crystalline hydrochloride, in Suppl. Figs. S4, S6 and S7. This confirmative piece of information did not change the message but might be of interest to the reader.

Reviewer #2 (Remarks to the Author):

This is the revised version of a manuscript that I have previously reviewed. I thank the authors for carefully addressing all my concerns. The inclusion of performance data such as the determination of the dynamic binding capacity of the chromatographic material and data on its chemical stability is crucial for wider application. Regarding the kinetic measurement of cis-state stability, I still believe that increasing the measurement time would give a more robust fit, but since this is not an experiment that is crucial to the validity of the developed method, setting an asymptotic limit in the fitting routine and performing a single exponential decay fit will suffice. The manuscript has been greatly improved and from my side I gladly support its publication.

Au: Thanks for acknowledging our efforts.

Reviewer #3 (Remarks to the Author):

In this revision, the authors added new experimental results and discussion and commented on the effort to make the materials accessible to the community. My concerns have been addressed.

Au: Thanks. We have now also made the plasmid and the sequence of the engineered E. coli strain publicly accessible as indicated in the Data Availability statement.

Reviewer #5 (Remarks to the Author):

Key results

The manuscript entitled "Protein purification with light" presents a strategy to purify protein samples using a light-sensitive tag. The tag comprises the azobenzene based unnatural amino acid known as AzoF (here named Baf), which allows the binding of the protein in the dark or under day-light to an α -cyclodextrin resin. Elution of the protein of interest is then facilitated by irradiation with UV light. The presented research appears to be original and innovative, and a lot of work and thought has been put into the success of this strategy from a biochemical point of view. I am particularly convinced that this method will have an impact on diverse fields of protein chemistry due to the extremely high binding efficiency and purity of the eluted samples. I therefore very much like to see this manuscript published. However, I have two major concerns regarding the publication in a high-impact journal such as Nature Communications, which should be addressed appropriately.

Key concerns

First, I have to point out that its photochemical/physical behavior is misrepresented (as I will explain in more detail below). As a result thereof, data were misinterpreted and other data that are crucial to show the reproducibility of this technique are missing. Second, although they nicely emphasized on the benefits of this strategy and minimized some limitations of it, the authors missed to discuss the (remaining) major limitations of using an unnatural amino acid in a protein tag. I have provided detailed arguments below sorted into major and minor points. These illustrate how the manuscript should be revised to entirely cover the state-of-the-art and how it should be complemented with further experiments to further analyze the principle and improve the reproducibility of this purification strategy (which in my opinion might also be necessary for the pending patent).

Major points

Misrepresentation of the photochemical/physical behavior of azobenzenes [throughout the manuscript (particularly p. 3)].

- The trans and cis configurations of azobenzenes (including AzoF) do not exhibit distinct absorption bands that allow their individual spectroscopic detection. The two signals in the UV and visible range originate from the excitation of the $\pi \rightarrow \pi^*$ and $n \rightarrow \pi^*$ transition of the azobond, respectively. The azobond is the central moiety in trans and cis. If you follow the isomerization over time you will not follow the appearance of a second peak for cis and the decline of the first peak for trans as for chemical reactions. Instead both peaks change their intensity and maximum smoothly with the isomerization.
- Azobenzenes (including AzoF) cannot switch quantitatively because of their overlapping spectra. As a result, a photo-induced equilibrium named photostationary state (PSS) is established. Each irradiation step switches between a trans-enriched and cis-enriched PSS. This means that during irradiation with any wavelength that falls in the range of AzoF absorption, AzoF will switch back and forth constantly keeping up the equilibrium. The intensity and wavelength of light thereby defines the rate of this back-and-forth switching.
- The composition between trans and cis in the PSS does not only depend on the extinction coefficients ϵ but also the quantum yield Φ . Importantly, both factors are dependent on the wavelength in azobenzenes. Since the quantum efficiency is often difficult to measure, the best wavelength for the photoswitch is usually determined by trying different LEDs with different wavelength maxima.
- Moreover, both ϵ and Φ are extremely dependent on the environment, such as buffer, ions, acids/bases. This means AzoF will exhibit a different trans/cis composition at the

PSS and a different rate of isomerization when incorporated into proteins compared to its isolated form (and between different proteins).

- Due to the overlap of the trans and cis spectra, spectral evaluation only allows for an estimation of the trans/cis composition at a defined state.
 - Finally, owing to the wavelength dependence of ϵ and Φ , different LEDs result in different isomerization rates and trans/cis compositions at the PSS. This can even be the case for two LEDs with the same peak wavelength. The reason lies within the different width of the Gaussian wavelength spectrum of the LEDs. Hence, using different LEDs can have a major impact on the reproducibility of any experiment with photoswitches.
- ⇒ In addition to my explanations, I suggest to go back to literature and research the photochemistry of azobenzenes. Works of Viktor Szymanski, Ben Feringa, Dirk Trauner, Alex Deiters, and Andrea Hupfeld (née Kneuttinger) will help.
- ⇒ Subsequently, I find it important to update the state-of-the-art on azobenzenes in the introduction (currently missing) and the results section.

Misinterpretation of data.

p.3, l. 99-142: “However, the spectroscopic properties of Baf have been only partially characterized up to now ... Therefore, we prepared Baf following the published procedure and investigated its absorption spectra as well as its isomerization between the trans- and cis-state in greater detail...”

- Owing to the sensitivity of ϵ and Φ on the environment, the trans/cis composition, isomerization rate and thermal stability of isolated AzoF is only limitedly representative to AzoF incorporated into proteins
- ⇒ Please provide spectral analyses for at least two of the chosen proteins with incorporated AzoF to compare with isolated AzoF.

p. 3, l. 109-111: “Quantitative switch to the excited state, i.e. the cis-configuration, was easily achieved by illuminating the solution in a quartz cuvette from the top (Suppl. Fig. S2) with mild UV light at 355 nm using a 1.2–2.4 mW LED for =30 min (Fig. 1E)”

- The cis configuration is not the excited state, it is instead the product of excitation (to S2) and return to the ground state via isomerization. Excited azobenzenes also fall back to the ground state without isomerization due to their low quantum yield.
- ⇒ Please correct this statement.
- As explained above, azobenzenes cannot switch quantitatively. Instead, irradiation results in a trans- or cis-enriched PSS.
- ⇒ Determine the composition of trans and cis of the state shown in Fig.1D(red), e.g. by using spectral analysis (<https://www.biorxiv.org/content/10.1101/2024.07.04.602025v1>, <https://doi.org/10.1016/bs.mie.2022.12.003>, <https://chemistry-europe.onlinelibrary.wiley.com/doi/10.1002/chem.202004061>)
- Isomerization and trans/cis composition at the PSS depend on the intensity and wavelength of irradiation. Hence, it is more important for the reproducibility to provide the actual intensity of light that was used for irradiation and the wavelength spectrum of the LED than the technical specification.
- ⇒ Please provide the intensity of irradiation (to be exact measure with a sensor in the same distance to the LED as the sample; or at least provide the voltage and current that were applied) and the wavelength spectrum of the used LED.

p.3, l. 111-124: “The resulting solution of cis-Baf ... Baf shows distinct absorption bands that not only allow the individual spectroscopic detection of its trans- and cis states, ... However,

to optimize the equilibrium in the photostationary state..., we chose a wavelength of 355 nm for efficiently switching trans→cis, corresponding to a high ratio $\epsilon_{\text{trans}}/\epsilon_{\text{cis}} = 8.3$ (Suppl. Fig. S3)”

- As explained the trans/cis composition depends not only on ϵ but also Φ making the analysis in Suppl. Fig. S3 redundant.
- ⇒ Either find a different logical reason for choosing 355 nm or provide a wavelength screen, in which the trans/cis composition at the PSS for different wavelengths is shown.

p.4, l.185: “When a 5 mM solution of Baf was applied under daylight (i.e. in its trans-state) ...”

- Under daylight AzoF is also isomerizing. We have to cover all our AzoF samples since daylight or even our lab light is able to switch AzoF. The thereby established PSS might not be cis-enriched, however back- and forth isomerization is still taking place during light exposure.
- ⇒ Please correct this statement.

p.5, l.193-195: “Measurement of the absorption spectrum of the eluted fraction revealed the presence of Baf in its cis-state by showing only absorptions at 293 nm and 426 nm and no peak at 326 nm.”

- The results are not provided.
- ⇒ Please provide the respective UV/Vis spectra and determine the trans/cis composition.

Consequences on the principle and reproducibility of the strategy.

p.2, l.87: “... (Scheme 1).”

- The description of the principle in the main text is quite brief. For example, the new name “Excitography” is missing.
- The authors only roughly describe the principle of their new chromatography. What is implied to the reader is that the proteins are loaded in a 100% (fixed) trans state of AzoF with high affinity to the resin. After washing (still 100% trans), the column is irradiated so that AzoF isomerizes to 100% (fixed) cis. Concomitantly, the protein loses its affinity and can be eluted concomitantly.
- As I have explained, AzoF cannot be switched quantitatively to 100% cis and exposure to daylight can lead to its isomerization resulting in a trans/cis composition of <100% trans.
- Following the implied principle, AzoF could neither be fully loaded nor fully eluted.
- However, the provided results indicate that AzoF-labeled proteins can at least be fully eluted (even though this has only been shown qualitatively with pictures of the columns). In my opinion, this can be explained by the methodical realization. Fig. 2I and Suppl. Fig. S8F show that the column was simultaneously irradiated during elution. Hence, AzoF was continuously switching between trans and cis during the elution step allowing the full elution of the AzoF-labelled protein. It appears to me that excitography works by regulating the retention strength with light. In the dark, only trans AzoF will bind to the resin, whereas cis AzoF will flow through. If the column is exposed to weak daylight, trans/cis switching of AzoF is initiated, though with a probably trans-enriched PSS and with probably slow isomerization rates. This results in high retention and probably slow elution. High intensity UV light instead creates a cis-enriched PSS with high isomerization rates. This results in low retention and faster elution.

- ⇒ Please reconsider the principle of excitography.

⇒ In case of agreement, please explain the principle in greater detail in the main text, revise Scheme 1 and adjust the manuscript, e.g. the conclusion, in which the authors already point out some discrepancy with their hypothesis.

p. 3, l. 138: “... at $t_{1/2} \sim 196$ h (Fig. 1F).”

- The thermal stability of the cis isomer of AzoF loses its importance for the strategy following the proposed principle
- Moreover, the thermal relaxation of the cis-enriched PSS of AzoF was measured for 100 h and the data show a nearly linear course. The exponential fit is therefore not reliable.

⇒ I suggest to keep the data as they are, but remove the $t_{1/2}$ because the exact value is not significant for the purification

p.4, l.187-192: “The compound was largely retained in this zone, showing just modest movement with the flow of the mobile phase when washing the column...”

- No data are provided on the amount of AzoF in the flow-through or on the column.
- To substantiate the principle of the purification strategy, it would be important to know whether AzoF also moves modestly with the flow of the mobile phase in the dark.
- Suppl. Fig. S4 shows that here elution occurred after irradiation. Hence, some of the AzoF might still be retained on the column due to the non-quantitative switching.
- In regard to the fact that most purchasable LEDs vary either in their peak wavelength or the width of the wavelength spectrum, it would be beneficial for the reproducibility to test the effect of various LEDs (with different peaks and widths) on the rate of elution.

⇒ I suggest to test the rate and efficiency of elution by determining the absorbance of AzoF similar to Suppl. Fig. S10B for multiple light sources including daylight and different LEDs. Besides the change in wavelength, it is also important to measure the influence of the light intensity, since this may also affect the rate of elution.

⇒ Compare these data with the flow of AzoF in the dark.

⇒ If possible determine the amount of AzoF that is retained on the column after elution since this is important to exclude contaminants in the next purification procedure.

p.8, l. 346-354, Suppl. Fig. S6 and

p.9, l. 394-400: „While these experiments clearly demonstrated the efficient purification of POIs carrying a C-terminal Azo-tag on an α -CD affinity column, even from a whole cell extract, there was always a proportion of the recombinant protein that did not absorb to the affinity column (cf. Supp. Fig. S6 and Fig. 3A, C) and appeared in the flow-through. UV/Vis absorption spectra as well as ESI-MS analyses indicated that this was due to a fraction of the biosynthetic protein that lacked the C-terminal Baf amino acid.”

- While the UV/Vis data for the conclusion that the C-terminal AzoF is lacking appears to be provided in Suppl. Fig. S6, the ESI-MS data are missing.
- Even though this conclusion is logical, it is somewhat astounding since the authors expressed in a Δ RF1 strain, in which the termination at the amber stop codon is minimized.
- Suppl. Fig. S6B makes no sense to me. The eluate should not have a pronounced AzoF peak at ~ 320 nm, since most of the AzoF has switched to the cis isomer with irradiation. Moreover, the authors state that in the flow-through the trans-AzoF absorption was largely absent. This is true; however, the spectrum looks to me like a normal UV/Vis trace of the cis-enriched PSS. Are the authors sure that both spectra were not mixed up? It would make more sense for the flow-through to have a trans-enriched PSS, and for the eluate to have a cis-enriched PSS.

⇒ Please provide the ESI-MS data for the lack of AzoF in the C-terminal tag of the POIs.

- ⇒ Please discuss why AzoF could be missing despite expression in the Δ RF1 strain.
- ⇒ Please check the UV/Vis raw traces shown in Suppl. Fig. S6B or remeasure to make sure that they were not mixed up.
- ⇒ Determine the trans/cis composition in the two states shown in Suppl. Fig. S6B (see above).

UV-trace in Suppl. Fig. S10B and SDS-PAGE in Suppl. Fig S8E

- Both show that the POIs are still found in the flow-through and the wash fractions after switch to the N-terminal Azo-tag and optimization of the neighboring residues.
- As suggested above, this might be explained by loading and washing the column under daylight, which is able to switch AzoF.
- ⇒ Please present experiments, in which the loading and washing steps are performed in the dark to increase the retention of AzoF on the column.

Limitations of Excitography.

P. 1, l. 28: “ ...in a functional state ...”

- This might be true for most proteins of interest. However: i) the Azo-Tag may affect protein/enzyme activity, e.g. when the termini are involved in complex formation, ii) the Azo-Tag may drastically reduce the solubility and therefore expression yields (for example, my group was not able to produce AzoF-Ubiquitin despite one year of trying different approaches), iii) light exposure can affect the function of light-sensitive proteins, particularly if the protein has light-sensitive cofactors bound (e.g. PLP or FAD), iv) AzoF may be a critical epitope in in vivo studies.
- In part these limitations might be solved by a cleavable Azo-Tag.
- For light-sensitive proteins/cofactors, elution could be performed during visible light irradiation. This probably reduces the rate of elution, however, complete elution of the POI should still be possible.
- ⇒ Please discuss the mentioned limitations.
- ⇒ If possible, include LEDs emitting visible light in the above suggested experiments.

p.10-11, l.479-482: „The resulting purified protein solutions were directly suitable for subsequent assays, without the need for buffer exchange or removal of elution agents, as here demonstrated with enzyme activity assays and antigen binding assays, respectively (Fig. 3F,G and Suppl. Fig. S11B, C).”

- ⇒ Please include a comparison to the activity of the wildtype enzyme (without Azo-Tag and purified using e.g. Strep-Tag) showing whether the Azo-Tag somehow influences the activity.

Further limitations are: i) the expression yields are reduced with unnatural amino acids (rule-of-thumb: to 10%), ii) the chromatography is difficult to use for xenoproteins since incorporation of two different UAAs is still hard to accomplish.

- ⇒ Please discuss the mentioned limitations.

Minor points

p.2, l.83: “non-natural amino acid (nnAa)”

- ⇒ Please use the established terminology to improve the visibility of this manuscript. It is either non-canonical amino acids (ncAA) or unnatural amino acids (UAA)

p. 2, l. 93: “L-benzazo-phenylalanine (Baf). Baf was previously incorporated (then dubbed AzoPhe)”

Established terminology serves to increase visibility. The described photoswitchable unnatural amino acid is named azobenzene-4'-phenylalanine and its common abbreviation is "AzoF".

⇒ Please use the common term to improve the visibility of this manuscript.

p. 2, l.94-96: "... protein-functional studies into the Escherichia coli catabolite activator protein CAP, into myoglobin from sperm whale and into the ImGPS glutaminase subunit HisH from Thermotoga maritima."

It does not become clear to the reader for what purpose AzoF is generally employed. The field of photoxenoprotein engineering is growing, particularly due to the high interest to control protein activity with light in various research fields.

⇒ I therefore recommend to add a brief description of AzoF, how it works and some examples for how it is used in the introduction (in this regard, please note that there are two more publications on AzoF in ImGPS).

p. 2, l.95: "...into the ImGPS glutaminase."

⇒ Please provide an explanation for the acronym ImGPS

p.5, l.207-211: "..., we chose the pyrrolysyl-tRNA synthetase (PylRS) from Methanosarcina mazei with altered substrate specificity and its cognate tRNA^{Pyl}. This naturally evolved orthogonal pair is known for its more robust amber suppression in E. coli and less cross-reactivity with endogenous canonical amino acids."

- It is unclear to me why the authors chose to evolve a new aaRS for AzoF, when three different ones have already been available: MjTyrRS based from Schultz (DOI 10.1021/ja055467u), MjTyrRS based from Amiram (DOI 10.1002/adfm.202011276) and MbPylRS based from Deiters (DOI 10.1002/cbic.201800226).
 - Although the mentioned advantages of PylRS are true, TyrRSs are generally more suitable for AzoF since they have a larger binding pocket that is better for bulky and hydrophobic UAAs than the one of PylRS. In this regard, it would be interesting to have a direct comparison of the efficiency of all four evolved systems (including the one here).
 - Unfortunately, the only indication of efficiency for the newly evolved PylRS is given in form of the reduced amount of AzoF required for expression (l. 277). It would be good to supplement the yields of protein in mg per expression medium.
 - A discussion comparing the difference of the newly evolved PylRS to the established aaRSs is missing.
- ⇒ The higher efficiency of the newly evolved PylRS for AzoF incorporation over the other three established aaRSs should be proved by a comparison of expression yields with each aaRS.
- ⇒ Please provide a discussion that compares the four aaRSs for AzoF incorporation, possibly including a structural comparison, e.g. provided with AlphaFold.

p.8, l.337-339: "Notably, exposure of the column to 355 nm UV light ... – in the absence of daylight – triggered the elution of both bound Azo-tagged POIs without delay, thus obviating the need for an additional incubation period."

⇒ Please improve this sentence for clarity, particularly the "need for an additional incubation period".